# Context-Aware Kernel Search for Bayesian Optimization with Large Language Models

## Abstract

The efficiency of Bayesian optimization (BO) relies on careful selection of the surrogate model to balance exploration and exploitation under limited budget. Traditional BO methods often struggle with sub-optimal kernel choices when using Gaussian processes (GPs) as the surrogate model. When the kernel is inadequately chosen, BO may converge slowly or even get stuck at an undesired local minimum. To address such drawback, we propose the novel Context-Aware Kernel Search (CAKES) to automate optimal kernel design in BO with large language models (LLMs). Concretely, CAKES exploits LLMs as crossover and mutation operators to adaptively generate and refine GP kernels based on the observed data. CAKES works entirely in-context and can be easily integrated into existing systems without requiring any fine-tuning. We further present a theoretical analysis demonstrating that our method achieves sub-linear regret relative to the budget for any input dimension. Experimental results demonstrate that CAKES outperforms various salient baseline methods in numerous synthetic and real-world optimization tasks. Notably, CAKES improves the overall performance on benchmark functions by roughly 36%. In hyperparameter tuning tasks, CAKES can effectively leverage fewer data samples to quickly identify high-performing configurations and consistently ranks first across various datasets. As an encouraging real application, we successfully applied CAKES to design photonic chips, achieving significant improvements in key performance indicators while speeding up the design cycle by a factor of ten compared to the baselines. Our code is accessible at
`https://github.com/cakes4bo/cakes`.

## 1 Introduction

Many important scientific and engineering problems require optimizing objective functions that are noisy and expensive to evaluate. These objective functions often lack closed-form expressions, let alone gradient information, making optimization particularly difficult (Wang et al., 2023). Nonetheless, Bayesian optimization (BO) has shown remarkable success in optimizing such functions, due to its ability to operate on limited data and incorporate prior knowledge to guide the optimization process (Garnett, 2023). In the past couple of decades, BO has been used for diverse tasks ranging from tuning hyperparameters in machine learning (Snoek et al., 2012; Liu et al., 2024) to designing policies in robotics (Calandra et al., 2016; Martinez-Cantin, 2017) and recommending new molecules in drug discovery (Korovina et al., 2019; Tripp & Hernández-Lobato, 2024). The main idea behind BO is to first construct a *surrogate model*, typically using a Gaussian process (GP, Rasmussen & Williams, 2006), to represent the prior belief about the objective function. Then, by conditioning on the observations and the prior, the posterior is calculated using Bayes' rule to reflect the updated belief about the objective function. Based on this posterior, an *acquisition function* is further used to determine the next promising positions to query by balancing exploration (moving to regions with high uncertainty) and exploitation (moving to regions with high expected value).

Although the past decades have witnessed rapid development of BO, much of the focus has been on designing novel acquisition functions (Ament et al., 2023; Aglietti et al., 2024). In comparison, the challenge of appropriately choosing the surrogate model has received comparatively less attention (Shahriari et al., 2016). Many practitioners simply regard surrogate modeling as a "black art" and perceive it as less critical to the optimization outcome compared to the acquisition function design (Malkomes & Garnett, 2018). In the context of GPs, most off-the-shelf BO methods simply use

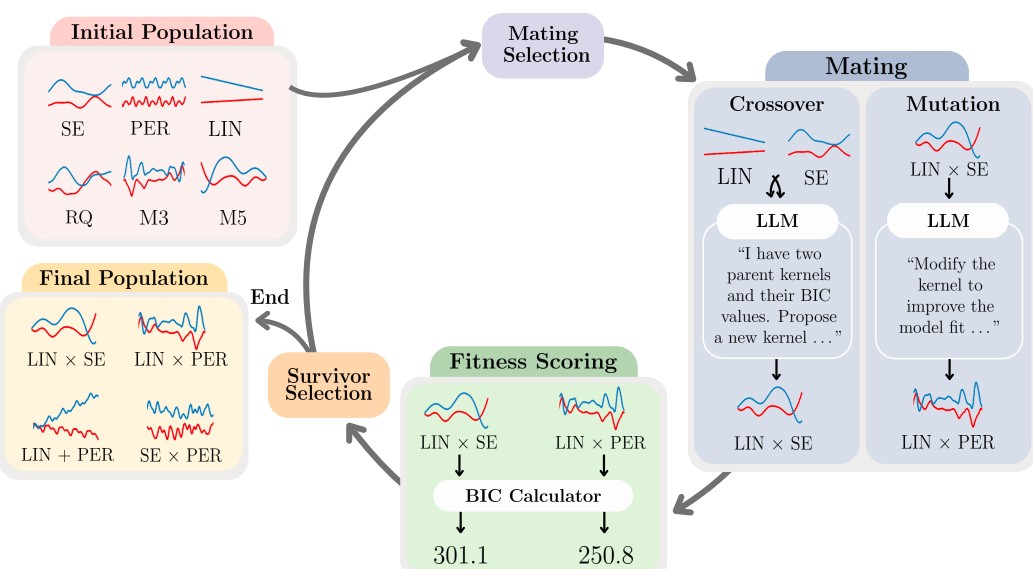

Figure 1: Overview of the proposed CAKES framework. Starting with an initial population of kernels, the LLM acts as crossover and mutation operators, proposing new kernels based on the given prompts. The proposed kernels are then evaluated using a fitness calculator, and the fittest ones advance to the next generation.

general-purpose kernels, such as the squared exponential kernel or Matérn-5/2 kernel (Snoek et al., 2012). While this seems reasonable at the first glance, the so-called *one-size-fits-all* approach has significant limitations. Since the kernel encodes our assumptions about the underlying objective function, the GP will be a poor fit if these assumptions do not match the reality (Roman et al., 2019). It has also been studied that with a poor choice of the kernel, BO may converge very slowly, especially when optimizing complex functions in moderate-to-high dimensional spaces (Gardner et al., 2017). These considerations underscore the need for a more sophisticated kernel design in BO.

While the pursuit for optimal kernel design is not new, existing approaches might not be straightforward in the setting of BO. In BO, the objective function is typically expensive to evaluate, resulting in limited observations for tuning the kernel. The lack of known functional structure and gradient information further complicates the kernel selection process, making it difficult to choose the most suitable kernel using traditional optimization-based methods (Bach, 2008; Gönen & Alpaydin, 2011). At its core, these challenges can be framed within the *few-shot learning* setting, where swift learning and generalization from limited data are required. Notably, these challenges align with the strengths of large language models (LLMs, Wei et al., 2022a), which excel at generalizing from few-shot samples (Brown et al., 2020), thus enabling efficient exploration with limited data. The *in-context learning* capability of LLMs also acts as implicit Bayesian inference (Xie et al., 2022; Han et al., 2023), allowing them to encode prior knowledge about the optimization task, search space, and other relevant information. LLMs' ability in performing complex reasoning further enhances their capacity to process contextual information and improve search performance (Yang et al., 2024). Moreover, LLMs are also pre-trained on massive internet data, which potentially contains transferable domain knowledge applicable to various optimization tasks. Given these insights, we aim to investigate the following question,

> "Can LLMs, with their encoded knowledge and few-shot prompting, help to enhance the kernel design in BO?"

**Contributions.** In this paper, we reveal the great potential of extending the capabilities of LLMs beyond standard natural language tasks to automated kernel design in BO. Our primary contributions are as follows:

- We conduct a thorough study to reveal the urgency for a more careful treatment of kernel design in BO, as the conventional setups, which often rely on fixed kernels or naive selection strategies, do not perform uniformly well across tasks.

- We propose CAKES, an adaptive kernel design method for BO that exploits LLMs as genetic operators to generate and refine kernels based on the observed data (see Figure 1). Our method can be applied entirely in-context and does not require any fine-tuning.

- We present a theoretical analysis based on regret and information gain, demonstrating that our method achieves sub-linear regret relative to the budget for any input dimension.

- We demonstrate that our method consistently outperforms other baselines across numerous optimization tasks, including globally optimizing benchmark functions, tuning hyperparameters of machine learning models, and optimizing photonic chip design.

## 2 PRELIMINARIES

We first review some key concepts in BO and the so-called *kernel grammar*, which serve as the bedrock for the method proposed in this paper.

### 2.1 BAYESIAN OPTIMIZATION

Consider optimizing an objective function $f : \mathcal{X} \subset \mathbb{R}^d \to \mathbb{R}$,

$$\mathbf{x}^* = \arg \max_{\mathbf{x} \in \mathcal{X}} f(\mathbf{x}). \tag{1}$$

Here, $f$ is treated as a "black-box" function, meaning we can only obtain noisy evaluations of the form $y = f(\mathbf{x}) + \epsilon$, where $\epsilon \sim \mathcal{N}(0, \sigma_\epsilon^2)$ represents the Gaussian noise. The evaluations of $f$ are also costly, limiting the number of queries we can make. In addition, the gradient of $f$ is not directly accessible and must be estimated from the noisy evaluations, further complicating the optimization process. To address these challenges, Bayesian optimization (BO) employs a probabilistic surrogate model $g$ to approximate $f$ on the fly (Garnett, 2023). A popular choice for $g$ is the family of Gaussian processes (GP, Rasmussen & Williams, 2006), where a GP can be fully defined by a mean function $m : \mathcal{X} \to \mathbb{R}$ and a positive semi-definite covariance (or kernel) function $k : \mathcal{X} \times \mathcal{X} \to \mathbb{R}$. In practice, the mean function is often set to zero, allowing the GP to be solely characterized by the choice of the kernel function. At each iteration $t$, the calibration of the posterior distribution $p(g_t \mid \mathcal{D}_t)$ given the previous observations $\mathcal{D}_t = \{(\mathbf{x}_i, y_i)\}_{i=1}^{t-1}$ informs where to explore and exploit in $\mathcal{X}$. Then, based on $p(g_t \mid \mathcal{D}_t)$, an acquisition function $\alpha : \mathcal{X} \to \mathbb{R}$ defines a policy to choose the next best point to evaluate. Common choices of $\alpha$ include expected improvement (EI, Jones et al., 1998), upper confidence bound (UCB, Srinivas et al., 2012), and Thompson sampling (TS, Thompson, 1933). After each evaluation, the surrogate model $g$ is refined to reflect the updated belief about $f$. We refer the readers to Appendix B.1 for a thorough treatment of BO with GPs.

### 2.2 KERNEL GRAMMAR

The kernel grammar introduced by Duvenaud et al. (2013) defines a comprehensive and flexible space of kernels. Such space exploits the closure properties of kernel functions under addition and multiplication, which ensures that: if $k_1(\mathbf{x}, \mathbf{x}')$ and $k_2(\mathbf{x}, \mathbf{x}')$ are valid kernels, then both $k_1(\mathbf{x}, \mathbf{x}') + k_2(\mathbf{x}, \mathbf{x}')$ and $k_1(\mathbf{x}, \mathbf{x}') \times k_2(\mathbf{x}, \mathbf{x}')$ are also valid kernels (Smola & Schölkopf, 1998). Starting from a set of base kernels, such as the squared exponential (SE), linear (LIN), periodic (PER), and rational quadratic (RQ), one can construct more expressive kernels by combining such base kernels via the addition and multiplication operators. For instance, LIN + PER kernel can capture periodic structure with a linear trend and SE × PER kernel can capture locally periodic components. Let $\mathcal{B}$ denote a base kernel and $\mathcal{S}$ denote a subexpression, the entire kernel space is described by all kernels that can be constructed using the following grammar rules:

1. Add a base kernel to a subexpression: $\mathcal{S} \to \mathcal{S} + \mathcal{B}$;

2. Multiply a subexpression with a base kernel: $\mathcal{S} \to \mathcal{S} \times \mathcal{B}$;

3. Replace a base kernel with another base kernel: $\mathcal{B} \to \mathcal{B}$.

---

**Algorithm 1:** Context-Aware Kernel Search (CAKES)

---

**Input:** Objective $f$, observations $\mathcal{D}$, budget $T$, number of crossovers $n_c$, probability of mutation $p_m$, population size $n_p$

Initialize the population $\mathbb{K}$

**for** $t = 1, \ldots, T$ **do**

    Feed $\mathcal{D}$ to the LLM with system prompt (see Fig. 4)

    Measure the fitness of each kernel $k \in \mathbb{K}$

    **for** $c = 1, \ldots, n_c$ **do**

        Sample two parent kernels $k_1, k_2$ from $\mathbb{K}$

        Generate kernel through crossover (see Fig. 5a): $k_c \sim \texttt{crossover}(k_1, k_2)$

        Add $k_c$ to $\mathbb{K}$ and measure its fitness

    **if** $\texttt{rand()} < p_m$ **then**

        Select the fittest kernel $k_f$ from $\mathbb{K}$

        Refine kernel through mutation (see Fig. 5b): $k_m \sim \texttt{mutation}(k_f)$

        Add $k_m$ to $\mathbb{K}$ and measure its fitness

    Update population $\mathbb{K}$ by selecting the $n_p$ fittest kernels

    Obtain candidate points: $\mathbf{x}_{t,k} = \arg\max_{\mathbf{x} \in \mathcal{X}} \alpha(\mathbf{x}; \mathcal{D}, k)$ for each $k \in \mathbb{K}$

    Set $k^* = \arg\max_{k \in \mathbb{K}} w_k \alpha(\mathbf{x}_{t,k}; \mathcal{D}, k)$ and let $\mathbf{x}_t = \mathbf{x}_{t,k^*}$

    Update observations: $\mathcal{D} \leftarrow \mathcal{D} \cup \{(\mathbf{x}_t, y_t)\}$

---

## 3 CONTEXT-AWARE KERNEL SEARCH

We propose Context-Aware Kernel Search (CAKES) to automate GP surrogate modeling in BO with LLMs. CAKES uses LLMs as genetic operators to adaptively propose kernels based on the observed data. The main motivation is to refine the kernel choice at each iteration before determining the next query point. The complete procedure of CAKES is summarized in Algorithm 1 and the designed prompts are shown in Appendix D.1.

**Conditioning the LLM.** We begin by randomly sampling $n$ points from the input space $\mathcal{X}$ to initialize the observations $\mathcal{D} = \{(\mathbf{x}_i, y_i)\}_{i=1}^n$. These observations are then used as *few-shot samples* to prompt the LLM for analysis. The prompt is designed based on the concept of conditioning on high performance, as suggested by Zhou et al. (2023). Specifically, we start the prompt with a statement, *"You are an expert in machine learning, specializing in Gaussian processes",* to simulate the reasoning of a human expert in the field. It has also been shown that *chain-of-thought reasoning*, or generating intermediate reasoning steps, can improve the performance of LLMs (Wei et al., 2022b; Kojima et al., 2022). Motivated by this, we instruct the LLM to analyze the provided observations and identify patterns that can be represented by kernel functions, before proposing the kernels at each iteration. To enhance the LLM's understanding, we embed domain-specific knowledge by including examples of kernel functions along with their characteristics.

**Initializing the population.** We draw some inspiration from the genetic algorithm (Holland, 1992), where we maintain a population of candidates (kernels) throughout the optimization process. Our initial population consists of six base kernels: squared exponential (SE), periodic (PER), linear (LIN), rational quadratic (RQ), Matérn-3/2 (M3), and Matérn-5/2 (M5). We denote this initial set as $\mathbb{K} = \{\text{SE}, \text{PER}, \text{LIN}, \text{RQ}, \text{M3}, \text{M5}\}$. For each kernel $k \in \mathbb{K}$, we measure its *fitness* using the Bayesian Information Criterion (BIC, Schwarz, 1978), which balances model fit and complexity. BIC is commonly used for model selection and can be seen as an approximation to the Laplace approximation (Murphy, 2022). More detailed discussions on BIC and model selection for GPs can be found in Appendix B.3.

**Proposing the kernels.** We consider a generalized notion of the kernel grammar, which involves a set of base kernels $\{\mathcal{B}_1, \ldots, \mathcal{B}_r\}$ and a set of operators $\{\mathcal{T}_1, \ldots, \mathcal{T}_l\}$, where $r, l \in \mathbb{N}$. Each operator $\mathcal{T}_j : \mathcal{K} \times \mathcal{K} \to \mathcal{K}$, for $j = 1, \ldots, l$, is a closed operator (e.g., addition and multiplication) on the space of kernels $\mathcal{K}$. The grammar rules can be summarized in the following way:

1. Apply operator $\mathcal{T}_j$ on a subexpression and a base kernel: $\mathcal{S} \to \mathcal{T}_j(\mathcal{S}, \mathcal{B})$;

2. Replace a base kernel with another base kernel: $\mathcal{B} \to \mathcal{B}$.

Based on these grammar rules, we can formally define the kernel grammar space as follows:

**Definition 1.** *Let $\{k_1, \ldots, k_r\}$, where $r \in \mathbb{N}$, be a set of base kernels, and let $\{\mathcal{T}_1, \ldots, \mathcal{T}_l\}$, where $l \in \mathbb{N}$, be a set of operators defined by $\mathcal{T}_j : \mathcal{K} \times \mathcal{K} \to \mathcal{K}$ for $j = 1, \ldots, l$. For $i \in \mathbb{N}$, we can define the kernel grammar space recursively as follows:*

$$\mathbb{K}_0 := \{k_1, \ldots, k_r\},$$
$$\mathbb{K}_i := \{\mathcal{T}_j(k_1, k_2) \,|\, k_1, k_2 \in \mathbb{K}_{i-1}, j = 1, \ldots, l\} \cup \mathbb{K}_{i-1}.$$

When we refer to a kernel $k$ from the kernel grammar space, we actually refer to the associated kernel family over its hyperparameters $\{k_{\boldsymbol{\theta}} \,|\, \boldsymbol{\theta} \in \Theta\}$. Hence, if we apply an operator $\mathcal{T}$ on two kernels $k_1$ and $k_2$, we obtain another kernel family $\{\mathcal{T}(k_{1,\boldsymbol{\theta}_1}, k_{2,\boldsymbol{\theta}_2}) \,|\, \boldsymbol{\theta}_1 \in \Theta_1, \boldsymbol{\theta}_2 \in \Theta_2\}$. By leveraging this kernel grammar space, we can use the LLM as *genetic operators* to propose kernels through the following steps:

- **Crossover:** We perform $n_c$ crossover operations. For each crossover, we sample two parent kernels $k_1, k_2$ from $\mathbb{K}$ with probability proportional to their fitness. We then prompt the LLM to propose a new kernel $k_c$ that has a lower BIC value, by applying an operator on the parent kernels.

- **Mutation:** With probability $p_m$, we perform a mutation operation. We select the fittest kernel $k_f$ from $\mathbb{K}$ and prompt the LLM to suggest a new kernel $k_m$, by replacing one of the base kernels in $k_f$ with another base kernel.

In the prompts, we also ask the LLM to report a brief analysis of its reasoning behind the proposed kernels. This serves as a sanity check, enabling us to verify and interpret the choices made by the LLM. The proposed kernels from both steps are added to $\mathbb{K}$, and their fitnesses are measured. Then, we select the top $n_p$ fittest kernels to form the next generation of $\mathbb{K}$.

**Choosing the next query point.** In our experiments, we observed that some kernels may promise a good fit, but the actual improvement is not as substantial as expected when the proposed query points are evaluated. For this reason, we do not always rely on the fittest kernel to choose the next query point. Instead, we adopt the following weighting strategy,

$$k^* = \arg\max_{k \in \mathbb{K}} w_k \alpha(\mathbf{x}_{t,k}; \mathcal{D}, k), \tag{2}$$

where $w_k = \exp(-\mathrm{BIC}_k) / \sum_{j \in \mathbb{K}} \exp(-\mathrm{BIC}_j)$ and $\mathbf{x}_{t,k}$ denotes the proposed query point for that kernel. This strategy allows us to balance the kernel's ability to fit the data (as indicated by $w_k$) with the expected improvement at the proposed query point (as measured by $\alpha$). After determining $k^*$, we set the next query point at $\mathbf{x}_t = \mathbf{x}_{t,k^*}$ and update the observations as $\mathcal{D} \leftarrow \mathcal{D} \cup \{(\mathbf{x}_t, y_t)\}$. This iterative process continues until a predefined budget $T$ is exhausted.

## 4 THEORETICAL ANALYSIS

To establish the theoretical guarantee of our proposed method, we will use *regret* as a measure of convergence and *information gain* to assess the informativeness of the data samples. The instantaneous regret at iteration $t$ is defined as $r_t = f(\mathbf{x}^*) - f(\mathbf{x}_t)$, and the corresponding cumulative regret after $T$ iterations is $R_T = \sum_{t=1}^{T} r_t$. If we can show that the cumulative regret is *sub-linear* for a given algorithm, then the algorithm converges to a value arbitrarily close to the optimal (Berkenkamp et al., 2019). To measure the reduction in uncertainty about $f$ from observing $\mathbf{y}_{\mathcal{A}}$ for a set of points $\mathcal{A} = \{\mathbf{x}_1, \ldots, \mathbf{x}_T\} \subset \mathcal{X}$, we use information gain, which is defined as the mutual information between $f$ and $\mathbf{y}_{\mathcal{A}}$,

$$\mathcal{I}(\mathbf{y}_{\mathcal{A}}; f) = H(\mathbf{y}_{\mathcal{A}}) - H(\mathbf{y}_{\mathcal{A}} \,|\, f), \tag{3}$$

where $H(\cdot)$ denotes the entropy function. Following Wang & de Freitas (2014), we can define the *maximum information gain* after $T$ iterations as,

$$\gamma_T = \max_{\mathcal{A} \subset \mathcal{X} : |\mathcal{A}| = T} \mathcal{I}(\mathbf{y}_{\mathcal{A}}; f) = \max_{\mathcal{A} \subset \mathcal{X} : |\mathcal{A}| = T} \frac{1}{2} \log |\mathbf{I} + \sigma_\epsilon^{-2} \mathbf{K}_{\mathcal{A};\boldsymbol{\theta}}|, \tag{4}$$

where $\mathbf{I}$ is the identity matrix and $\mathbf{K}_{\mathcal{A};\boldsymbol{\theta}}$ is the kernel matrix evaluated at the observed points. Before presenting our main result, we first make the following assumption:

**Assumption 1** (Theorem 5 of (Ghosal & Roy, 2006)). *Let $\mathcal{X} \subset \mathbb{R}^d$ be compact and convex. There exist constants $a, b, L > 0$ such that for any $f(\mathbf{x}) \sim \mathcal{GP}(\mathbf{0}, k(\mathbf{x}, \mathbf{x}'))$,*

$$\Pr\left\{\sup_{\mathbf{x} \in \mathcal{X}} \left| \frac{\partial f(\mathbf{x})}{\partial x_j} \right| > L \right\} \leq ae^{-(L/b)^2}, \ \ \forall j = 1,,\ldots,d. \tag{5}$$

The cumulative regret can be bounded according to the following theorem:

**Theorem 1.** *Pick $\delta \in (0, 1)$. Under Assumption 1, with probability $1 - \delta$, the cumulative regret of CAKES over $T$ iterations is bounded by,*

$$R_T \leq \sqrt{C_1 T \beta_T \gamma_T} + \frac{\pi^2}{6}, \ \ \forall T \geq 1,$$

*where $C_1 = 8/(\log 1 + \sigma_\epsilon^{-2})$ and $\beta_t = 2\log(t^2 2\pi^2/(3\delta)) + 4d\log(dtbr\sqrt{\log(4da/\delta)})$.*

The above theorem demonstrates that, with high probability, the cumulative regret of our method is bounded in terms of the maximum information gain $\gamma_T$. We can further bound $\gamma_T$ using the following results from Srinivas et al. (2012):

**Theorem 2** (Theorem 5 of (Srinivas et al., 2012)). *Let $\mathcal{X} \subset \mathbb{R}^d$ be compact and convex, $d \in \mathbb{N}$. Assume the kernel function satisfies $k(\mathbf{x}, \mathbf{x}') \leq 1$.*

  1. *For the $d$-dimensional linear kernel: $\gamma_T = \mathcal{O}(d\log T)$.*

  2. *For the squared exponential kernel: $\gamma_T = \mathcal{O}((\log T)^{d+1})$.*

  3. *For Matérn kernels with $\nu > 1$: $\gamma_T = \mathcal{O}(T^{d(d+1)/(2\nu+d(d+1))}(\log T))$*

Combining these results with Theorem 1, our main result can be summarized as follows:

**Theorem 3.** *With high probability, the cumulative regret of CAKES is bounded by $\tilde{\mathcal{O}}(\sqrt{dT\gamma_T})$, where $\tilde{\mathcal{O}}$ suppresses the logarithmic factors. Applying the information gain bounds from Theorem 2, this yields sub-linear regret in $T$ for any dimension $d$.*

We provide the detailed proof of our results in Appendix C. We also provide additional empirical analysis in Section F.

## 5 RELATED WORK

**Kernel design.** Traditionally, kernel design relied on visually inspecting patterns in the data and manually combining elementary kernels, such as squared exponential kernel and periodic kernel (Rasmussen & Williams, 2006). However, this approach is highly subjective and requires expert knowledge, making it impractical for complex data patterns. As a result, several methods have been developed to automate the kernel selection process. One such method involved multiple kernel learning techniques, which leverage data to identify the optimal kernel configuration by optimizing a linear or nonlinear combination of elementary kernels (Bach, 2008; Gönen & Alpaydin, 2011). Yet, these methods impose restrictions on the kernel space and require the hyperparameters to be specified in advance. Another line of research focused on designing flexible kernel families that can approximate arbitrary stationary kernels in the frequency domain (Lázaro-Gredilla et al., 2010; Wilson & Adams, 2013; Yin et al., 2020). While these methods can capture a wide range of kernels, they are limited by the stationarity assumption. Other works have also attempted to develop more expressive kernels by integrating GPs with deep neural networks (Damianou & Lawrence, 2013; Wilson et al., 2016). Unfortunately, these methods often require complex inference techniques that are computationally more expensive than traditional kernels. In contrast to these approaches, our method is based on the kernel grammar and in-context learning with pre-trained LLMs, which offers a flexible yet computationally feasible approach.

**Surrogate modeling in BO.** When using GPs as the surrogate model in BO, the kernel is typically selected *a priori* based on expert knowledge about the problem at hand. Unfortunately, when there is no prior knowledge available, most BO methods simply use default kernels such as squared exponential kernel or Matérn-5/2 kernel (Snoek et al., 2012). While this seems reasonable, it has

Table 1: Average normalized regret (↓) over 20 random seeds for different test functions and methods. The best value is highlighted in **bold** and the second best value is underlined.

| Function | Fixed | | Adaptive | | | DGP | EGP | CAKES |
|---|---|---|---|---|---|---|---|---|
| | SE | M5 | Random | Utility | BIC | | | |
| Ackley-2 | 0.1773 | 0.1220 | 0.1358 | 0.1062 | 0.1863 | 0.2510 | 0.1878 | **0.0783** |
| Ackley-5 | 0.3185 | 0.2369 | **0.1722** | 0.2278 | 0.3590 | 0.3110 | 0.2285 | 0.1732 |
| Beale | 0.3554 | 0.3522 | 0.2855 | 0.4410 | 0.3571 | 0.4775 | 0.3088 | **0.2565** |
| Branin | 0.0183 | 0.0055 | 0.0227 | 0.0372 | 0.0371 | 0.4810 | 0.2045 | **0.0070** |
| Dropwave | 0.5110 | 0.5411 | 0.5460 | 0.5265 | 0.5461 | 0.5560 | 0.6290 | **0.4690** |
| Eggholder | 0.4941 | 0.3545 | 0.4015 | 0.4855 | 0.5485 | 0.4535 | 0.4345 | **0.1241** |
| Griewank-2 | 0.1196 | 0.1282 | 0.1295 | 0.1310 | 0.1272 | 0.1156 | 0.0935 | **0.0267** |
| Griewank-5 | 0.0204 | 0.0223 | 0.0232 | **0.0178** | 0.0281 | 0.0815 | 0.0478 | 0.0185 |
| Hartmann | 0.0007 | 0.0019 | 0.0021 | 0.0358 | 0.6800 | 0.1305 | 0.1780 | **0.0001** |
| Levy-2 | 0.1562 | 0.0418 | 0.0835 | **0.0255** | 0.1145 | 0.1965 | 0.0765 | 0.0353 |
| Levy-3 | 0.1141 | 0.1422 | 0.1495 | 0.0880 | 0.1125 | 0.2265 | 0.0805 | **0.0505** |
| Rastrigin-2 | 0.4325 | 0.4251 | 0.5310 | 0.3455 | 0.4490 | 0.5405 | 0.3869 | **0.3341** |
| Rastrigin-4 | 0.5765 | 0.5461 | 0.4815 | 0.5905 | 0.5200 | 0.5340 | 0.3270 | **0.3128** |
| Rosenbrock | 0.1025 | 0.0898 | 0.1015 | 0.1405 | 0.1475 | 0.5340 | 0.6040 | **0.0483** |
| Six-Hump Camel | 0.2840 | 0.1507 | 0.3455 | 0.3310 | 0.3265 | 0.4940 | 0.5345 | **0.1015** |
| Mean regret | 0.2454 | 0.2111 | 0.2271 | 0.2355 | 0.3021 | 0.3585 | 0.2881 | **0.1358** |
| Median regret | 0.1773 | 0.1422 | 0.1495 | 0.1400 | 0.3265 | 0.4530 | 0.2285 | **0.0783** |

been studied that with poor or overly general choices of the kernel, BO can converge very slowly (Gardner et al., 2017). For this reason, deep GPs (DGPs) have been proposed as a promising alternative to traditional GPs in BO, particularly for capturing non-stationary behaviors (Hebbal et al., 2021). However, DGPs suffer from high computational complexity, which grows with the number of layers. Other prior works have explored adaptively changing the kernel during the optimization process. For example, Ginsbourger et al. (2008) used a discrete mixture of GPs, while Viana et al. (2013) and Roman et al. (2019) maintained multiple GPs with different kernels in parallel. More recently, Lu et al. (2023) proposed using an ensemble of GPs to adaptively select the surrogate model by adjusting the weights on the fly. We extend this line of research by using pre-trained LLMs to automatically generate and refine kernels during the optimization process, enabling a new-fashioned adaptive kernel design.

**LLMs as genetic operators.** As the model size and amount of training data increase, LLMs exhibit several emergent abilities that improve their performance across diverse tasks (Wei et al., 2022a). One such ability is few-shot learning, where the model can generalize from a few samples without adjusting its parameters (Brown et al., 2020). This generalization is achieved through in-context learning, in which the model identifies the patterns in the samples and extrapolates them via next-token prediction. Inspired by these abilities, recent works have explored the use of LLMs for optimization tasks, notably as genetic operators in evolutionary algorithms. For example, Meyerson et al. (2024) utilized LLMs to propose evolutionary crossovers on tasks such as image and code generation, while Lehman et al. (2023) trained an LLM on computer programs to act as a mutation operator for robot simulations. Chen et al. (2024) combined evolutionary search with soft prompt tuning to evolve neural network architectures using LLMs. To the best of our knowledge, the current work is the first to use LLMs as genetic operators for GP kernel design in BO. Compared to the state-of-the-art, our method can be applied entirely in-context and does not require any fine-tuning.

# 6 EXPERIMENTS

We evaluate the performance of our proposed method against several baselines across different tasks, including globally optimizing benchmark functions, tuning hyperparameters of machine learning models, and designing photonic chips with optimal configurations.

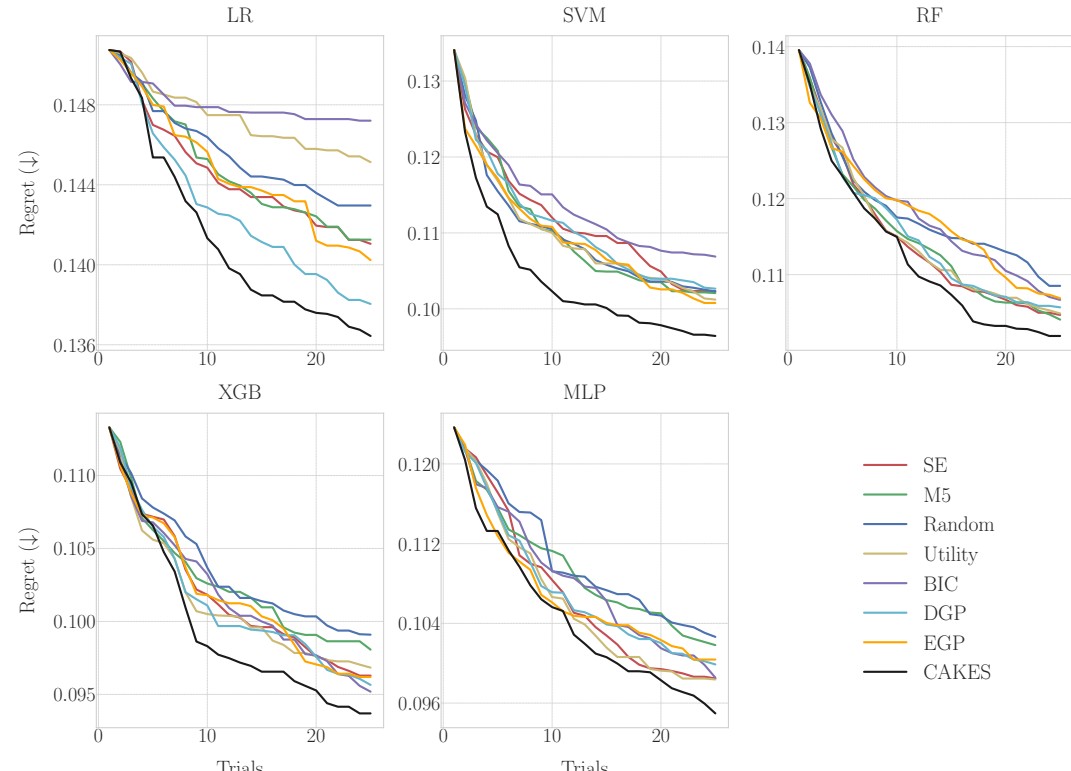

Figure 2: Average regret (↓) over 5 random seeds for different ML models on all datasets.

**Setup.** Our experiments are conducted using the software package BoTorch (Balandat et al., 2020), where we use the expected improvement (EI) as the acquisition function. For the LLM, we use OpenAI's `gpt-4o-mini` model, with the hyperparameters `temperature=0.7` and `top_p=0.95`. We give additional insights on the selection of these hyperparameters in Appendix D.1. To facilitate reproducibility, our code is publicly available at `https://github.com/cakes4bo/cakes`.

**Baselines.** We compare our proposed method against the following baselines:

- **Fixed:** This is the default method in BO, where we fix the kernel throughout the optimization process. We consider the two most commonly-used kernels in BO: squared exponential (SE) and Matérn-5/2 (M5).
- **Adaptive:** This is the adaptive kernel selection method proposed in (Roman et al., 2019). We employ the same six base kernels as in CAKES and use three selection criteria to adaptively change the kernel: *Random*, *Utility*, and *BIC*.
- **Deep GP:** This baseline uses a deep GP (DGP) as the surrogate model, implemeted through a functional composition of stationary GPs (Hebbal et al., 2021).
- **Ensemble GP:** This baseline uses an ensemble of GPs (EGP) to adaptively select the surrogate model (Lu et al., 2023). The kernel dictionary consists of the same six kernels used in CAKES.

In the interest of space, we provide more details of the chosen baselines in Appendix D.2.

### 6.1 OPTIMIZATION OF BENCHMARK FUNCTIONS

**Setup.** Our first set of tasks involve optimizing test functions commonly used as benchmark for optimization (Surjanovic & Bingham, 2013). The goal is to find the global minimum of each test function, where the maximum number of function evaluations is limited to 10 times the dimensionality of the function domain, i.e., $T = 10 \times d$. We provide more information about the chosen functions in Appendix E.1.

**Evaluation.** To evaluate the performance of each method, we consider the normalized regret (Arango et al., 2021):

$$\frac{f(\mathbf{x}_{\text{opt}}) - f(\mathbf{x}_{\text{best}})}{f(\mathbf{x}_{\text{opt}}) - f(\mathbf{x}_{\text{init}})}, \tag{6}$$

where $f(\mathbf{x}_{\text{init}})$ is the best function value among the initial points, $f(\mathbf{x}_{\text{best}})$ is the best value found by the method, and $f(\mathbf{x}_{\text{opt}})$ is the ground truth optimum. This metric is useful as it provides a normalized and task-agnostic measure to compare the optimization performance across different tasks.

**Results.** Table 1 shows the normalized regret averaged over 20 seeds for different functions and methods. Our results demonstrate that CAKES outperforms the baselines, achieving roughly a 36% improvement in the mean regret and a 44% improvement in the median regret compared to the runner-up method. Notably, CAKES ranks among the top two for all functions and achieves the best performance in 12 out of 15 functions. Our results also reveal that fixed kernels are not universally effective, underscoring the need for more adaptive and problem-specific methods. Specifically, the default kernels in BO like SE and M5, actually underperform on many test functions. Due to space limitations, we include additional results in Appendix G.1.

## 6.2 HYPERPARAMETER TUNING TASK

**Setup.** We consider the hyperparameter tuning tasks available in the HPOBench package (Eggensperger et al., 2021). We included a total of 60 tasks, comprising 12 OpenML datasets and 5 machine learning (ML) models: logistic regression (LR), support vector machine (SVM), random forest (RF), XGBoost (XGB), and multi-layer perceptron (MLP). We show more details on these tasks in Appendix E.2.

**Evaluation.** Our goal for each task is to maximize the accuracy of the ML model on the unseen test data. The performance of each method is measured by the simple regret, $f(\mathbf{x}^*) - \max_{1 \leq i \leq t} f(\mathbf{x}_i)$. We executed $T = 25$ trials using five different random seeds, where we ensure that all models share the same initialization for each seed.

**Results.** Figure 2 shows the regret for all tasks averaged over 5 random seeds. The results demonstrate that CAKES consistently achieves the lowest regret, and therefore the best tuning performance, compared to the other methods across all tasks. It is also worth noting that CAKES excels in the earlier stages of the optimization process, when fewer observations are available. This suggests that CAKES is able to effectively leverage fewer data samples to quickly converge to high-performing configurations. Table 8 further highlights the superior performance of CAKES, which achieves the best average rank at the end of each tuning task. Our results also reveal significant variations in performance for the fixed and adaptive kernel approaches. For instance, the SE kernel performs reasonably well in tuning XGB and MLP models, but struggles with tuning SVM and RF. In contrast, CAKES demonstrates more consistent performance across all tasks, exhibiting less sensitivity to the specific model being optimized. To provide a more comprehensive comparison, we report the individual task results for each ML model and dataset in Appendix G.2. We have also included two additional experiments that involve tuning a controller for the robot pushing task and the lunar lander problem in Appendix G.4 and G.5 respectively.

## 6.3 PHOTONIC CHIP DESIGN

**Motivation.** To further evaluate our method in real-world optimization tasks, we consider optimizing the design parameters of photonic chips. This task has become a challenging problem in physics and engineering (Li et al., 2023; Zhang et al., 2024), as it is infeasible to try all the combinations (or even a few) of the parameters due to the high cost of the fabrication process. As a result, one has to rely on extensive computer simulations to assess the chip's performance without knowing any gradient information or explicit form of the objective function. This task can be formulated as an inverse design problem, where the goal is to optimize the chip to meet specific performance criteria.

**Setup.** In this problem, we consider five key indicators for assessing the chip's performance: Q-factor ($f_1$), wavelength ($f_2$), lasing area ($f_3$), power ($f_4$), and divergence angle ($f_5$). Based on these

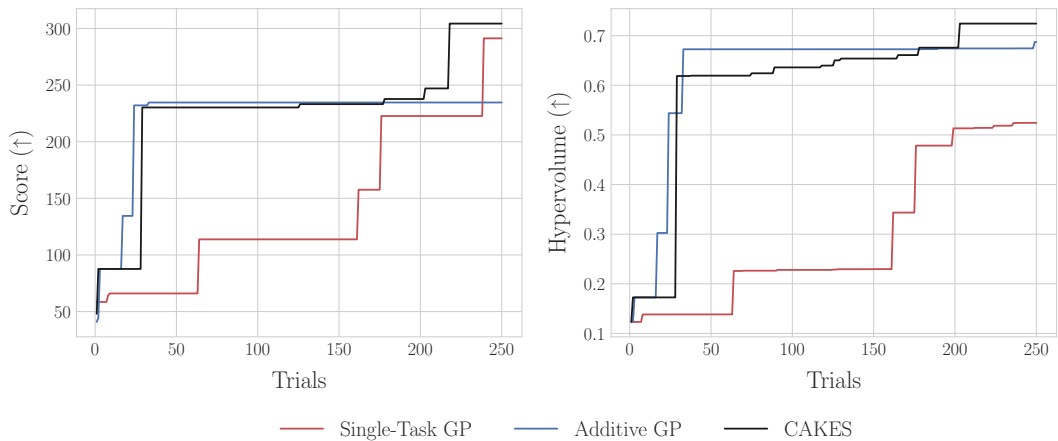

Figure 3: Score (↑) and hypervolume (↑) of the designed chip over 250 trials.

performance indicators, we can calculate the overall score for a given set of parameters:

$$\alpha f_1 + \beta f_2 + \gamma f_3 + \delta f_4 + \epsilon f_5, \tag{7}$$

where we set $\alpha = \beta = 1$, $\gamma = \delta = 100$, and $\epsilon = 20$ to unify the dimensions between different objectives. We provide more details about the objectives and problem setup in Appendix E.3.

**Baselines.** For the baselines, we consider two widely-used BO-based methods in the literature. The first is the *Single-Task GP*, which serves as the default method, where a single Gaussian Process (GP) is employed to model each objective independently, typically utilizing the M5 kernel. The second method is the *Additive GP*, which models the overall objective as a sum of independent GPs, with each GP utilizing a squared exponential (SE) kernel. We provide more comparisons with additional baselines from the literature in Appendix G.3. For all methods, we set $T = 250$ with 6 different random initializations and use the expected hypervolume improvement (EHVI) as the acquisition function to identify the Pareto optimal solution (Couckuyt et al., 2014; Yang et al., 2019). We refer the reader to Appendix B.4 for more detailed definitions of the Pareto optimal solution and hypervolume.

**Results.** Figure 3 shows the score and hypervolume of the designed chip obtained by CAKES against competing baselines. Compared to the baselines, CAKES achieved the highest score and hypervolume, resulting in the best overall chip performance. The superiority of CAKES likely stems from its ability to utilize different kernels tailored to specific objectives, unlike competing methods that rely on a single kernel to model all objectives. Moreover, we can observe that the score of the chip was significantly improved by CAKES in less than 30 trials, which is equivalent to a tenfold speedup in the design cycle compared to the baselines. This could lead to significant reductions in both time and cost needed to design the chips, thus accelerating the research and development (R&D) process. We showcase the individual results on each objective in Appendix G.3.

## 7 CONCLUSION

We introduced CAKES, a novel kernel design method that integrates LLMs to automate surrogate modeling in BO. By leveraging LLMs as crossover and mutation operators, CAKES adaptively proposes kernels based on the observed data, bridging the gap between kernel selection and domain expertise. Our theoretical analysis demonstrates that CAKES achieves sub-linear regret relative to the budget for any input dimension. Experimental results strongly support our theoretical findings. In benchmark function optimization, CAKES consistently ranked among the top two for all test functions, achieving the best performance in 12 out of 15 cases. In hyperparameter tuning tasks, CAKES obtained the lowest regret across all models, particularly excelling in the early stage of optimization with fewer observations. In the photonic chip design problem, CAKES significantly enhanced chip performance in fewer than 30 trials, resulting in a tenfold speedup of the design cycle compared to baseline methods. We believe this work represents a solid step towards a fully automated BO system that can be used by non-experts on arbitrary objectives.

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

## A ETHICS AND REPRODUCIBILITY STATEMENTS

**Ethics.** In this work, we follow the recommendations from Azure OpenAI service when using GPT models, where we ensure via the agreement that sensitive data is not sent for human review or stored, thereby respecting the guidelines given by the dataset providers.

**Reproducibility.** Experimental results are shown in Section 6 with further details of the implementations, baselines, and benchmarks, included in Appendix D and Appendix E. Additional results are also given in Appendix G. We provide the code to reproduce our results at `https://github.com/cakes4bo/cakes`.

## B MATHEMATICAL DETAILS

### B.1 BAYESIAN OPTIMIZATION WITH GAUSSIAN PROCESSES

A Gaussian process (GP) describes a collection of random variables, any finite number of which have a joint Gaussian distribution (Rasmussen & Williams, 2006). Mathematically, a GP can be expressed as, $\mathcal{GP}\left(m(\mathbf{x}), k_{\boldsymbol{\theta}}(\mathbf{x}, \mathbf{x}'; \boldsymbol{\theta})\right)$, where $m(\mathbf{x})$ is the mean function and $k_{\boldsymbol{\theta}}(\mathbf{x}, \mathbf{x}'; \boldsymbol{\theta})$ is the covariance (kernel) function parameterized by some hyperparameters $\boldsymbol{\theta}$. Given any finite collection of inputs $\mathbf{X} = \mathbf{x}_{1:t}$, the outputs are jointly Gaussian,

$$f(\mathbf{X}) \sim \mathcal{N}\left(\boldsymbol{m}_{\mathbf{X}}, \mathbf{K}(\mathbf{X}, \mathbf{X}; \boldsymbol{\theta})\right), \tag{8}$$

where $\boldsymbol{m}_{\mathbf{X}} = m(\mathbf{X}) \in \mathbb{R}^t$ is the mean function vector evaluated at $\mathbf{X}$, often assumed to be $\mathbf{0}$ in practice, and $\mathbf{K}(\mathbf{X}, \mathbf{X}; \boldsymbol{\theta}) \in \mathbb{R}^{t \times t}$ is the covariance matrix with entries $[\mathbf{K}(\mathbf{X}, \mathbf{X}; \boldsymbol{\theta})]_{i,j} = k_{\boldsymbol{\theta}}(\mathbf{x}_i, \mathbf{x}_j)$. We assume that the evaluations of $f$ at any point $\mathbf{x}_t$ are corrupted by a $\sigma$-sub-Gaussian noise,

$$y_t = f(\mathbf{x}_t) + \epsilon_t, \tag{9}$$

where $\epsilon_t \sim \mathcal{N}(0, \sigma_\epsilon^2)$. Given the observed data $\mathcal{D}_t = \{\mathbf{X}, \mathbf{y}\}$, where $\mathbf{y} = \mathbf{y}_{1:t}$, the joint Gaussian distribution of the observed data and an arbitrary query point $\mathbf{x}$ is:

$$\begin{bmatrix} \mathbf{y} \\ f(\mathbf{x}) \end{bmatrix} \sim \mathcal{N}\left(\mathbf{0}, \begin{bmatrix} \mathbf{K}_{t;\boldsymbol{\theta}} + \sigma_\epsilon^2 \mathbf{I} & \mathbf{k}_{t;\boldsymbol{\theta}}(\mathbf{x}) \\ \mathbf{k}_{t;\boldsymbol{\theta}}^\top(\mathbf{x}) & k_{\boldsymbol{\theta}}(\mathbf{x}, \mathbf{x}) \end{bmatrix}\right), \tag{10}$$

where $\mathbf{K}_{t;\boldsymbol{\theta}} = \mathbf{K}(\mathbf{X}, \mathbf{X}; \boldsymbol{\theta})$ and $\mathbf{k}_{t;\boldsymbol{\theta}}(\mathbf{x}) = \mathbf{k}_{\boldsymbol{\theta}}(\mathbf{X}, \mathbf{x})$. It follows that, the posterior distribution of any query point $\mathbf{x}$ is marginally Gaussian,

$$f(\mathbf{x})|\mathcal{D}_t; \boldsymbol{\theta} \sim \mathcal{N}(\mu_t(\mathbf{x}; \boldsymbol{\theta}), \sigma_t^2(\mathbf{x}; \boldsymbol{\theta})), \tag{11}$$

where

$$\mu_t(\mathbf{x}; \boldsymbol{\theta}) = \mathbb{E}[f(\mathbf{x})|\mathcal{D}_t] = \mathbf{k}_{t,\boldsymbol{\theta}}^\top(\mathbf{x})(\mathbf{K}_{t;\boldsymbol{\theta}} + \sigma_\epsilon^2 \mathbf{I})^{-1}\mathbf{y}, \tag{12a}$$

$$\sigma_t^2(\mathbf{x}; \boldsymbol{\theta}) = \mathbb{E}[f(\mathbf{x})f(\mathbf{x})|\mathcal{D}_t] = \mathbf{k}_{\boldsymbol{\theta}}(\mathbf{x}, \mathbf{x}) - \mathbf{k}_{t,\boldsymbol{\theta}}^\top(\mathbf{x})(\mathbf{K}_{t;\boldsymbol{\theta}} + \sigma_\epsilon^2 \mathbf{I})^{-1}\mathbf{k}_{t,\boldsymbol{\theta}}(\mathbf{x}). \tag{12b}$$

Based on the posterior distribution, the acquisition function must use its statistics to trade-off exploitation (where $\mu_t(\mathbf{x}; \boldsymbol{\theta})$ is high) and exploration (where $\sigma_t^2(\mathbf{x}; \boldsymbol{\theta})$ is high) effectively. Among the various acquisition functions proposed, expected improvement (EI) remains the default choice in many BO applications (Snoek et al., 2012). Let us define $\mu_{\boldsymbol{\theta}}^+ = \max_{\mathbf{x} \in \mathcal{X}} \mu_t(\mathbf{x}; \boldsymbol{\theta})$ as the best mean value. The EI acquisition function can then be expressed in closed form as:

$$\alpha(\mathbf{x}; \mathcal{D}_t) = \mathbb{E}[\max\{0, f(\mathbf{x}) - \mu_{\boldsymbol{\theta}}^+\}] = \sigma_t(\mathbf{x}; \boldsymbol{\theta})[u\Phi(u) + \phi(u)], \tag{13}$$

where $u = (\mu_t(\mathbf{x}; \boldsymbol{\theta}) - \mu_{\boldsymbol{\theta}}^+)/\sigma_t(\mathbf{x}; \boldsymbol{\theta})$, and $\phi(\cdot)$ and $\Phi(\cdot)$ are the standard normal density and cumulative distribution functions, respectively.

### B.2 SUB-GAUSSIAN NOISE

We say $\epsilon$ is $\sigma$-sub-Gaussian if there exists $\sigma \geq 0$ such that,

$$\mathbb{E}[\exp(\rho\epsilon)] \leq \exp\left(\frac{\rho^2 \sigma^2}{2}\right), \quad \forall \rho \in \mathbb{R}. \tag{14}$$

It is easy to show that if $\epsilon$ is $\sigma$-sub-Gaussian, then $\mathbb{E}[\epsilon] = 0$ and $\text{Var}[\epsilon] \leq \sigma^2$. Examples of sub-Gaussian variables include, zero-mean Gaussian random variables with variance $\sigma^2$, symmetric Bernoulli random variables, and symmetric uniform distributions (Bach, 2024).

### B.3 MODEL SELECTION

We conduct model selection over a discrete, infinite space of kernels $\mathcal{K} = \{k_1, k_2, \ldots\}$. As each kernel comes with its own hyperparameters, we are actually dealing with a space of kernel families. Thus, when referring to a kernel $k$, we actually associate it with its whole family over hyperparameters $\{k_{\boldsymbol{\theta}} \mid \boldsymbol{\theta} \in \Theta\}$. Given some model selection criteria $h : \mathcal{K} \to \mathbb{R}$, our goal is to identify the optimal kernel,

$$k^* = \arg\max_{k \in \mathcal{K}} h(k \mid \mathcal{D}). \tag{15}$$

A commonly-used criterion for probabilistic models, such as GPs, is the marginal log-likelihood (Murphy, 2022),

$$h(k \mid \mathcal{D}) = \log p(\mathbf{y} \mid \mathbf{X}, k) = \log \int p(\mathbf{y} \mid \mathbf{X}, \boldsymbol{\theta}, k) p(\boldsymbol{\theta}) d\boldsymbol{\theta}. \tag{16}$$

Unfortunately, the above likelihood is generally intractable for GPs (Rasmussen & Williams, 2006), so we resort to the Laplace approximation,

$$\log p(\mathbf{y} \mid \mathbf{X}, k) \approx \log p(\mathbf{y} \mid \mathbf{X}, \hat{\boldsymbol{\theta}}, k) + \log p(\hat{\boldsymbol{\theta}}) - \frac{1}{2} \log \det \boldsymbol{\Sigma}^{-1} + \frac{d_{\boldsymbol{\theta}}}{2} \log 2\pi \tag{17}$$

where $\hat{\boldsymbol{\theta}}$ denotes the maximum a posteriori (MAP) estimate of the hyperparameters with $d_{\boldsymbol{\theta}}$ being its dimension. The term $\boldsymbol{\Sigma}^{-1} = -\nabla^2 \log p(\boldsymbol{\theta} \mid \mathcal{D}, k)|_{\boldsymbol{\theta}=\hat{\boldsymbol{\theta}}}$ represents the Hessian matrix evaluated at the MAP estimate. Note that Eq. (17) can be interpreted as rewarding model fit while penalizing model complexity. In our experiments, we use the BIC (Schwarz, 1978), which was also employed by Duvenaud et al. (2013) and can be viewed as an approximation of the Laplace approximation.

### B.4 MULTI-OBJECTIVE OPTIMIZATION

**Setup.** In our photonic chip inverse design problem, the solutions need to satisfy multiple, potentially competing objectives simultaneously. Multi-objective optimization aims to find the *Pareto optimal solution*, where no objective can be improved without degrading another (Lin et al., 2022). A simplistic approach to jointly optimize the given objectives $\{f_i(\cdot)\}_{i=1}^M$ is to consider an aggregate objective, such as the weighted sum of all individual objectives:

$$\mathbf{x}^* = \arg\max_{\mathbf{x} \in \mathcal{X}} \sum_i w_i f_i(\mathbf{x}) \tag{18}$$

where $w_i$ is the weight of the $i$-th objective, considered as a hyperparameter. However, determining appropriate weights for each objective function is a nontrivial task.

**Pareto optimal solution.** A more rigorous approach to multi-objective optimization involves introducing partial order and considering solutions from the Pareto frontier (Yang et al., 2019). In this context, partial order is defined by comparing all objectives $\{f_i(\cdot)\}_{i=1}^M$ for the given solutions. We say a solution $\mathbf{x}'$ surpasses $\mathbf{x}$ if every objective evaluated on $\mathbf{x}'$ is greater than or equal to the same objective evaluated on $\mathbf{x}$ (assuming maximization of objectives):

$$\mathbf{x}' \succeq \mathbf{x} \Leftrightarrow \forall i, f_i(\mathbf{x}') \geq f_i(\mathbf{x}) \tag{19}$$

For a given set of solutions $S = \{\mathbf{x}_j\}_{j=1}^m$, the *Pareto frontier* $P(S)$ is defined as the set of non-dominated solutions. For every solution $\mathbf{x} \in P(S)$, there is no other solution in $S$ surpassing $\mathbf{x}$:

$$P(S) = \{\mathbf{x} \in S : \{\mathbf{x}' \in S : \mathbf{x}' \succeq \mathbf{x}, \mathbf{x}' \neq \mathbf{x}\} = \emptyset\} \tag{20}$$

**Hypervolume.** Assuming bounded objectives, two Pareto frontiers can be compared by evaluating their hypervolume (Couckuyt et al., 2014):

$$\text{Volume}(P(S)) = \text{Volume} \left( \bigcup_{\mathbf{x} \in P(S)} H(\mathbf{x}) \right) \tag{21}$$

where $H(\mathbf{x})$ is the hyperrectangle associated with the objectives evaluated on solution $\mathbf{x}$, and $\text{Volume}(\cdot)$ computes the Euclidean volume of the input set. In our experiments, we use EHVI as the acquisition function and adopt the implementation outlined in (Daulton et al., 2020) for efficient computation of EHVI.

## C  PROOF OF RESULTS

### C.1  PROOF OF THEOREM 1

**Key lemmas.** In this section, we provide the detailed proof of Theorem 1. We first observe that we have confidence on all the decisions chosen according to the following lemma:

**Lemma 1** (Lemma 5.5 of (Srinivas et al., 2012)). *Pick $\delta \in (0, 1)$ and set $\beta_t = 2 \log(\pi_t/\delta)$, where $\sum_{t \geq 1} \pi_t^{-1} = 1, \pi_t > 0$. Then,*

$$|f(\mathbf{x}_t) - \mu_{t-1}(\mathbf{x}_t)| \leq \beta_t^{1/2} \sigma_{t-1}(\mathbf{x}_t), \ \ \forall t \geq 1,$$

*holds with probability $\geq 1 - \delta$.*

*Proof.* Fix $t \geq 1$ and $\mathbf{x} \in \mathcal{X}$. Conditioned on $\mathbf{y}_{t-1} = [y_1, y_2, \ldots, y_{t-1}]^\top$, $\mathbf{x}_1, \ldots, \mathbf{x}_{t-1}$ are deterministic and $f(\mathbf{x}) \sim \mathcal{N}(\mu_{t-1}(\mathbf{x}), \sigma_{t-1}^2(\mathbf{x}))$. If $r \sim \mathcal{N}(0, 1)$, then

$$\Pr\{r > c\} = e^{-c^2/2}(2\pi)^{-1/2} \int e^{-(r-c)^2/2 - c(r-c)} \leq e^{-c^2/2} \Pr\{r > 0\} = \frac{1}{2}e^{-c^2/2}.$$

Let $r = (f(\mathbf{x}) - \mu_{t-1}(\mathbf{x}))/\sigma_{t-1}(\mathbf{x})$, then $\Pr\{|f(\mathbf{x}) - \mu_{t-1}(\mathbf{x})| > \beta_t^{1/2} \sigma_{t-1}(\mathbf{x})\} \leq e^{-\beta_t/2}$. By using $e^{-\beta_t/2} = \delta/\pi_t$ and applying the union bound for $t \in \mathbb{N}$, the statement holds. $\square$

For the sake of analysis, we consider a set of discretizations $\mathcal{X}_t \subset \mathcal{X}$, where $\mathcal{X}_t$ will be used at iteration $t$ in the analysis. The following lemma provides a confidence bound for these subsets.

**Lemma 2** (Lemma 5.6 of (Srinivas et al., 2012)). *Pick $\delta \in (0, 1)$ and set $\beta_t = 2 \log(|\mathcal{X}_t|\pi_t/\delta)$, where $\sum_{t \geq 1} \pi_t^{-1} = 1, \pi_t > 0$. Then,*

$$|f(\mathbf{x}_t) - \mu_{t-1}(\mathbf{x}_t)| \leq \beta_t^{1/2} \sigma_{t-1}(\mathbf{x}_t), \ \ \forall \mathbf{x} \in \mathcal{X}_t, \forall t \geq 1,$$

*holds with probability $\geq 1 - \delta$.*

*Proof.* The proof is similar to that of Lemma 1, except now we use $\mathcal{X}_t$ at each iteration. $\square$

Now, by Assumption 1 and the union bound, we have

$$\Pr\left\{\left|\frac{\partial f}{\partial x_j}\right| < L\right\} \geq 1 - dae^{-L^2/b^2}, \ \ \forall \mathbf{x} \in \mathcal{X}, \forall j, \tag{22}$$

which implies that, with probability greater than $1 - dae^{-L^2/b^2}$,

$$|f(\mathbf{x}) - f(\mathbf{x}')| \leq L\|\mathbf{x} - \mathbf{x}'\|_1, \ \ \forall \mathbf{x} \in \mathcal{X}. \tag{23}$$

This allows us to obtain confidence on $\mathbf{x}^*$ as follows. Let us choose a discretization $\mathcal{X}_t$ of size $(\tau_t)^d$ such that for all $\mathbf{x} \in \mathcal{X}_t$,

$$\|\mathbf{x} - [\mathbf{x}]_t\|_1 \leq rd/\tau_t, \tag{24}$$

where $[\mathbf{x}]_t$ is the closest point in $\mathcal{X}_t$ to $\mathbf{x}$.

**Lemma 3** (Lemma 5.7 of (Srinivas et al., 2012)). *Pick $\delta \in (0, 1)$ and set $\beta_t = 2 \log(2\pi_t/\beta_t) + 4d \log(dtbr\sqrt{\log(2da/\delta)})$, where $\sum_{t \geq 1} \pi_t^{-1} = 1, \pi_t > 0$. Let $\tau_t = dt^2br\sqrt{\log(2da/\delta)}$, then*

$$|f(\mathbf{x}^*) - \mu_{t-1}([\mathbf{x}^*]_t)| \leq \beta_t^{1/2} \sigma_{t-1}([\mathbf{x}^*]_t) + \frac{1}{t^2}, \ \ \forall t \geq 1,$$

*holds with probability $\geq 1 - \delta$.*

*Proof.* Using Eq. (23), we have,

$$|f(\mathbf{x}) - f(\mathbf{x}')| \leq b\sqrt{\log(2da/\delta)}\|\mathbf{x} - \mathbf{x}'\|_1, \ \ \forall \mathbf{x} \in \mathcal{X}.$$

with probability greater than $1 - \delta/2$. It follows that,

$$|f(\mathbf{x}) - f([\mathbf{x}]_t)| \le rdb\sqrt{\log(2da/\delta)}/\tau_t, \quad \forall \mathbf{x} \in \mathcal{X}.$$

By choosing $\tau_t = dt^2 br\sqrt{\log(2da/\delta)}$, we have

$$|f(\mathbf{x}) - f([\mathbf{x}]_t)| \le \frac{1}{t^2},$$

which implies that $|\mathcal{X}_t| = (dt^2 br\sqrt{\log(2da/\delta)})^d$. Using $\delta/2$ in Lemma 2, we can apply the confidence bound to $[\mathbf{x}^*]_t$ to obtain the result. $\qquad\square$

Next, we would like to bound the regret using the following lemma:

**Lemma 4** (Lemma 5.8 of (Srinivas et al., 2012))**.** *Pick* $\delta \in (0,1)$ *and set* $\beta_t = 2\log(4\pi_t/\beta_t) + 4d\log(dtbr\sqrt{\log(4da/\delta)})$, *where* $\sum_{t \ge 1} \pi_t^{-1} = 1, \pi_t > 0$. *Then,*

$$r_t \le 2\beta_t^{1/2}\sigma_{t-1}(\mathbf{x}_t) + \frac{1}{t^2}, \quad \forall t \in \mathbb{N},$$

*holds with probability* $\ge 1 - \delta$.

*Proof.* We use $\delta/2$ in Lemma 1 and Lemma 2 such that the events hold with probability greater than $1 - \delta$. By using the definition of $\mathbf{x}_t$,

$$\mu_{t-1}(\mathbf{x}_t) + \beta_t^{1/2}\sigma_{t-1}(\mathbf{x}_t) \ge \mu_{t-1}([\mathbf{x}^*]_t) + \beta_t^{1/2}\sigma_{t-1}([\mathbf{x}^*]_t).$$

By Lemma 3, we further have $\mu_{t-1}([\mathbf{x}^*]_t) + \beta_t^{1/2}\sigma_{t-1}([\mathbf{x}^*]_t) + \frac{1}{t^2} \ge f(\mathbf{x}^*)$, which implies

$$\mu_{t-1}(\mathbf{x}_t) + \beta_t^{1/2}\sigma_{t-1}(\mathbf{x}_t) \ge f(\mathbf{x}^*) - \frac{1}{t^2}.$$

Therefore,

$$r_t = f(\mathbf{x}^*) - f(\mathbf{x}_t) \le \mu_{t-1}(\mathbf{x}_t) + \beta_t^{1/2}\sigma_{t-1}(\mathbf{x}_t) + \frac{1}{t^2} - f(\mathbf{x}_t) \le 2\beta_t^{1/2}\sigma_{t-1}(\mathbf{x}_t) + \frac{1}{t^2}.$$

$$\square$$

**Proof of Theorem 1.** Now, we are ready to complete the proof of Theorem 1. Since $\beta_t$ is non-decreasing,

$$4\beta_t\sigma_{t-1}^2(\mathbf{x}_t) \le 4\beta_T\sigma_\epsilon^2(\sigma_\epsilon^{-2}\sigma_{t-1}^2(\mathbf{x}_t)) \le 4\beta_T\sigma_\epsilon^2 C_2\log(1 + \sigma_\epsilon^{-2}\sigma_{t-1}^2(\mathbf{x}_t)),$$

with $C_2 = \sigma_\epsilon^2/\log(1 + \sigma_\epsilon^{-2}) \ge 1$, since $s^2 \le C_2\log(1 + s^2)$ for $s \in [0, \sigma_\epsilon^{-2}]$ and $\sigma_\epsilon^{-2}\sigma_{t-1}^2(\mathbf{x}_t) \le \sigma_\epsilon^{-2}k(\mathbf{x}_t, \mathbf{x}_t) \le \sigma_\epsilon^{-2}$. By Lemma 5.3 of (Srinivas et al., 2012), for a finite set of observed points $\mathcal{A} = \{\mathbf{x}_1, \dots, \mathbf{x}_T\} \subset \mathcal{X}$, the information gain can be expressed as,

$$\mathcal{I}(\mathbf{y}_\mathcal{A}; f) = \frac{1}{2}\sum_{t=1}^T \log\left(1 + \sigma^{-2}\sigma_{t-1}^2(\mathbf{x}_t)\right). \tag{25}$$

Thus, by substituting the above expression and setting $C_1 = 8\sigma_\epsilon^2 C_2$, we can obtain

$$\sum_{t=1}^T 4\beta_t\sigma_{t-1}^2(\mathbf{x}_t) \le C_1\beta_T\mathcal{I}(\mathbf{y}_\mathcal{A}; f) \le C_1\beta_T\gamma_T, \quad \forall T \ge 1, \tag{26}$$

with probability greater than $1 - \delta$. By Cauchy-Schwarz,

$$\sum_{t=1}^T 2\beta_t^{1/2}\sigma_{t-1}(\mathbf{x}_t) \le \sqrt{C_1 T \beta_T \gamma_T}, \quad \forall T \ge 1. \tag{27}$$

Finally, by choosing $\pi_t = \pi^2 t^2/6$ in Lemma 4, we have

$$R_T = \sum_{t=1}^T r_t \le \sum_{t=1}^T 2\beta_t^{1/2}\sigma_{t-1}(\mathbf{x}_t) + \sum_{t=1}^T \frac{1}{t^2} \le \sqrt{C_1 T \beta_T \gamma_T} + \frac{\pi^2}{6}, \quad \forall T \ge 1.$$

This completes the proof of Theorem 1.

### C.2 PROOF OF THEOREM 3

In this section, we provide the proof of our main result, which is summarized in Theorem 3. From Theorem 1, we have that with probability at least $1 - \delta$,

$$R_T \leq \sqrt{C_1 T \beta_T \gamma_T} + \frac{\pi^2}{6},$$

where $C_1 = 8/(\log(1 + \sigma_\epsilon^{-2}))$ and

$$\beta_T = 2 \log\left(\frac{T^2 2\pi^2}{3\delta}\right) + 4d \log\left(dtbr\sqrt{\log\left(\frac{4da}{\delta}\right)}\right).$$

Since $\beta_T$ involves logarithmic terms and constants, we can express it using the $\tilde{\mathcal{O}}$ notation as $\beta_T = \tilde{\mathcal{O}}(d)$. Similarly, $C_1$ is a constant that can be absorbed into the $\tilde{\mathcal{O}}$ notation. Therefore, the cumulative regret bound simplifies to

$$R_T \leq \tilde{\mathcal{O}}\left(\sqrt{Td\gamma_T}\right).$$

We now combine this result with the bounds on $\gamma_T$ from Theorem 2 for different kernels used in this work:

1. **Linear Kernel:** Substituting $\gamma_T = \mathcal{O}(d \log T)$ into the regret bound:
   $$R_T \leq \tilde{\mathcal{O}}\left(\sqrt{Td \cdot d \log T}\right) = \tilde{\mathcal{O}}\left(d\sqrt{T \log T}\right).$$

2. **Squared Exponential Kernel:** Substituting $\gamma_T = \mathcal{O}((\log T)^{d+1})$ into the regret bound:
   $$R_T \leq \tilde{\mathcal{O}}\left(\sqrt{Td(\log T)^{d+1}}\right) = \tilde{\mathcal{O}}\left(\sqrt{T(\log T)^{d+1}}\right).$$

3. **Matérn Kernel with $\nu > 1$:** Substituting $\gamma_T = \mathcal{O}\left(T^{\frac{d(d+1)}{2\nu+d(d+1)}} \log T\right)$ into the regret bound:
   $$R_T \leq \tilde{\mathcal{O}}\left(\sqrt{Td \cdot T^{\frac{d(d+1)}{2\nu+d(d+1)}} \log T}\right) = \tilde{\mathcal{O}}\left(T^{\frac{1}{2} + \frac{d(d+1)}{4\nu+2d(d+1)}}\sqrt{\log T}\right).$$

In all cases, the cumulative regret $R_T$ grows sub-linearly with $T$, ensuring that the regret per iteration diminishes as $T$ increases. This completes the proof for Theorem 3.

## D IMPLEMENTATION DETAILS

In this section, we outline the implementation details of our method and considered baselines.

### D.1 CAKES SETUP

**Choice of LLM.** We chose the `gpt-4o-mini` model as it offers an excellent balance of affordability, fast inference, and intelligence for our implementation. We access the model through the OpenAI API, which provides two key hyperparameters that allow us to control the randomness and diversity of the generated output: `temperature` and `top_p`. Concretely, `temperature` adjusts the "sharpness" of the probability distribution used to select the next token. Lower values result in more deterministic outputs, while higher values yield more random results. Conversely, `top_p` (or nucleus sampling) filters the probability distribution to consider only the most likely tokens that cumulatively reach a specified threshold, typically between 0 and 1. In our experiments, we found that setting `temperature=0.7` and `top_p=0.95` strikes a good balance between generating relevant outputs and introducing diversity. For further details, we refer the interested reader to the official documentation: `https://platform.openai.com/docs/overview`. In our experiments, we set the number of crossovers $n_c = 5$, probability of mutation $p_m = 0.7$, and population size $n_p = 10$.

**Prompt design.** We show the designed system prompt in Figure 4 as well as the sample prompts for the mutation and crossover steps in Figures 5b and 5a, respectively. Note that in all figures, {} is used to indicate placeholders.

---

**System Prompt**

You are an expert in machine learning, specializing in Gaussian processes. Here are the observations we have collected so far: {observations}

Please analyze these observations to identify patterns in the data that can be captured by a kernel function. You can use squared exponential (SE) kernel to capture smoothness, periodic (PER) kernel to capture periodicity, linear (LIN) kernel to capture linear trends, and rational quadratic (RQ) kernel to capture varying data patterns. You can also combine these kernels using the $+$ and $*$ operators to capture more complex patterns. For example, LIN $+$ PER can capture a linear trend with periodic fluctuations and LIN $*$ PER can capture a periodic pattern with linearly increasing amplitude.

---

Figure 4: The designed system prompt for the LLM.

---

**Crossover Prompt**

You are given two parent kernels and their Bayesian Information Criterion (BIC) values: {parent_kernel1} (BIC: {bic1}), {parent_kernel2} (BIC: {bic2}).

Please recommend a kernel that has a lower BIC value. You can either combine the parent kernels with $+$ or $*$ operator. Your output should follow the following format:

Kernel: <your proposed kernel here, only use the kernel abbreviations and operators>
Analysis: <your analysis here, explaining your reason behind the proposed kernel>

---

**Mutation Prompt**

You are given a kernel and its Bayesian Information Criterion (BIC) value: {kernel} (BIC: {bic}).

Please recommend a kernel that has a lower BIC value. You can only replace one of the base kernels in the kernel with another base kernel. Your output should follow the following format:

Kernel: <your proposed kernel here, only use the kernel abbreviations and operators>
Analysis: <your analysis here, explaining your reason behind the proposed kernel>

---

(a) Crossover prompt.          (b) Mutation prompt.

Figure 5: Sample prompts for proposing the kernels.

## D.2 BASELINES

To test our proposed method, we consider four categories of baselines: fixed kernel, adaptive kernel, deep GP, and ensemble GP.

**Fixed.** For the fixed kernel baseline, we consider the following widely-used kernels in BO:

1. **Squared exponential (SE):**

$$k(\mathbf{x}, \mathbf{x}') = \exp\left(-\frac{1}{2}(\mathbf{x} - \mathbf{x}')^\top \Theta^{-2}(\mathbf{x} - \mathbf{x}')\right), \tag{28}$$

  where $\Theta$ is the lengthscale parameter.

2. **Periodic (PER):**

$$k(\mathbf{x}, \mathbf{x}') = \exp\left(-2\sum_i \frac{\sin^2\left(\pi(x_i - x_i')/p\right)}{\lambda}\right), \tag{29}$$

  where $p$ is the period length parameter and $\lambda$ is the lengthscale parameter.

3. **Linear (LIN):**

$$k(\mathbf{x}, \mathbf{x}') = v\mathbf{x}^\top \mathbf{x}', \tag{30}$$

  where $v$ is the variance parameter.

4. **Rational quadratic (RQ):**

$$k(\mathbf{x}, \mathbf{x}') = \left(1 + \frac{1}{2\alpha}(\mathbf{x} - \mathbf{x}')^\top \Theta^{-2}(\mathbf{x} - \mathbf{x}')\right)^{-\alpha}, \tag{31}$$

where $\Theta$ is the lengthscale parameter and $\alpha$ is the relative weighting parameter.

5. **Matérn:**

$$k(\mathbf{x}, \mathbf{x}') = \frac{2^{1-\nu}}{\Gamma(\nu)} \left(\sqrt{2\nu}D\right)^\nu K_\nu \left(\sqrt{2\nu}D\right), \tag{32}$$

where

$$D = (\mathbf{x} - \mathbf{x}')^\top \Theta^{-2}(\mathbf{x} - \mathbf{x}') \tag{33}$$

is the distance between $\mathbf{x}$ and $\mathbf{x}'$ scaled by the lengthscale parameter and $K_\nu$ is the modified Bessel function. In our experiments, the smoothness parameter $\nu$ is set to 1/2, 3/2, or 5/2, corresponding to Matérn-1/2 (M1), Matérn-3/2 (M3), or Matérn-5/2 (M5) kernels respectively.

**Adaptive.** For the adaptive kernel baseline, we adopt the implementation from Roman et al. (2019) and apply the following selection criteria:

1. **Random:** This criterion selects a kernel randomly from the set of available kernels.

2. **Utility:** Based on the proposed query points from each kernel, this criterion selects the kernel with the highest utility (acquisition) value:

$$k^* = \arg\max_{k \in \mathbb{K}} \alpha(\mathbf{x}_{t,k}; \mathcal{D}, k). \tag{34}$$

3. **BIC:** This criterion selects the kernel with the lowest BIC value:

$$k^* = \arg\min_{k \in \mathbb{K}} \mathrm{BIC}(k; \mathcal{D}). \tag{35}$$

**Deep GP.** For the deep GP baseline, we use the `DeepGP` implementation from GPyTorch, where training and inference are conducted using the doubly stochastic variational inference method (Salimbeni & Deisenroth, 2017).

**Ensemble GP.** For the ensemble GP baseline, we follow the implementation suggested by Lu et al. (2023), using the six base kernels used in CAKES to form the kernel dictionary.

# E  EXPERIMENTAL DETAILS

In this section, we provide additional details on the benchmarks employed in our experiments.

## E.1  OPTIMIZATION OF BENCHMARK FUNCTIONS

**Test functions.** We provide additional details on test functions, including the function domain and dimensionality, in Table 2. The analytic expression as well as the global optimum of these functions can be found at `https://www.sfu.ca/~ssurjano/optimization.html`. We visualize the optimization landscapes of the two-dimensional test functions in Figure 6. From the figure, one can see the challenging nature of these functions, which is characterized by many local minima, multi-modality, and steep ridges.

## E.2  HYPERPARAMETER TUNING TASK

**Datasets.** We include 12 OpenML datasets available in the HPOBench package (Eggensperger et al., 2021). The details of the selected datasets are given in Table 3.

**Search space.** We follow the search space designated in HPOBench, where we discretize the search space to facilitate efficient tabular lookup operations for various configurations (Eggensperger et al., 2013). The search space for each ML model is summarized below, i.e., {type: [space, lower_bound, upper_bound]}:

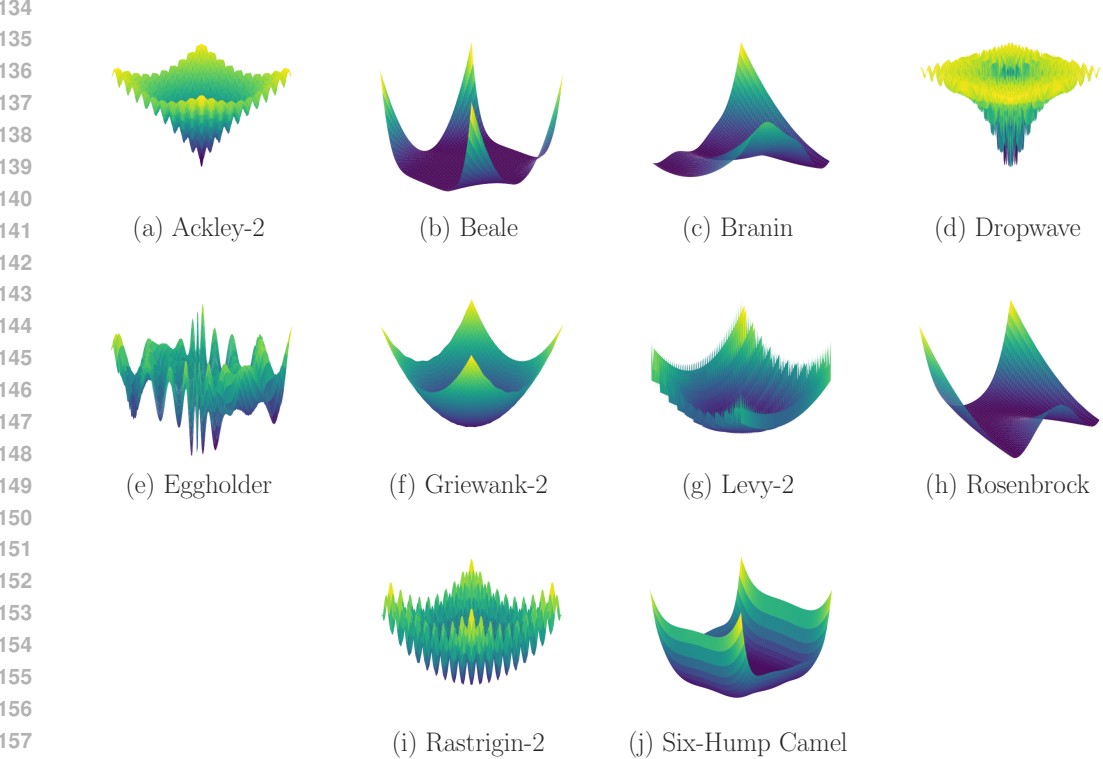

(a) Ackley-2      (b) Beale      (c) Branin      (d) Dropwave

(e) Eggholder      (f) Griewank-2      (g) Levy-2      (h) Rosenbrock

(i) Rastrigin-2      (j) Six-Hump Camel

Figure 6: Visualization of the optimization landscapes of two-dimensional test functions.

Table 2: Details of the test functions used in the experiments.

| Function | Domain | $d$ |
|---|---|---|
| Ackley-$d$ | $[-5, 5]^d$ | 2, 5 |
| Beale | $[-1, 1]^2$ | 2 |
| Branin | $[-5, 10]^2$ | 2 |
| Dropwave | $[-5.12, 5.12]^2$ | 2 |
| Eggholder | $[-512, 512]^2$ | 2 |
| Griewank-d | $[-600, 600]^d$ | 2, 5 |
| Hartmann | $[0, 1]^3$ | 3 |
| Levy | $[-10, 10]^d$ | 2, 3 |
| Rastringin-$d$ | $[-5.12, 5.12]^d$ | 2, 4 |
| Rosenbrock | $[-5, 10]^2$ | 2 |
| Six-Hump Camel | $[-3, 3] \times [-2, 2]$ | 2 |

- **Logistic Regression** ($d = 2$): {`alpha`: [log, 0.001, 1.0], `eta0`: [log, 0.001, 1.0]}

- **Support Vector Machine** ($d = 2$): {`C`: [log, 0.01, 10.0], `gamma`: [log, 0.001, 1.0]}

- **Random Forest** ($d = 4$): {`max_depth`: [log, 1, 50], `max_features`: [linear, 0.0, 1.0], `min_samples_leaf`: [linear, 1, 2], `min_samples_split`: [log, 2, 128]}

- **XGBoost** ($d = 4$): {`colsample_bytree`: [linear, 0.1, 1.0], `eta`: [log, 0.001, 1.0], `max_depth`: [log, 1, 50], `reg_lambda`: [log, 0.1, 10.0]}

- **Multi-Layer Perceptron** ($d = 5$): {`alpha`: [log, 0.001, 1.0], `batch_size`: [log, 16, 128], `depth`: [linear, 1, 3], `learning_rate_init`: [log, 0.001, 1.0]}, `width`: [log, 16, 128]}

Table 3: Details of the OpenML datasets used in the experiments. More information can be found at https://www.openml.org.

| Dataset | Task ID | Number of Instances | Number of Features |
|---|---|---|---|
| credit_g | 31 | 1000 | 21 |
| vehicle | 53 | 846 | 19 |
| kc1 | 2109 | 2109 | 22 |
| phoneme | 9952 | 5404 | 6 |
| blood_transfusion | 10101 | 748 | 5 |
| australian | 146818 | 690 | 15 |
| car | 146821 | 1728 | 7 |
| segment | 146822 | 2310 | 20 |
| heart_h | 50 | 294 | 14 |
| tic_tac_toe | 145804 | 958 | 10 |
| kr_vs_kp | 3 | 3196 | 37 |
| qsar | 9957 | 1055 | 42 |

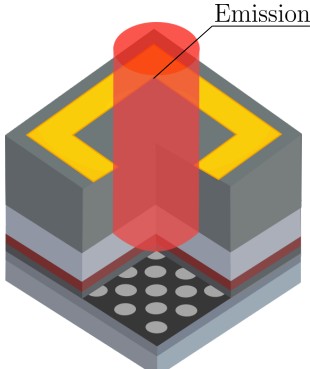

**Design Parameters**

| | |
|---|---|
| Length | Lattice constant |
| Width | Size of holes |
| Layer thickness | Filling factor |
| Shape of holes | Material properties |

| Indicator | Target |
|---|---|
| Q-factor (Q) | $Q^* = 2 \times 10^6$ |
| Wavelength ($\lambda$) | $\lambda^* = 1310$ |
| Lasing area (A) | $A^* = 3.6 \times 10^{-13}$ |
| Power ($P$) | $P^* = 0.5$ |
| Divergence angle ($\omega$) | $\omega^* = 1.0$ |

Figure 7: Illustration of a photonic chip and its design parameters. The inverse design problem focuses on optimizing these parameters to satisfy certain performance indicators.

### E.3 PHOTONIC CHIP DESIGN

**Objective.** The design parameters and indicators for optimizing the photonic chip are detailed in Figure 7. We established target values based on our experiments and a review of the literature to identify optimal standards for high-quality chip lasing (Li et al., 2023). For instance, a wavelength of 1310 nm is crucial for telecommunications and satellite applications, while high output power is essential for fields such as autonomous driving and medicine. Additionally, a small divergence angle is vital for ensuring high beam quality and effective long-distance light propagation. Based on these performance indicators, we can define the objectives as follows,

$$f_1 = 1 - \frac{Q^* - Q}{Q^*}, \tag{36}$$

$$f_2 = 1 - \frac{|\lambda^* - \lambda|}{\lambda^*}, \tag{37}$$

$$f_3 = 1 - \frac{A^* - A}{A^*}, \tag{38}$$

$$f_4 = 1 - \frac{P^* - P}{P^*}, \tag{39}$$

$$f_5 = 1 + \frac{\omega^* - \omega}{\omega^*}. \tag{40}$$

Q-factor ($f_1$) is related to the loss and threshold of the laser, wavelength ($f_2$) is the operation wavelength of the laser, lasing area ($f_3$) is the area of the laser beam at the laser's top surface, power

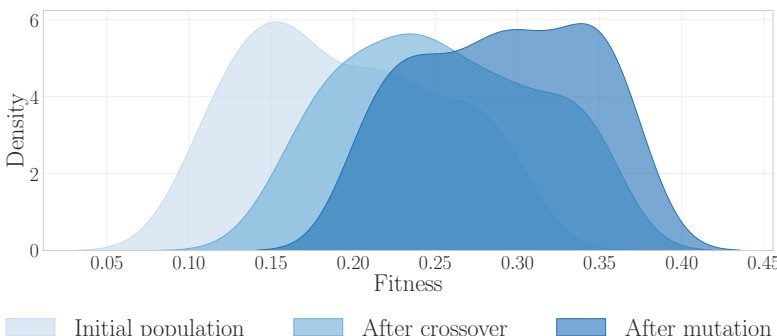

Figure 8: Comparison of the fitness distributions between an initial population of kernels before and after applying crossover and mutation operations.

($f_4$) is the lasing power of the laser in watts, and divergence angle ($f_5$) is the angle between outer boundary and centerline of the laser beam.

## F    EMPIRICAL ANALYSIS

In this section, we provide additional empirical analysis on our proposed method in Section 3.

### F.1    CROSSOVER AND MUTATION WITH LLMs

To demonstrate the benefits of using LLMs for kernel generation, we conduct an experiment to analyze the fitness distributions of an initial population of kernels before and after applying crossover and mutation operations. Specifically, we consider the hyper-parameter tuning tasks in Section 6.2 and evaluate the fitness values of the initial population. We then use the LLM to perform crossover and mutation on this population, and re-evaluate the fitness values. Figure 8 shows the comparison of fitness distributions across these three different stages. From the figure, we can observe that the initial population exhibits a wide distribution, with a noticeable peak at lower fitness values. After performing crossover, the distribution shifts slightly toward higher fitness values, indicating that the LLM can effectively combine the kernels and enhance the overall fitness of the population. Furthermore, we can see that applying mutation shifts the distribution toward even higher fitness values, suggesting that further refining the kernels can yield higher fitness values. Overall, these findings demonstrate that LLM can act as a good crossover and mutation operators.

### F.2    CASE STUDY

We conduct a case study to analyze the output from the LLM and test whether or not the LLM can give helpful responses. Specifically, we show sample responses from the LLM for the crossover and mutation steps, which are depicted in Figure 9. In Figure 9a, given the parent kernels LIN and SE along with their BIC values, the LLM suggests combining them using the addition operator (+) to minimize the BIC. The analysis explains the rationale behind this combination: the LIN kernel captures linear trends, while the SE kernel models smoothness. By combining these properties, the LIN + SE kernel effectively represents both linearity and smoothness in the data. In Figure 9b, given the LIN + SE kernel and its BIC, the LLM recommends replacing the SE kernel with the RQ kernel to achieve a lower BIC value. The analysis indicates that while the SE kernel assumes fixed smoothness, the RQ kernel introduces varying smoothness, allowing the model to adapt better to data with fluctuating noise levels. These examples illustrate that the LLM is familiar with kernel properties and their applications. Its responses demonstrate a strong understanding of the relationships between different kernels and their capacity to address specific data characteristics, enabling it to make informed suggestions for the kernel selection.

---

**Crossover Response**

**Prompt:**
You are given two parent kernels and their Bayesian Information Criterion (BIC) values: LIN (BIC: 93.42),
SE (BIC: 118.59).

Please recommend a kernel that has a lower BIC value. You can either combine the parent kernels with $+$ or $*$ operator. Your output should follow the following format:

Kernel: <your proposed kernel here, only use the kernel abbreviations and operators>
Analysis: <your analysis here, explaining your reason behind the proposed kernel>

- - - - - - - - - - - - - - - - - - - - - - - - - - - - - - - - - - - - - - - - - - - - - - - - - - - - - - - - - -

**Response:**
Kernel: LIN $+$ SE
Analysis: The LIN kernel captures linear trends in the data, which may be present given the varying function values across different inputs. The SE kernel, on the other hand, is effective in modeling smooth variations in the data, which could account for the fluctuations observed in the function values despite the overall linear trend. By combining these two kernels with the addition operator ($+$), we can leverage the linear trend from the LIN kernel while also allowing for smooth deviations from this trend through the SE kernel.

(a) Sample response for the crossover step.

---

**Mutation Response**

**Prompt:**
You are given a kernel and its Bayesian Information Criterion (BIC) value:
LIN + SE (BIC: 79.83).

Please recommend a kernel that has a lower BIC value. You can only replace one of the base kernels in the kernel with another base kernel. Your output should follow the following format:

Kernel: <your proposed kernel here, only use the kernel abbreviations and operators>
Analysis: <your analysis here, explaining your reason behind the proposed kernel>

- - - - - - - - - - - - - - - - - - - - - - - - - - - - - - - - - - - - - - - - - - - - - - - - - - - - - - - - - -

**Response:**
Kernel: LIN $+$ RQ
Analysis: The LIN $+$ SE kernel captures a linear trend with smooth variations. However, the SE kernel assumes a constant level of smoothness, which may not be appropriate for all data patterns. By replacing the SE kernel with the RQ kernel, we can introduce varying smoothness, allowing the model to adapt to different levels of noise in the data.

(b) Sample response for the mutation step.

Figure 9: Sample responses from the LLM for the kernel generation process.

## F.3 ABLATION STUDY

Our prompts are designed based on the following components:

- **Domain information:** This include the role assignment as well as description of kernel functions and their properties.

- **Problem context:** This includes the optimization history, which consists of the observations as well as the fitness values.

- **Task instruction:** This includes the guidelines on how to generate the kernels based on the kernel grammar.

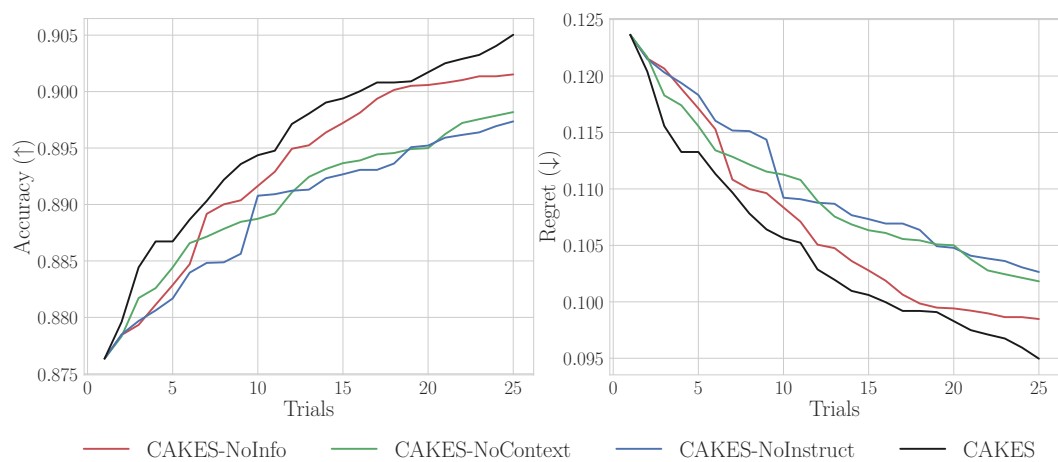

Figure 10: Ablation of prompt designs on HPOBench averaged over 20 different random seeds.

To evaluate the influence of each component on the performance, we conduct an ablation study with the following configurations:

- **CAKES:** This is the vanilla CAKES setup employed in our experiments, incorporating all three components.

- **CAKES-NoInfo:** This variant assesses the impact of domain-specific information on performance. Specifically, it omits the role assignment and kernel descriptions while keeping the other components (see Figure 15).

- **CAKES-NoContext:** This variant evaluates the importance of problem context by removing the optimization history (i.e., the observations and fitness values), while keeping the other components (see Figure 16).

- **CAKES-NoInstruct:** This variant excludes task instructions and omits the explicit kernel generation guidelines, while keeping the other components (see Figure 17).

We evaluate the performance of our ablation setups on HPOBench, as outlined in Section 6.2. The results, shown in Figure 10, reveal that the standard CAKES setup outperforms the other variants, highlighting the significance of each prompt component in enhancing overall performance. Notably, the CAKES-NoInfo variant demonstrated similar optimization performance to CAKES despite lacking domain-specific information. This suggests that the LLM may already possess valuable prior knowledge, even without role assignments and kernel descriptions in the prompt. In contrast, CAKES-NoContext exhibited worse performance than CAKES, indicating that optimization history is crucial and that the LLM effectively leverages this information. Furthermore, the results show that CAKES-NoInstruct performed poorly compared to the other variants. To investigate this, we examined the percentage of valid kernels, defined as the proportion of kernels that satisfy the necessary properties. We found that CAKES-NoInstruct generated valid kernels only $68.72\%(\pm0.45\%)$ of the time, while the other variants consistently produced valid kernels. This limitation can be attributed to the absence of generation guidelines from the kernel grammar. These findings underscore the importance of kernel grammar in the kernel generation process.

## F.4 CHOICE OF LLM

In this section, we analyze the impact of the LLM choice on the performance of our method. Specifically, we consider the following models: Meta's `llama-3.1-8b`, Google's `gemini-1.5-flash`, Anthrophic's `claude-3.5-haiku`, and OpenAI's `gpt-4o-mini`. We evaluate these models on the benchmark functions considered in Section 6.1. Based on the results presented in Table 4, we can observe that `gpt-4o-mini` outperforms the other models in 8 of 15 test functions. Claude-3.5-haiku also demonstrates competitive performance, securing the best

Table 4: Average normalized regret (↓) over 20 random seeds for different LLMs. The best value is highlighted in **bold** and the second best value is underlined.

| Function | llama-3.1-8b | gemini-1.5-flash | claude-3.5-haiku | gpt-4o-mini |
|---|---|---|---|---|
| Ackley-2 | 0.1019 | 0.0916 | 0.0812 | **0.0783** |
| Ackley-5 | 0.2106 | 0.1907 | **0.1621** | 0.1732 |
| Beale | 0.2621 | 0.2595 | 0.2640 | **0.2565** |
| Branin | 0.0131 | 0.0199 | 0.0091 | **0.0070** |
| Dropwave | 0.4941 | **0.4640** | 0.4878 | 0.4690 |
| Eggholder | 0.1323 | 0.1309 | **0.1156** | 0.1241 |
| Griewank-2 | 0.0337 | 0.0360 | 0.0289 | **0.0267** |
| Griewank-5 | 0.0301 | 0.0247 | **0.0158** | 0.0185 |
| Hartmann | 0.0012 | 0.0005 | **0.0001** | **0.0001** |
| Levy-2 | 0.0459 | 0.0426 | 0.0368 | **0.0353** |
| Levy-3 | 0.0710 | 0.0667 | **0.0480** | 0.0505 |
| Rastringin-2 | 0.3595 | 0.3611 | **0.3322** | 0.3341 |
| Rastringin-4 | 0.3301 | 0.3147 | 0.3253 | **0.3128** |
| Rosenbrock | 0.0623 | 0.0501 | 0.0507 | **0.0483** |
| Six-Hump Camel | 0.1103 | **0.1002** | 0.1073 | 0.1015 |

results in 6 out of 15 test functions. Moreover, as of November 2024, `gpt-4o-mini` is priced at \$0.15 per million input tokens and \$0.60 per million output tokens, making it significantly more affordable than `claude-3.5-haiku`, which costs \$1 per million input tokens and \$5 per million output tokens. In terms of speed, `gpt-4o-mini` delivers an output speed of approximately 104.5 tokens per second, outperforming `claude-3.5-haiku`'s 64.3 tokens per second. Considering the above metrics, we choose `gpt-4o-mini` as the default model in CAKES as it offers an excellent balance of cost, speed, and intelligence for our implementation.

# G  ADDITIONAL OPTIMIZATION RESULTS

In this section, we show additional experimental results on the optimization tasks considered in Section 6.

## G.1  OPTIMIZATION OF BENCHMARK FUNCTIONS

**Additional comparisons.** We provide more comprehensive comparisons between our proposed method and fixed kernel approaches in Table 5. Specifically, we include three additional kernels: rational quadratic (RQ), Matérn-1/2 (M1), and Matérn-3/2 (M3). Overall, CAKES outperforms all fixed kernel approaches in terms of the mean and median regret. Our results indicate that fixed kernels do not perform uniformly well across all tasks, which is consistent with our findings in Section 6.1.

**Additional baselines.** We compare our proposed method against the following baselines:

- **Bag-of-models (BOM):** This is a method proposed by Kandasamy et al. (2015) which maintains a collection of kernels, each capturing different structures, that best explains the observed data.

- **Markov chain Monte Carlo (MCMC):** This is a method proposed by Gardner et al. (2017) which uses a Metropolis-Hastings algorithm to randomly sample kernels from the grammar based on a proposal distribution.

- **Automated BO (ABO)[1]:** This is a method proposed by Malkomes & Garnett (2018), which uses BO to dynamically select kernels that best explains the observed data.

For the above three baselines, we follow the setup suggested by the authors. We limited the number of models to 50 and use the L-BFGS algorithm to optimize each model's hyperparameters. As in

---

[1]We use the code provided by the authors in `https://github.com/gustavomalkomes/abo`.

Table 5: Average normalized regret ($\downarrow$) over 20 random seeds for different test functions and methods. The best value is highlighted in **bold** and the second best value is underlined.

| Function | Fixed | | | | | CAKES |
|---|---|---|---|---|---|---|
| | SE | RQ | M1 | M3 | M5 | |
| Ackley-2 | 0.1773 | 0.1605 | 0.1309 | 0.0878 | 0.1220 | **0.0783** |
| Ackley-5 | 0.3185 | 0.2732 | 0.4949 | 0.2197 | 0.2369 | **0.1732** |
| Beale | 0.3554 | 0.2881 | **0.2455** | 0.2945 | 0.3522 | 0.2565 |
| Branin | 0.0183 | **0.0055** | 0.0323 | 0.0455 | 0.0117 | 0.0070 |
| Dropwave | 0.5110 | 0.5185 | 0.6669 | 0.5848 | 0.5411 | **0.4690** |
| Eggholder | 0.4941 | 0.5111 | 0.4195 | 0.4297 | 0.3545 | **0.1241** |
| Griewank-2 | 0.1196 | 0.1365 | 0.0687 | 0.1362 | 0.1282 | **0.0267** |
| Griewank-5 | 0.0204 | 0.0210 | 0.0419 | 0.0200 | 0.0223 | **0.0185** |
| Hartmann | 0.0007 | 0.0003 | 0.0915 | 0.0131 | 0.0019 | **0.0001** |
| Levy-2 | 0.1562 | 0.0758 | 0.1190 | 0.0595 | 0.0418 | **0.0353** |
| Levy-3 | 0.1141 | 0.1630 | 0.0873 | 0.1226 | 0.1422 | **0.0505** |
| Rastringin-2 | 0.4325 | 0.4417 | 0.3573 | 0.4624 | 0.4251 | **0.3341** |
| Rastringin-4 | 0.5765 | 0.5752 | 0.4009 | 0.4290 | 0.5461 | **0.3128** |
| Rosenbrock | 0.1030 | 0.1054 | 0.3375 | 0.2460 | 0.0898 | **0.0483** |
| Six-Hump Camel | 0.2840 | 0.3105 | 0.4410 | 0.1431 | 0.1507 | **0.1015** |
| Mean regret | 0.2454 | 0.2391 | 0.2623 | 0.2196 | 0.2111 | **0.1358** |
| Median regret | 0.1773 | 0.1630 | 0.2455 | 0.1431 | 0.1422 | **0.0783** |

Table 6: Average normalized regret ($\downarrow$) over 20 random seeds for different test functions and methods. The best value is highlighted in **bold** and the second best value is underlined.

| Function | BOM | MCMC | ABO | CAKES |
|---|---|---|---|---|
| Ackley-2 | 0.1016 | 0.1012 | 0.1020 | **0.0783** |
| Ackley-5 | 0.5907 | **0.1312** | 0.1910 | 0.1732 |
| Beale | 0.3559 | 0.4040 | 0.3118 | **0.2565** |
| Branin | 0.0499 | 0.0301 | 0.0101 | **0.0070** |
| Dropwave | 0.5040 | 0.5788 | 0.5529 | **0.4690** |
| Eggholder | 0.4709 | 0.4536 | 0.4210 | **0.1241** |
| Griewank-2 | 0.0460 | 0.0589 | 0.0357 | **0.0267** |
| Griewank-5 | 0.2437 | 0.0258 | 0.0320 | **0.0185** |
| Hartmann | 0.0299 | **0.0001** | **0.0001** | **0.0001** |
| Levy-2 | 0.0626 | 0.0668 | 0.0519 | **0.0353** |
| Levy-3 | 0.0867 | 0.0580 | **0.0290** | 0.0505 |
| Rastringin-2 | 0.3618 | 0.3722 | 0.3420 | **0.3341** |
| Rastringin-4 | 0.3177 | 0.3285 | 0.3179 | **0.3128** |
| Rosenbrock | 0.0901 | 0.0907 | 0.0901 | **0.0483** |
| Six-Hump Camel | 0.1102 | 0.1071 | **0.1002** | 0.1015 |
| Mean regret | 0.2281 | 0.1871 | 0.1725 | **0.1358** |
| Median regret | 0.1102 | 0.1012 | 0.1002 | **0.0783** |

Section 6.1, we set $T = 10 \times d$ and repeat each experiment using 20 different random seeds. The results in 6 shows that CAKES is superior compared to all the baselines in terms of the mean and median regret. Notably, CAKES ranked first in 12 out of 15 instances and ranked a top-two ranking in all instances.

**Additional test functions.** To further evaluate our method in higher-dimensional settings, we conducted additional experiments using the Ackley and Levy functions across various input dimensions $d$. Specifically, we tested with $d \in \{10, 20, 30\}$. As illustrated in Figure 6, both of these functions are highly non-convex and possess numerous local minima, making them particularly challenging to optimize. We set $T = 10 \times d$ and repeat each experiment using 20 different random seeds. The

Table 7: Average normalized regret ($\downarrow$) over 20 random seeds for Ackley and Levy functions with varying dimensions. The best value is highlighted in **bold** and the second best value is underlined.

| Function | Fixed | | Adaptive | | | DGP | EGP | CAKES |
|---|---|---|---|---|---|---|---|---|
| | SE | M5 | Random | Utility | BIC | | | |
| Ackley-2 | 0.1773 | 0.1220 | 0.1358 | 0.1062 | 0.1863 | 0.2510 | 0.1878 | **0.0783** |
| Ackley-5 | 0.3185 | 0.2369 | **0.1722** | 0.2278 | 0.3590 | 0.3110 | 0.2285 | 0.1732 |
| Ackley-10 | 0.3645 | 0.3225 | 0.2725 | 0.4122 | 0.3157 | 0.4557 | 0.3108 | **0.2356** |
| Ackley-20 | 0.4523 | 0.4152 | 0.5103 | 0.5509 | 0.5021 | 0.5043 | 0.4237 | **0.3281** |
| Ackley-30 | 0.5331 | 0.5114 | 0.6460 | 0.5966 | 0.5664 | 0.6056 | 0.6210 | **0.4596** |
| Levy-2 | 0.1562 | 0.0418 | 0.0835 | **0.0255** | 0.1145 | 0.1965 | 0.0765 | 0.0353 |
| Levy-3 | 0.1141 | 0.1422 | 0.1495 | 0.0880 | 0.1125 | 0.2265 | 0.0805 | **0.0505** |
| Levy-10 | 0.3215 | 0.2514 | 0.2310 | 0.1545 | 0.1940 | 0.2845 | 0.1069 | **0.0941** |
| Levy-20 | 0.3657 | 0.4061 | 0.3815 | 0.1905 | 0.2762 | 0.3450 | 0.1703 | **0.1208** |
| Levy-30 | 0.4205 | 0.4598 | 0.4515 | 0.2405 | 0.3147 | 0.4534 | 0.2040 | **0.1483** |

Table 8: Average rank ($\downarrow$) $\pm$ standard deviation achieved by different methods over 5 random seeds.

| Model | | LR | SVM | RF | XGB | MLP |
|---|---|---|---|---|---|---|
| Fixed | SE | $3.33 \pm 2.09$ | $4.58 \pm 2.02$ | $4.00 \pm 2.35$ | $3.42 \pm 1.98$ | $3.42 \pm 1.93$ |
| | M5 | $4.25 \pm 1.30$ | $4.50 \pm 2.63$ | $4.08 \pm 1.93$ | $4.92 \pm 1.98$ | $5.58 \pm 1.75$ |
| Adaptive | Random | $5.17 \pm 1.40$ | $4.42 \pm 1.71$ | $5.08 \pm 2.36$ | $5.83 \pm 1.57$ | $6.08 \pm 1.66$ |
| | Utility | $6.17 \pm 1.77$ | $4.08 \pm 2.14$ | $3.92 \pm 1.85$ | $5.00 \pm 2.38$ | $4.42 \pm 2.50$ |
| | BIC | $7.33 \pm 1.18$ | $5.67 \pm 2.01$ | $5.25 \pm 2.05$ | $3.83 \pm 2.03$ | $4.83 \pm 2.70$ |
| DGP | | $3.33 \pm 2.36$ | $5.92 \pm 2.06$ | $5.08 \pm 2.63$ | $5.17 \pm 2.51$ | $4.08 \pm 2.18$ |
| EGP | | $4.17 \pm 1.40$ | $4.33 \pm 2.32$ | $5.17 \pm 2.73$ | $5.00 \pm 2.12$ | $4.92 \pm 1.80$ |
| CAKES | | $2.25 \pm 1.64$ | $2.50 \pm 1.50$ | $3.42 \pm 1.26$ | $2.83 \pm 1.95$ | $2.67 \pm 1.49$ |

results, presented in Table 7, demonstrate that CAKES consistently outperforms the other methods, even as $d$ increases. This suggests that our method can scale effectively in more complex search spaces. In addition, the results reveal that fixed kernel methods (SE and M5) perform poorly compared to the adaptive and multiple kernel approaches.

G.2 HYPERPARAMETER TUNING TASK

**Individual task results.** We present the individual task results from HPOBench for CAKES and other methods in Figures 18-27. Our findings reveal that there are significant variations in performance for fixed and adaptive kernel baselines, which align with the results presented in Section 6.2. Table 8 further highlights the superior performance of CAKES, which achieves the best average rank at the end of each tuning task.

**Tuning accuracy.** Figure 11 depicts the average accuracy in tuning different ML models over five random seeds. We further show the accuracy achieved by different methods on individual tasks in Figures 23-27 Overall, CAKES consistently achieves the highest accuracy in tuning all machine learning models compared to the other methods.

G.3 PHOTONIC CHIP DESIGN

**Individual objective results.** We plot the individual performance on each objective in Figure 12. The results show that the CAKES algorithm outperforms the other baseline approaches across multiple performance indicators. Particularly for objectives $f_3$ and $f_4$, CAKES achieves higher objective values more quickly compared to the baselines. This is particularly desirable as the third and fourth objectives are much more important than the other ones, especially from an practical engineering (e.g., autonomous driving or military laser weapon) point of view. This is because we want the

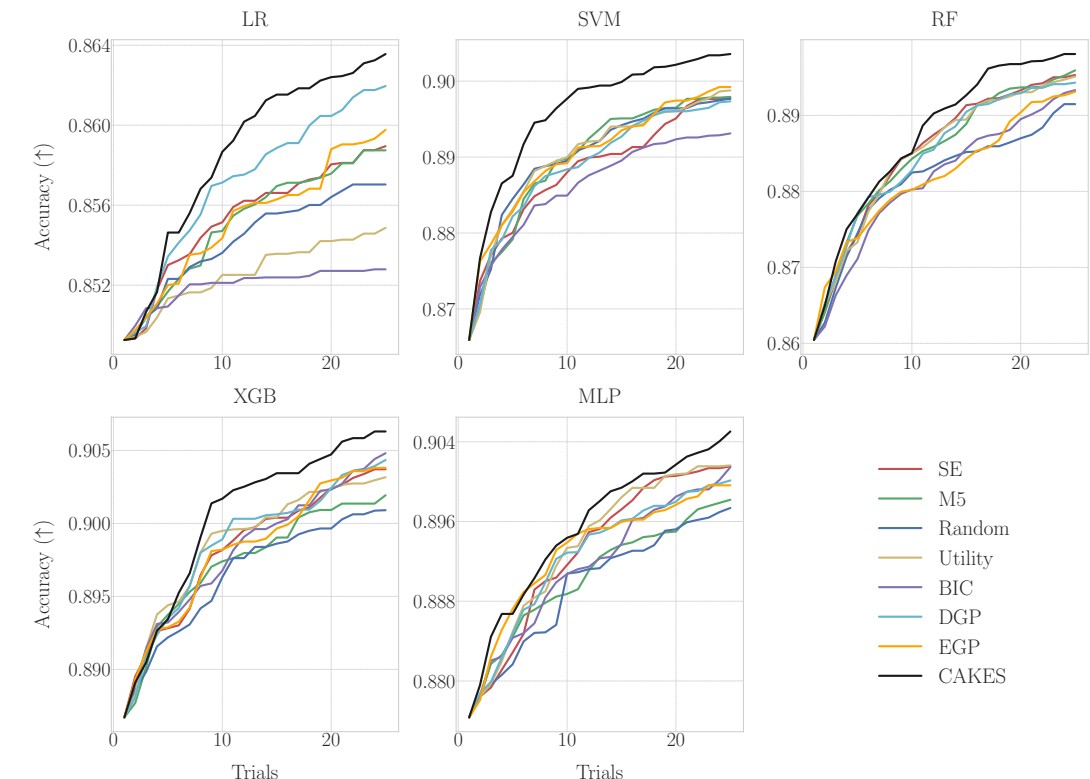

Figure 11: Average accuracy (↑) over 5 random seeds for different ML models on all datasets.

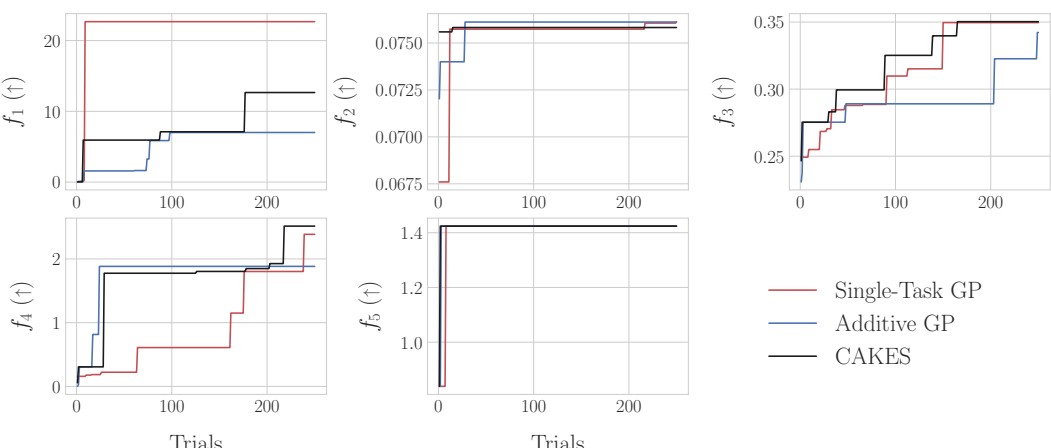

Figure 12: Comparison of individual objective results for photonic chip design across 250 trials.

photonic chip to have large lasing area ($f_3$) to produce a large light field coverage and have higher power ($f_4$) to transmit signal/beam faster and farther. Besides, CAKES also maintains a competitive performance for the other objectives throughout the trials. These results demonstrate that CAKES is not only effective in the early stages of the optimization process, but also capable of achieving long-term performance improvements.

**Additional baselines.** To further validate our results, we compare the optimized performance indicators obtained by our method with some additional baselines in the literature. In Table 9, we compare the five objectives ($f_1$ through $f_5$) against three state-of-the-art photonic chips of the same category from existing literature. Notably, our work is the only one to report power ($f_4$), which are crucial for LiDAR and military applications. In addition, we achieved the highest Q-factor, indicat-

Table 9: Comparison of optimized performance indicators obtained by our method and various baselines from the literature. NA denotes unreported values.

| Method | Q-factor | Wavelength | Lasing area | Power | Divergence |
|---|---|---|---|---|---|
| Ohnishi et al. (2004) | NA | 959.4 | $2.8 \times 10^{-9}$ | NA | 1.1 |
| Hsu et al. (2017) | $5.0 \times 10^3$ | 1299 | $1.8 \times 10^{-8}$ | NA | 2.0 |
| Chen et al. (2021) | $2.9 \times 10^3$ | 948 | $6.2 \times 10^{-9}$ | NA | 0.75 |
| Ours | $2.5 \times 10^7$ | 1309.3 | $1.3 \times 10^{-13}$ | 0.99 | 1.2 |

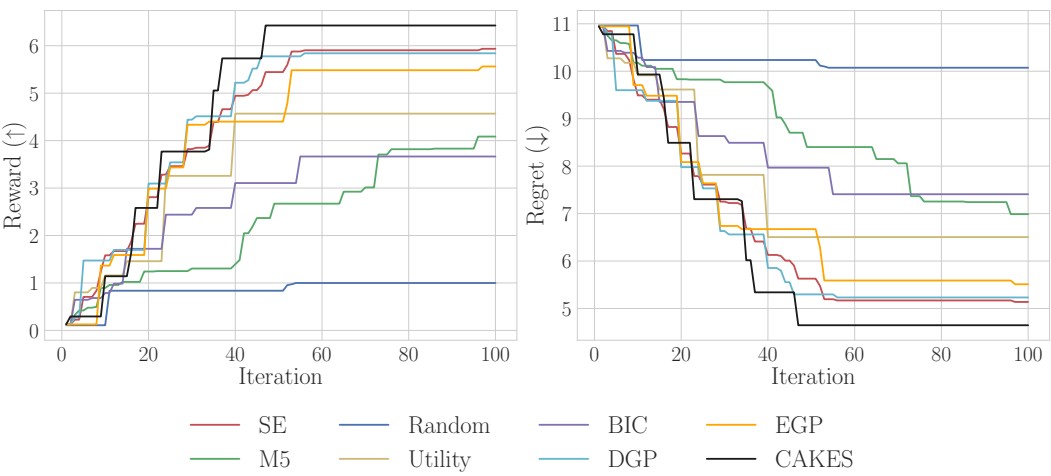

Figure 13: Reward ($\uparrow$) and regret ($\downarrow$) for the robot pushing problem.

ing high efficiency and a low threshold. Our lasing area is smaller than the literature benchmarks due to our chip size of 4 square microns, which is several orders of magnitude smaller. This miniaturization is essential for integration into circuits and for enabling high-performance AI computing chips. It is worth mentioning that we could easily increase the lasing area by scaling up the chip size. Overall, these results demonstrate that the chip designed by CAKES remains competitive or even superior to these additional baselines.

## G.4 ROBOT PUSHING

The robot pushing problem was initially considered by Wang et al. (2018). The objective is to develop a controller for two robotic hands tasked with pushing two objects to specified target positions. The robot controller is defined by $d = 14$ parameters that determine the position and orientation of the hands, the pushing speed, the direction of movement, and the duration of the push. The reward function is given by:

$$f(\mathbf{x}) = -\sum_{i=1}^{2} |\mathbf{x}_{gi} - \mathbf{x}_{si}| - |\mathbf{x}_{gi} - \mathbf{x}_{fi}| \tag{41}$$

where $\mathbf{x}_{si}$ represents the initial positions of the objects, $\mathbf{x}_{fi}$ denotes their final positions, and $\mathbf{x}_{gi}$ indicates the goal . The objective is to minimize the total distance from the initial and final positions of the objects to their respective goal positions, thus maximizing the reward. We run each method for a total of $T = 100$ iterations using 10 different random seeds. To evaluate the performance of each method, we conduct a series of simulations, running each method for a total of $T = 100$ iterations, using 10 different initial positions. The results, illustrated in Figure 13, show that the CAKES method outperformed the other approaches in terms of achieving the highest reward and minimizing the regret.

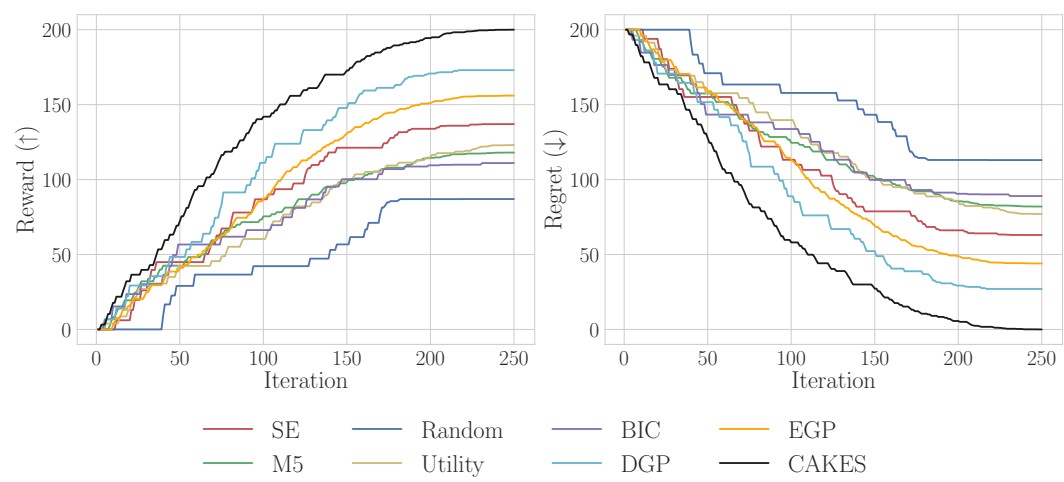

Figure 14: Reward ($\uparrow$) and regret ($\downarrow$) for the lunar lander problem.

### G.5 LUNAR LANDER

We consider tuning a controller for a lunar lander implemented in OpenAI Gym[2]. The state space of the lunar lander consists of its position, angle, time derivatives, and indicators of whether either leg is in contact with the ground. This results in an 8-dimensional state vector, which is passed to a handcrafted parameterized controller that determines the appropriate action to take. There are four possible actions for each frame: firing the main engine, fire the left orientation engine, fire the right orientation engine, or doing nothing. The handcrafted control policy is defined by $d = 12$ parameters, which parameterize linear score functions based on the state vector and establish thresholds that dictate which action to prioritize. The reward system is defined as follows:

- Successful landing yields $+100$ points, while crashing incurs a penalty of $-100$ points.
- Each leg in contact with the ground yields $+10$ points.
- Firing the main engine incurs a penalty of $-0.3$ points per frame, and firing side engines incurs a penalty of $-0.03$ points per frame.

The benchmark for winning the game is 200 points, so we use it as the target objective. We run the simulation for $T = 250$ iterations and average the results over 50 randomly generated terrains, initial positions, and velocities. The results, illustrated in Figure 13, demonstrate that the controller optimized by CAKES significantly outperforms other methods, achieving both the highest rewards and the lowest regret. Specifically, CAKES consistently reached or exceeded the target objective of 200 points across various terrains, while the other methods struggled to maintain high scores and often incurred greater penalties due to crashes or inefficient landings. In our simulations, we also found that CAKES not only optimized the landing strategies but also minimized unnecessary engine usage, resulting in lower penalties over time.

## H LIMITATIONS AND FUTURE WORK

**Computational cost.** While CAKES can be applied entirely in-context and does not require any fine-tuning, using LLMs for inference may result in a larger computational footprint compared to traditional BO methods. Despite this, our findings indicate that CAKES trades this off with improved sample efficiency, which is a particularly desirable property for black-box optimization tasks. This suggests the potential for integrating CAKES with more computationally efficient approaches, such as deploying it in the earlier stage of the optimization process.

**Choice of LLM.** We have demonstrated the potential of integrating LLMs to enhance kernel design in BO using the `gpt-4o-mini` model. We chose `gpt-4o-mini` as it offers an excellent balance

---

[2]https://www.gymlibrary.dev/environments/box2d/lunar_lander/

of affordability, fast inference, and intelligence for our implementation. While we acknowledge that the choice of the LLM may influence the results, the overall methodology and key insights presented in this work are generally applicable, and extend beyond the specifics of any particular LLM. A promising direction for future research would be to benchmark various LLMs to better understand their strengths and limitations across different BO problem settings.

**Generalized kernel grammar.** While we focus on addition and multiplication as initial proof-of-concept operators, these operators are in fact good enough to form a rich and expressive space of kernels. For example, by only using these operations, we can construct polynomial kernels to capture non-linear patterns as well as multi-dimensional kernels to model interactions among input features (Duvenaud et al., 2013). However, we would like that to note that the kernel grammar can be extended using other operators that preserve the closure properties of kernel functions, such as convolution, composition, and affine transformations (Smola & Schölkopf, 1998). We aim to explore these possibilities further in future work.

**CAKES-NoInfo**

~~You are an expert in machine learning, specializing in Gaussian processes.~~ Here are the observations we have collected so far: {observations}

Please analyze these observations to identify patterns in the data that can be captured by a kernel function. ~~You can use squared exponential (SE) kernel to capture smoothness, periodic (PER) kernel to capture periodicity, linear (LIN) kernel to capture linear trends, and rational quadratic (RQ) kernel to capture varying data patterns. You can also combine these kernels using the + and * operators to capture more complex patterns. For example, LIN + PER can capture a linear trend with periodic fluctuations and LIN * PER can capture a periodic pattern with linearly increasing amplitude.~~

- - - - - - - - - - - - - - - - - - - - - - - - - - - - - - - - - - - - - - - - - -

You are given two parent kernels and their Bayesian Information Criterion (BIC) values:
{parent_kernel1} (BIC: {bic1}),
{parent_kernel2} (BIC: {bic2}).

Please recommend a kernel that has a lower BIC value. You can either combine the parent kernels with + or * operator. Your output should follow the following format:

Kernel: <your proposed kernel here, only use the kernel abbreviations and operators>
Analysis: <your analysis here, explaining your reason behind the proposed kernel>

- - - - - - - - - - - - - - - - - - - - - - - - - - - - - - - - - - - - - - - - - -

You are given a kernel and its Bayesian Information Criterion (BIC) value:
{kernel} (BIC: {bic}).

Please recommend a kernel that has a lower BIC value. You can only replace one of the base kernels in the kernel with another base kernel. Your output should follow the following format:

Kernel: <your proposed kernel here, only use the kernel abbreviations and operators>
Analysis: <your analysis here, explaining your reason behind the proposed kernel>

Figure 15: The complete prompt for CAKES-NoInfo. ~~Strikethrough~~ indicate the parts that are removed.

**CAKES-NoContext**

You are an expert in machine learning, specializing in Gaussian processes. ~~Here are the observations we have collected so far: {observations}~~

~~Please analyze these observations to identify patterns in the data that can be captured by a kernel function.~~ You can use squared exponential (SE) kernel to capture smoothness, periodic (PER) kernel to capture periodicity, linear (LIN) kernel to capture linear trends, and rational quadratic (RQ) kernel to capture varying data patterns. You can also combine these kernels using the $+$ and $*$ operators to capture more complex patterns. For example, LIN $+$ PER can capture a linear trend with periodic fluctuations and LIN $*$ PER can capture a periodic pattern with linearly increasing amplitude.

- - - - - - - - - - - - - - - - - - - - - - - - - - - - - - - - - - - - - - - - - - - - -

You are given two parent kernels ~~and their Bayesian Information Criterion (BIC) values~~:
{parent_kernel1} ~~(BIC: {bic1})~~,
{parent_kernel2} ~~(BIC: {bic2})~~.

Please recommend a kernel that has a lower BIC value. You can either combine the parent kernels with $+$ or $*$ operator. Your output should follow the following format:

Kernel: <your proposed kernel here, only use the kernel abbreviations and operators>
Analysis: <your analysis here, explaining your reason behind the proposed kernel>

- - - - - - - - - - - - - - - - - - - - - - - - - - - - - - - - - - - - - - - - - - - - -

You are given a kernel ~~and its Bayesian Information Criterion (BIC) value~~:
{kernel} ~~(BIC: {bic})~~.

Please recommend a kernel that has a lower BIC value. You can only replace one of the base kernels in the kernel with another base kernel. Your output should follow the following format:

Kernel: <your proposed kernel here, only use the kernel abbreviations and operators>
Analysis: <your analysis here, explaining your reason behind the proposed kernel>

Figure 16: The complete prompt for CAKES-NoContext. ~~Strikethrough~~ indicate the parts that are removed.

---

**CAKES-NoInstruct**

You are an expert in machine learning, specializing in Gaussian processes. Here are the observations we have collected so far: {observations}

Please analyze these observations to identify patterns in the data that can be captured by a kernel function. You can use squared exponential (SE) kernel to capture smoothness, periodic (PER) kernel to capture periodicity, linear (LIN) kernel to capture linear trends, and rational quadratic (RQ) kernel to capture varying data patterns. ~~You can also combine these kernels using the + and ∗ operators to capture more complex patterns. For example, LIN + PER can capture a linear trend with periodic fluctuations and LIN ∗ PER can capture a periodic pattern with linearly increasing amplitude.~~

- - - - - - - - - - - - - - - - - - - - - - - - - - - - - - - - - - - - - - - - - - -

You are given two parent kernels and their Bayesian Information Criterion (BIC) values:
{parent_kernel1} (BIC: {bic1}),
{parent_kernel2} (BIC: {bic2}).

Please recommend a kernel that has a lower BIC value. ~~You can either combine the parent kernels with + or ∗ operator.~~ Your output should follow the following format:

Kernel: <your proposed kernel here, only use the kernel abbreviations and operators>
Analysis: <your analysis here, explaining your reason behind the proposed kernel>

- - - - - - - - - - - - - - - - - - - - - - - - - - - - - - - - - - - - - - - - - - -

You are given a kernel and its Bayesian Information Criterion (BIC) value:
{kernel} (BIC: {bic}).

Please recommend a kernel that has a lower BIC value. ~~You can only replace one of the base kernels in the kernel with another base kernel.~~ Your output should follow the following format:

Kernel: <your proposed kernel here, only use the kernel abbreviations and operators>
Analysis: <your analysis here, explaining your reason behind the proposed kernel>

---

Figure 17: The complete prompt for CAKES-NoInstruct. ~~Strikethrough~~ indicate the parts that are removed.

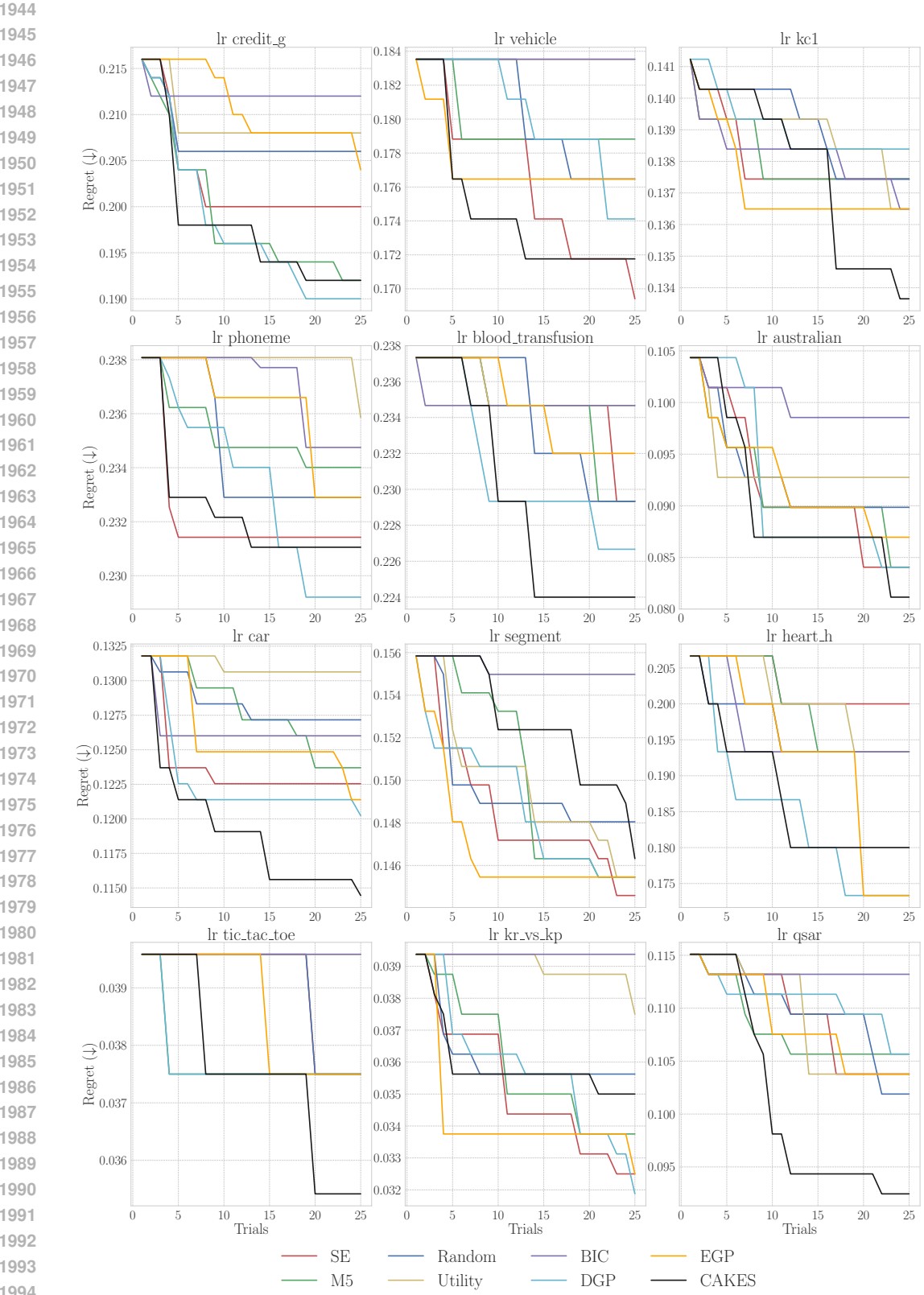

Figure 18: Average regret (↓) over 5 random seeds for the LR model tuned with different methods on individual tasks.

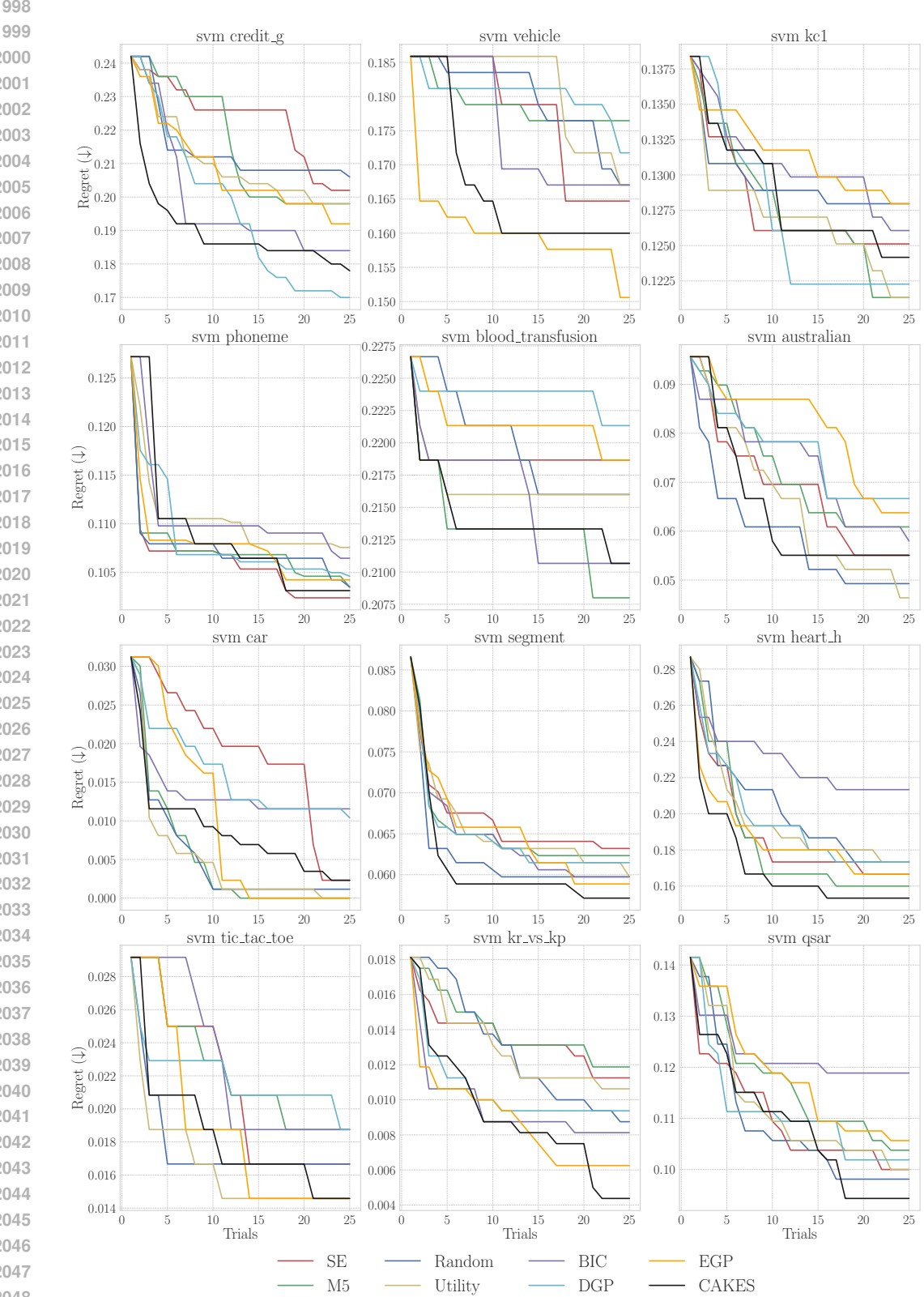

Figure 19: Average regret (↓) over 5 random seeds for the SVM model tuned with different methods on individual tasks.

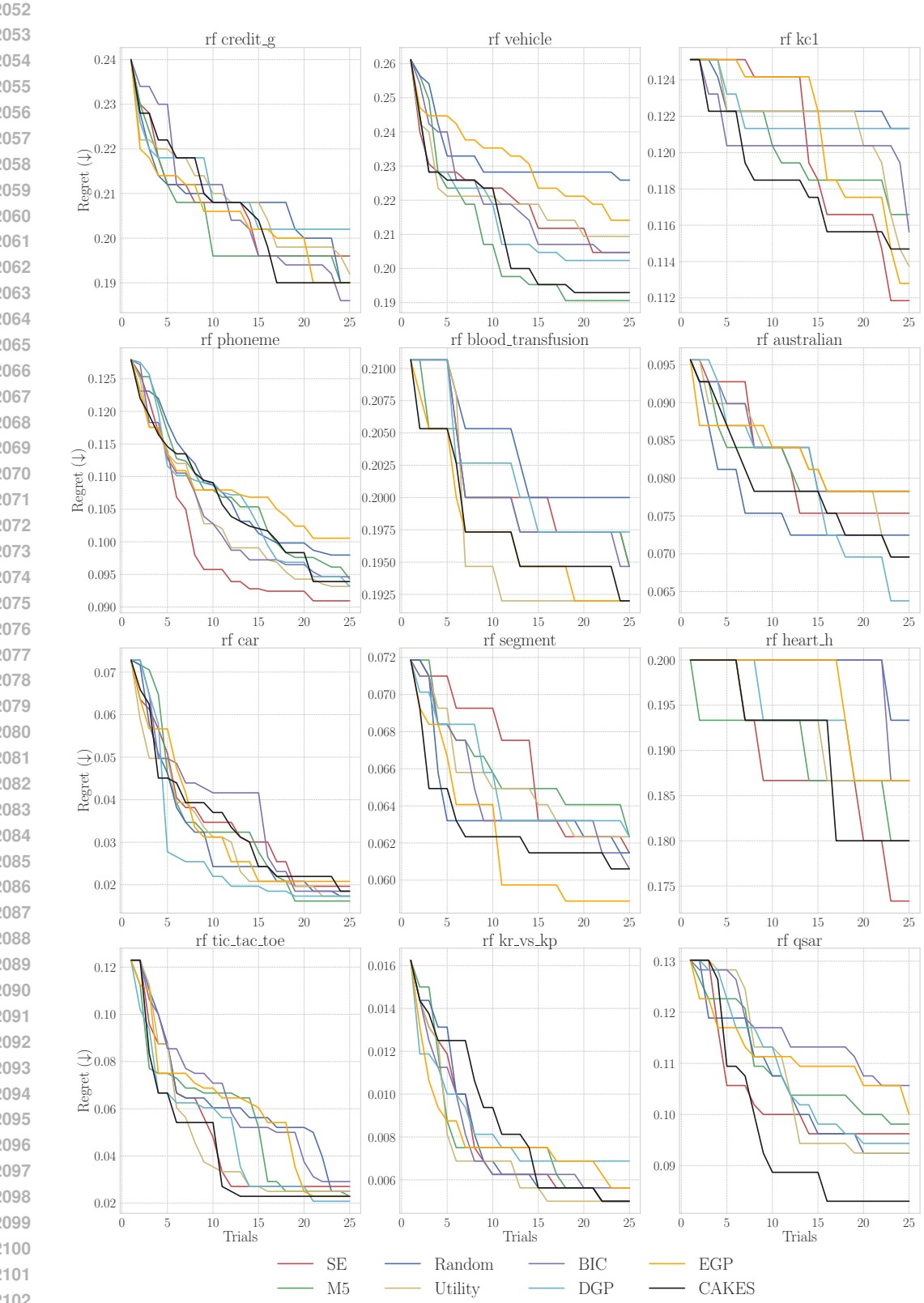

Figure 20: Average regret (↓) over 5 random seeds for the RF model tuned with different methods on individual tasks.

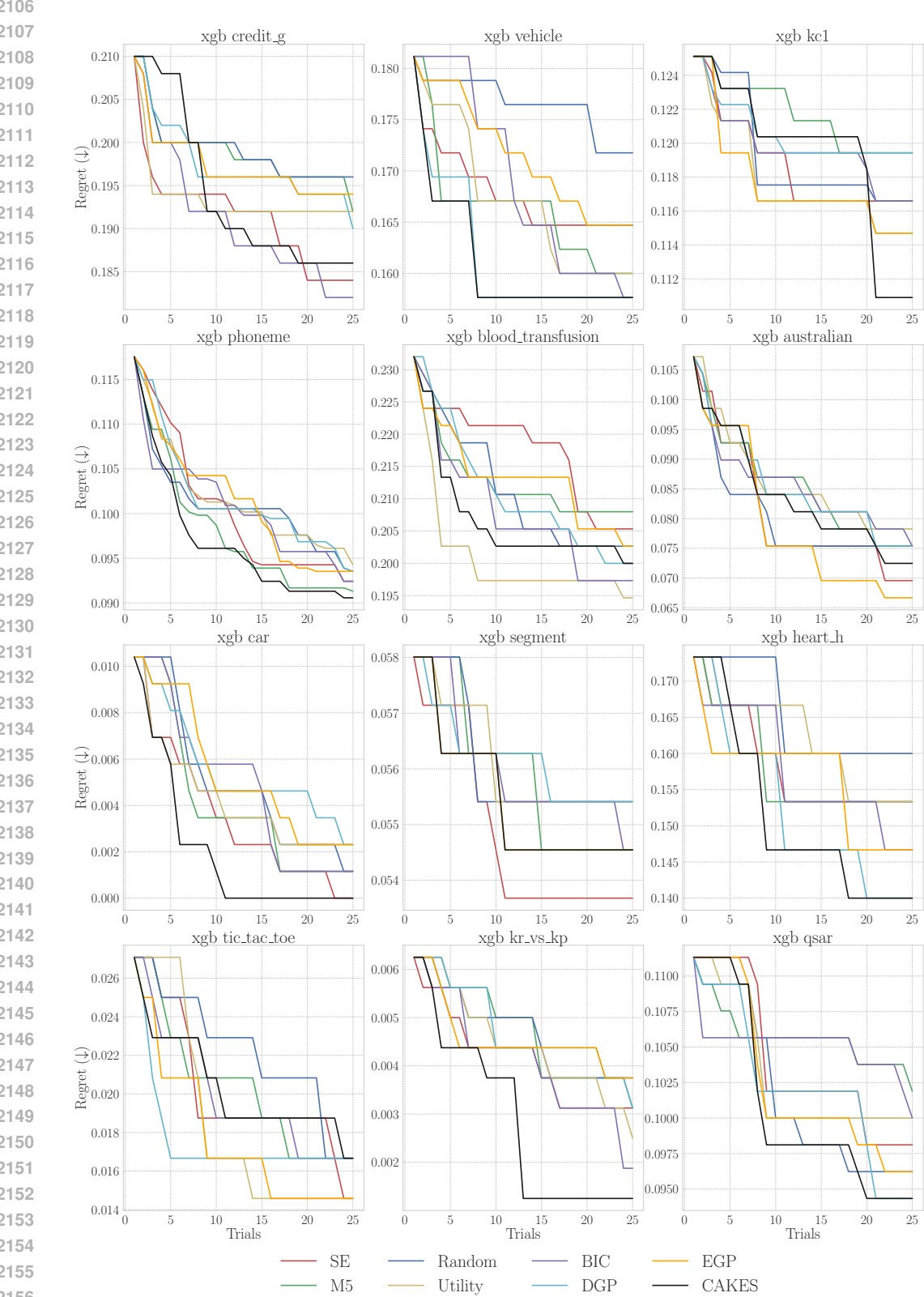

Figure 21: Average regret (↓) over 5 random seeds for the XGB model tuned with different methods on individual tasks.

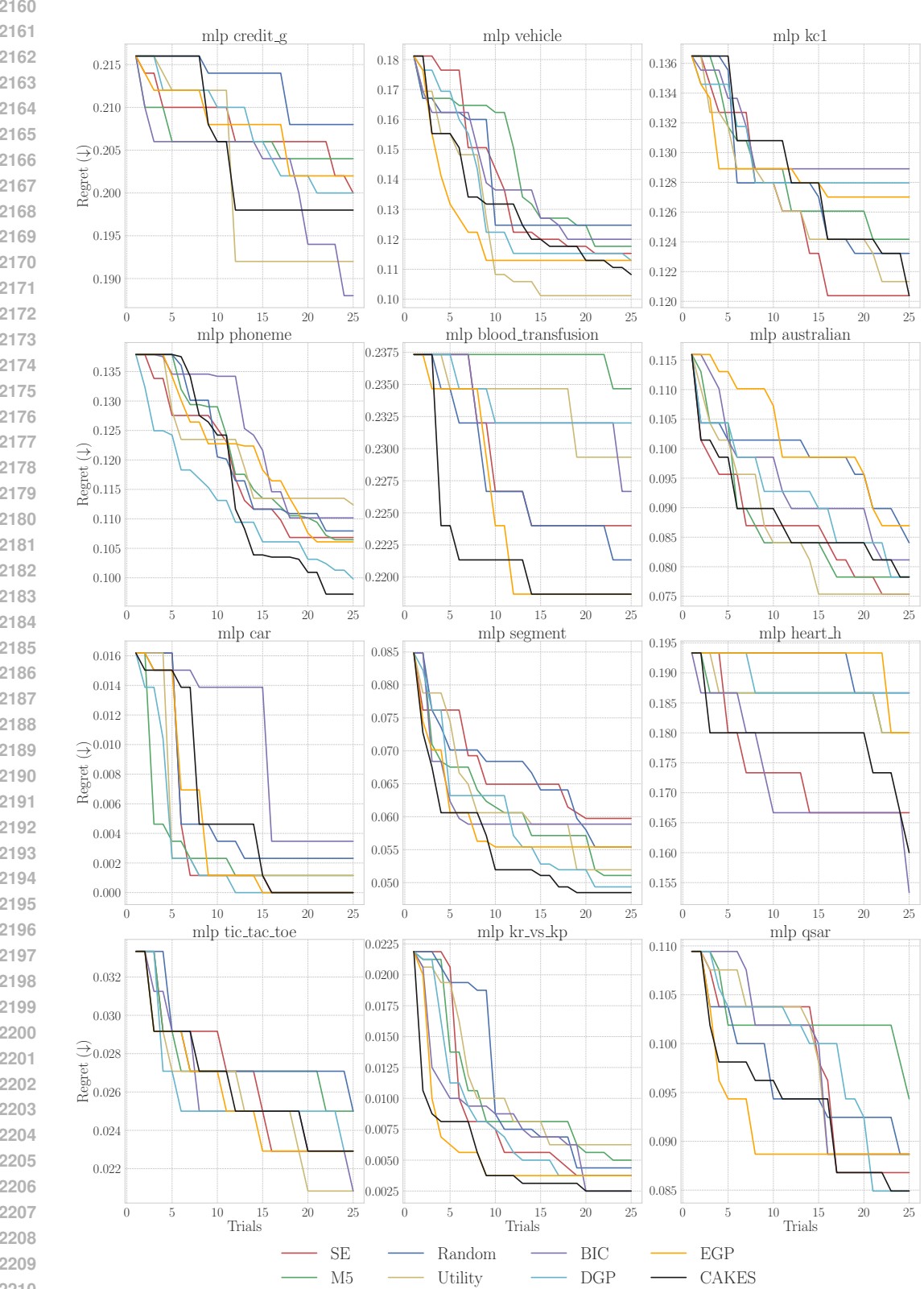

Figure 22: Average regret (↓) over 5 random seeds for the MLP model tuned with different methods on individual tasks.

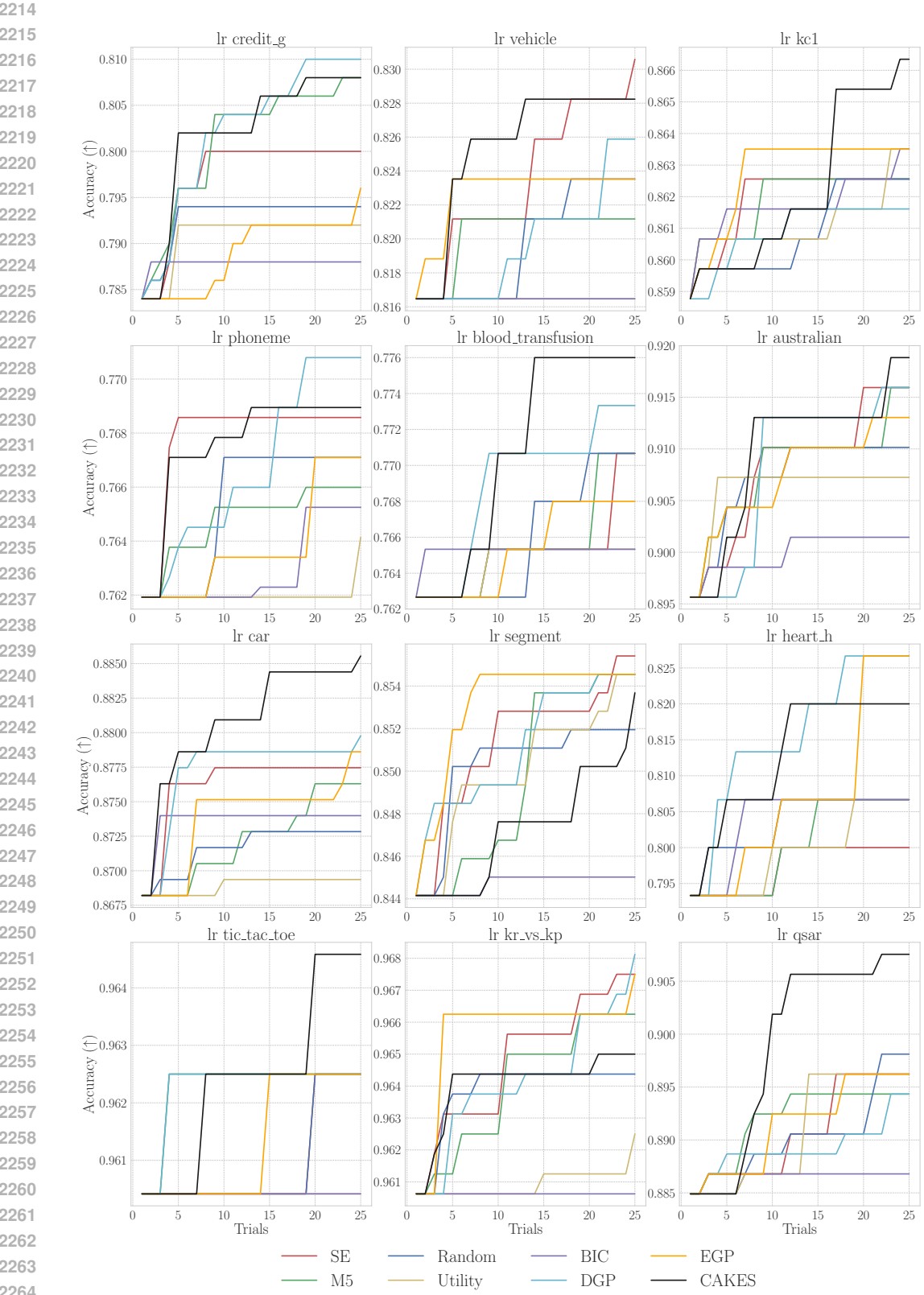

Figure 23: Average accuracy (↑) over 5 random seeds for the LR model tuned with different methods on individual tasks.

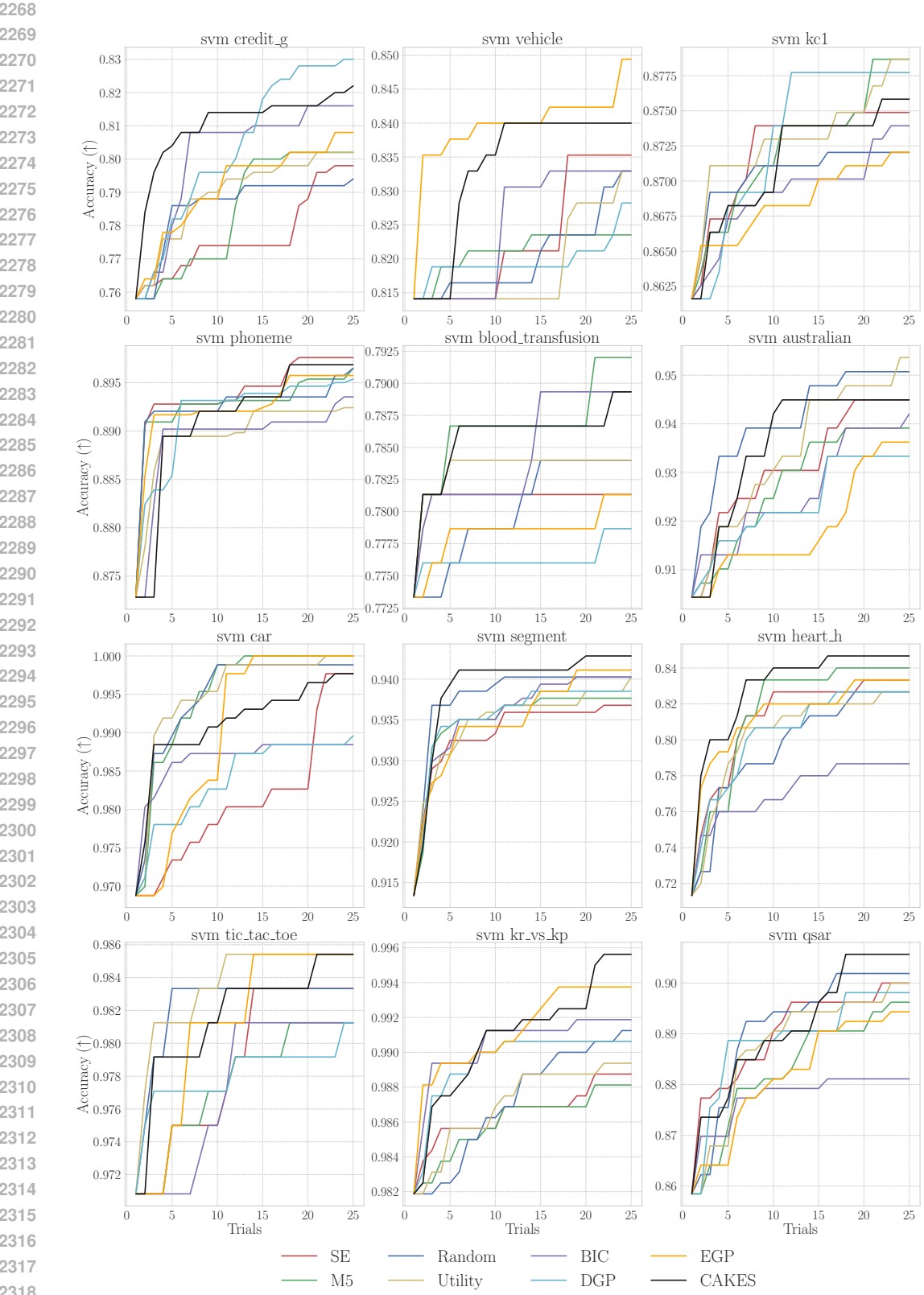

Figure 24: Average accuracy (↑) over 5 random seeds for the SVM model tuned with different methods on individual tasks.

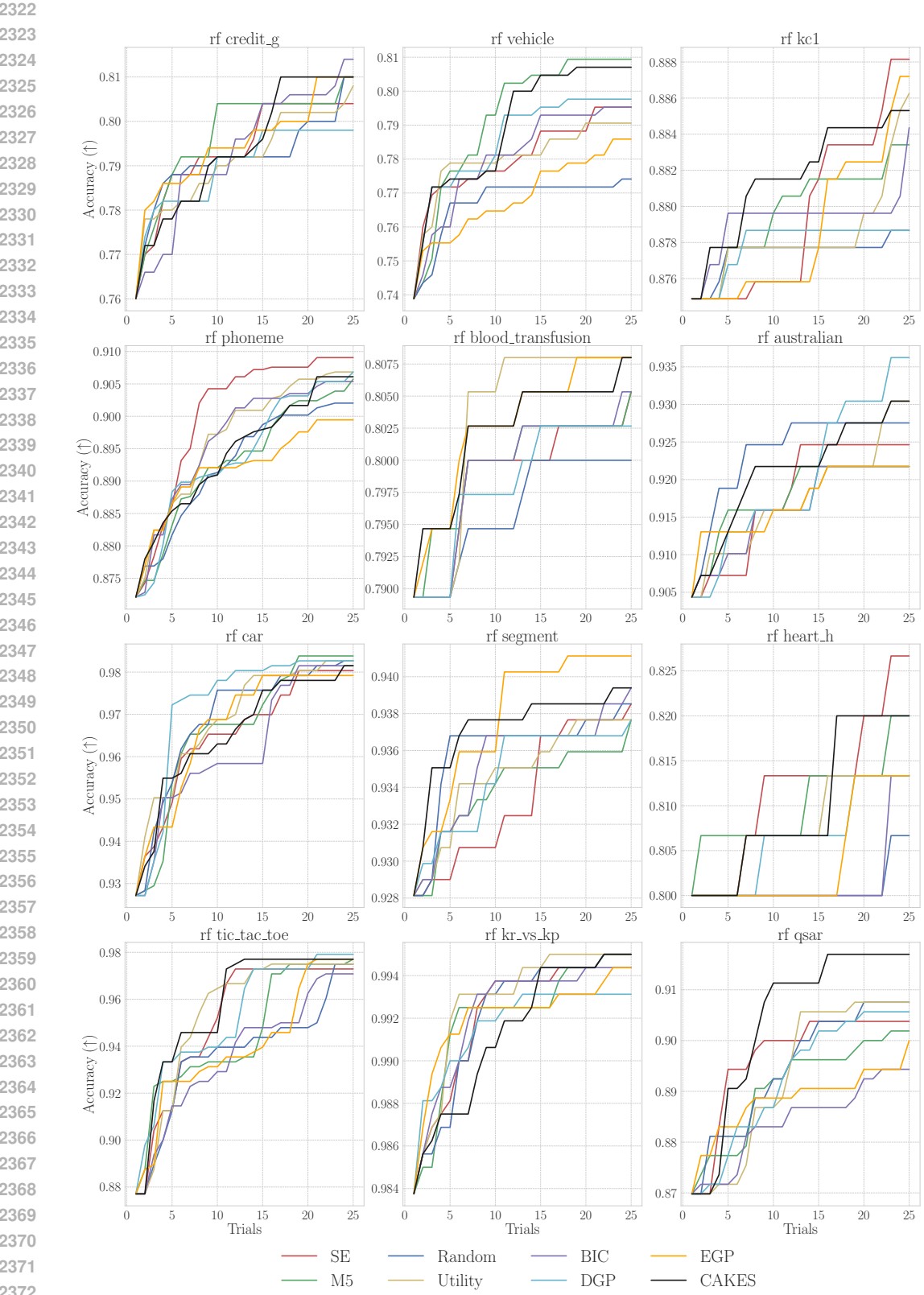

Figure 25: Average accuracy (↑) over 5 random seeds for the RF model tuned with different methods on individual tasks.

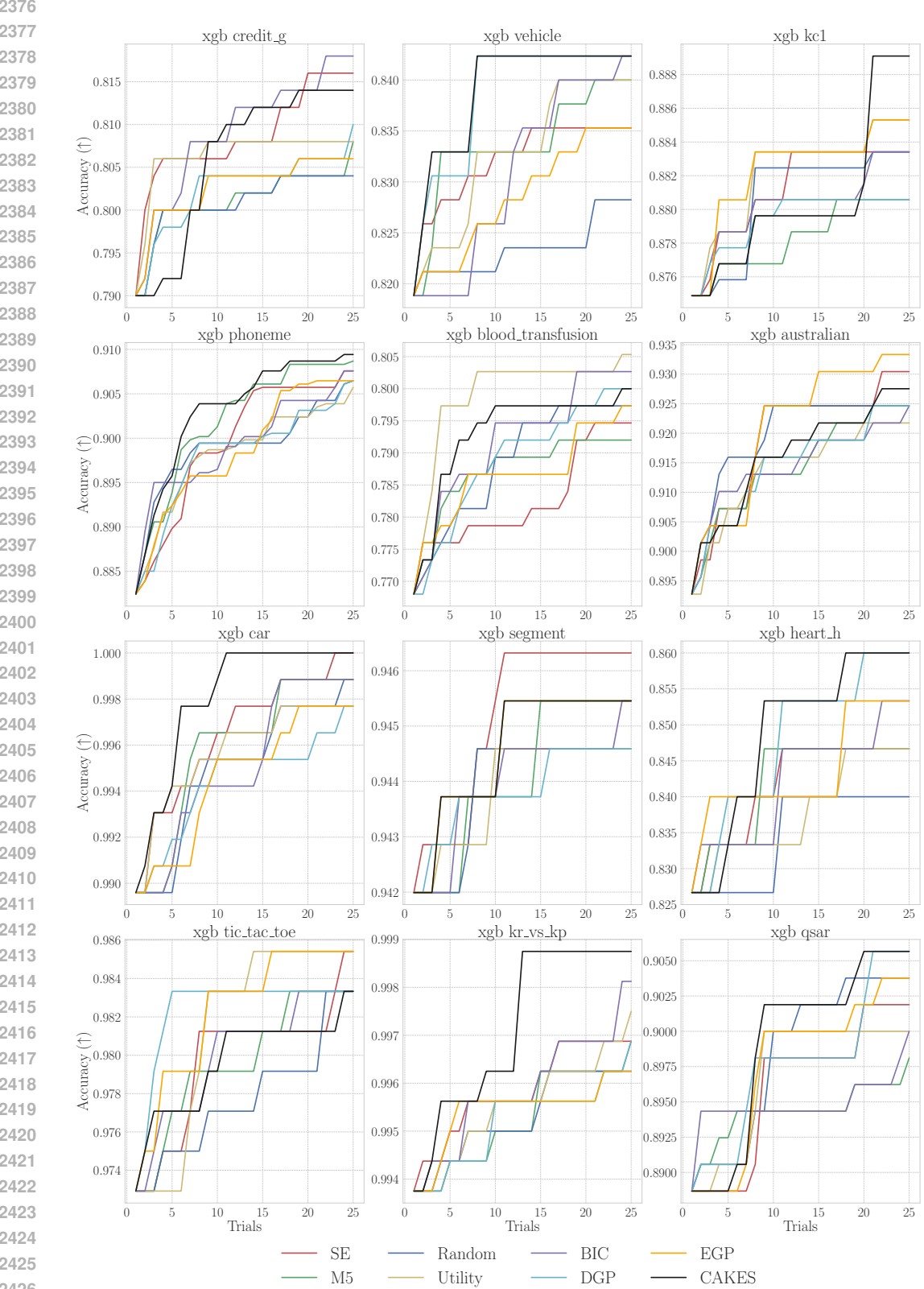

Figure 26: Average accuracy (↑) over 5 random seeds for the XGB model tuned with different methods on individual tasks.

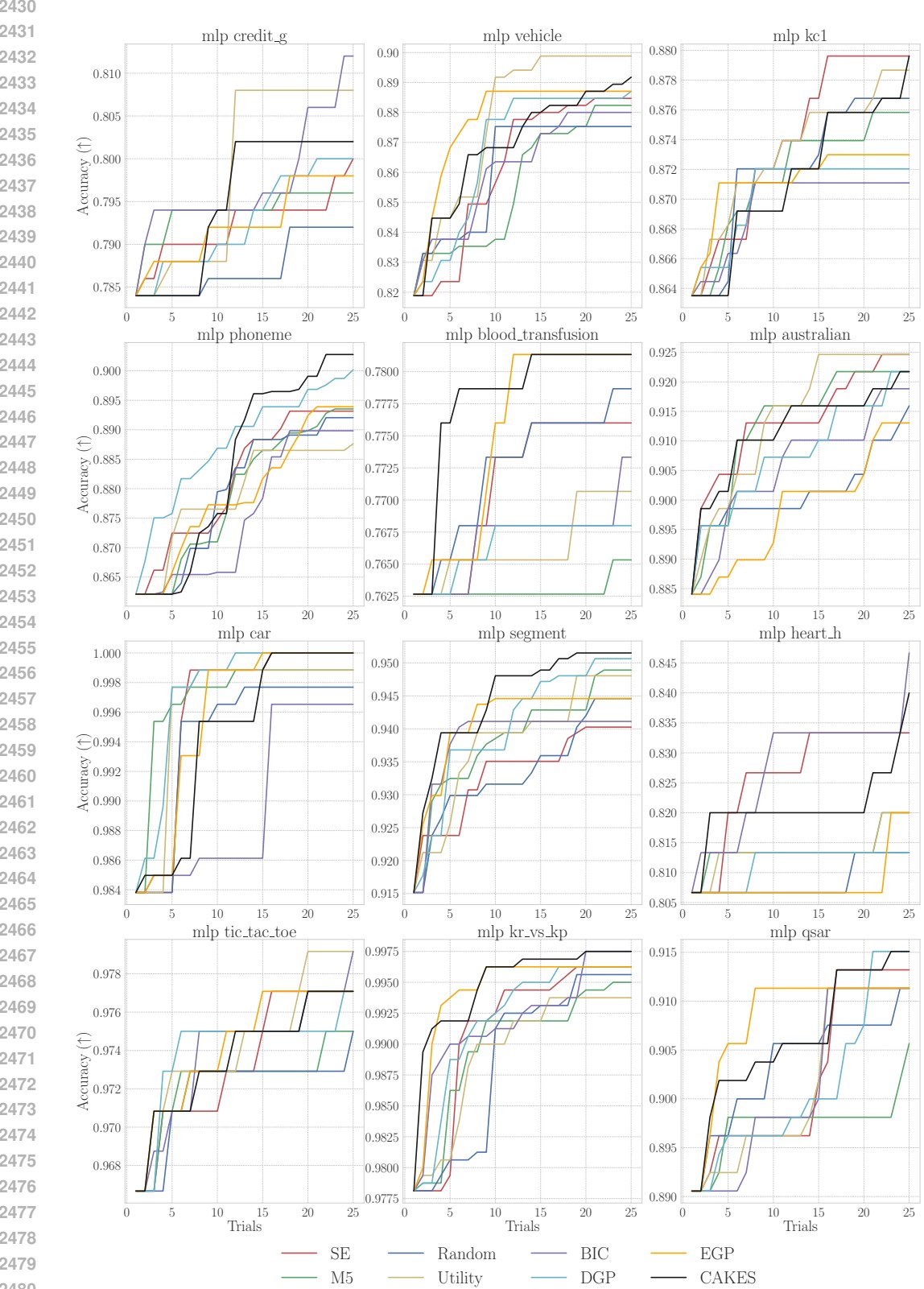

Figure 27: Average accuracy (↑) over 5 random seeds for the MLP model tuned with different methods on individual tasks.

