# OpenReview forum: "Context-Aware Kernel Search for Bayesian Optimization with Large Language Models"
_ICLR.cc/2025/Conference — ICLR 2025 Conference Withdrawn Submission_

### Official Review · Reviewer_Bsdq · 2024-10-28

**Soundness:** 4
**Presentation:** 3
**Contribution:** 3
**Rating:** 8
**Confidence:** 4

**Summary:**

Gaussian processes play a key role in Bayesian optimization, where the quality of the model heavily depends on the selection of the kernel type. This paper aims to address this selection issue with the assistance of large language models. Extensive experiments are conducted to verify the superior performance of their proposed framework.

**Strengths:**

The topic is compelling, as kernel design is crucial in Gaussian process modeling. The paper is well-written and easy to follow. The method's performance is excellent, potentially reducing the selection cost for BO practitioners.

**Weaknesses:**

- The presentation of the algorithm places too much emphasis on kernel search. However, BO is just one application of kernel design in Machine Learning. Therefore, I recommend the authors refine the presentation of this BO-specific method.
- The theoretical analysis is too simplistic and could apply to any adaptation method of kernel design, such as symbolic regression. LLM-related analysis may better describe the performance.

**Questions:**

- I am curious whether the setting of weights in photonic chip design, where the last three objectives are much more important than the first two, is a commonly recognized practice or carefully designed. This is particularly relevant as CAKES performs outstandingly well on the fourth objective but only fairly on the other objectives.

---

> ### Author Response · Authors · 2024-11-22
> **Response to Reviewer Bsdq (1/1)**
>
> Thank you for your time and effort in reviewing our work. Important changes have been highlighted with blue color in the manuscript. We respond to the questions and comments point-by-point as follows.
>
> ---
> > The presentation of the algorithm places too much emphasis on kernel search. However, BO is just one application of kernel design in Machine Learning. Therefore, I recommend the authors refine the presentation of this BO-specific method.
>
> Thank you for your feedback. We would like to clarify that our algorithm is not exclusively an application of BO. Instead, our algorithm serves as a **general framework for kernel design** that can automate surrogate modeling (with GP) in BO. We also emphasized this point in the conclusion part, where we hope that this work can represent a solid step towards a **fully automated BO system** that can be used by **non-experts on arbitrary objectives**.
>
> ---
> > The theoretical analysis is too simplistic and could apply to any adaptation method of kernel design, such as symbolic regression. LLM-related analysis may better describe the performance.
>
> Thank you for your insightful feedback regarding our theoretical analysis. We would like to first mention that our current analysis focuses on the regret bounds and information gain, which are the commonly-used tools for analyzing the performance of BO algorithms. However, we appreciate your suggestion to include an LLM-related analysis. In response, **we have added a new section in the Appendix**, where we provide empirical evidence that LLMs are indeed familiar with kernel design and can function effectively as good crossover and mutation operators. In Appendix F.1, **we have conducted an additional experiment** to analyze the fitness distributions of an initial population of kernels before and after applying crossover and mutation operations. Our results show that the initial population has a wide distribution with a peak at lower fitness values; after crossover, **the distribution shifts toward higher fitness values**, indicating the LLM's capacity to effectively combine kernels and enhance overall fitness. Furthermore, we found that applying mutation **further shifts the distribution toward even higher fitness values**, suggesting that refining the kernels yields improved outcomes. In Appendix F.2, **we have included additional analysis** reasoning the outputs from the LLM during the crossover and mutation steps. Based on the outputs (see Figure 9), we found that the LLM demonstrates a **strong understanding of the relationships between different kernels and their use cases** to address specific data characteristics.
>
> ---
> > I am curious whether the setting of weights in photonic chip design, where the last three objectives are much more important than the first two, is a commonly recognized practice or carefully designed. This is particularly relevant as CAKES performs outstandingly well on the fourth objective but only fairly on the other objectives.
>
> Thank you for the question. The last three objectives are indeed **much more important** than the other ones from a practical engineering (e.g., autonomous driving or military laser weapon) point of view. This is because we want the photonic chip to be equipped with large lasing area $(f_3)$ to produce a **large light field coverage**, higher power $(f_4)$ to **transmit signal/beam faster and farther**, and have a small divergence angle $(f_5)$ to **travel longer distance and focus better**. In this case, $f_1$ and $f_2$ are just auxiliary objectives. Since CAKES performed particularly well on the fourth objective, the designed photonic chip is **well-suited for LiDAR applications** (e.g., for autonomous driving), which demand an outstanding laser power $(f_4)$.

---

> > ### Comment · Reviewer_Bsdq · 2024-11-22
> >
> > Thank you for your responses. The work makes certain contribution to the community. I will maintain my recommendation for acceptance.

---

> > > ### Author Response · Authors · 2024-11-22
> > > **Reply to Reviewer Bsdq**
> > >
> > > Thank you for taking the time to review our paper and for your thoughtful comments. We sincerely appreciate your positive evaluation of our work.
> > >
> > > Building on your valuable insights, we kindly ask if there are any additional aspects or suggestions we could address to further enhance the quality and impact of the paper. We are more than willing to incorporate your recommendations to strengthen our contribution to the community.
> > >
> > > We look forward to any additional guidance you can provide to further improve our submission. Thank you once again for your feedback and for considering our work.

---

> > > > ### Author Response · Authors · 2024-11-29
> > > > **Updates for Reviewer Bsdq**
> > > >
> > > > Dear Reviewer Bsdq,
> > > >
> > > > Thank you once again for your valuable time and insightful feedback on our work. We have incorporated the following updates, especially regarding more LLM-related analysis, which we believe may be of interest to you:
> > > >
> > > > - **We conducted a prompt ablation study (see Appendix F.3) to analyze the different components of our prompt**. We found that **removing the domain-specific information (i.e., role assignment and kernel descriptions) does not negatively impact performance**. In fact, this variant still demonstrates **comparable performance** to the standard CAKES setup, suggesting that the **LLM possesses valuable prior knowledge that can help with kernel design**. However, when we remove the optimization history (i.e., observations and fitness values), the performance of this variant **degrades significantly**, indicating that **the optimization history is important and that the LLM can leverage this information effectively**. Finally, we considered a variant that excludes the explicit task instructions and kernel generation guidelines (e.g., how to apply crossover and mutation operators). We observed that this variant exhibits **significantly poorer performance**, primarily due to its **inability to generate valid kernels**. Overall, these findings demonstrate the importance of each component in our prompt to achieve a superior performance.
> > > > - **We have also conducted an additional experiment in Appendix F.4 to analyze the impact of different LLMs** on the performance of our method. **We considered several state-of-the-art language models**, including Meta's `llama-3.1-8b`, Google's `gemini-1.5-flash`, Anthropic's `claude-3.5-haiku`, and OpenAI's `gpt-4o-mini`. The results support our decision to choose `gpt-4o-mini` as the default model for CAKES, as it provides an **excellent balance of cost, speed, and intelligence**. While different LLMs may yield varying empirical results, we believe that **the overall methodology and key insights presented in our work extend beyond the specifics of any particular model**.
> > > >
> > > > We hope that above updates enhance your impression of our work and strengthen our contributions. If you any additional suggestions or concerns, we would be happy to discuss them further. If your concerns have been adequately addressed, we would greatly appreciate if you could reconsider your score.
> > > >
> > > > With thanks,
> > > >
> > > > Authors of #5709

---

> > > > ### Comment · Reviewer_Bsdq · 2024-11-29
> > > >
> > > > Can the author demonstrate any well-performing kernel using this method?
> > > > From the information in Figure 9, LLMs recommend a simple combination of base kernels.
> > > > Can we use symbolic regression to find optimal kernels?

---

> ### Author Response · Authors · 2024-11-29
> **Reply to Reviewer Bsdq**
>
> Thank you for your quick reply and questions. We address your questions one by one as follows:
>
> ---
> > Can the author demonstrate any well-performing kernel using this method? From the information in Figure 9, LLMs recommend a simple combination of base kernels.
>
> Thank you for mentioning this point. First, we would like to clarify that Figure 9 only displays a sample output **from a single iteration**, particularly in the early stage, of the optimization process. The main purpose of Figure 9 is to illustrate that the **LLM has a strong understanding of the relationships between different kernels and their capacity to address specific data characteristics**. While the recommended kernel may appear to be a simple combination of base kernels, our method can **adaptively refine the kernel and generate more complex kernels in the subsequent iterations**. Unfortunately, due to space constraints, we were unable to present all outputs from each iteration across all tasks. Another point that we would like mention is that, **there is no single kernel that performs well universally** across all tasks. The optimal kernel in BO is **problem-specific** and **may change as new data is collected**. This is a key motivation behind our work, as fixed kernels or naive selection strategies often **fail to deliver good results** across different tasks (see Section 6 and Appendix G). In light of this, as part of our contribution, we proposed a method that leverages the extensive knowledge and learning power of LLMs to adaptively propose kernels, such that the kernels are **tailored to the specific characteristics of the data at each iteration**.
>
> ---
> > Can we use symbolic regression to find optimal kernels?
>
> Thanks for bringing up the idea of symbolic regression to find the optimal kernels. While we acknowledge that we only have a basic understanding of it, we recognize that there are **significant differences** between symbolic regression and our method. Symbolic regression aims to identify the optimal mathematical expression for a fitting function that minimizes fitting error. This process involves both determining the mathematical operations and estimating parameters, **which has been proven to be NP-hard** [1]. We also afraid that **the computational time required for symbolic regression may be prohibitive in both modest and high-dimensional settings**. In contrast, our focus is on efficiently **identifying the best reproducing kernel Hilbert space (RKHS) for a surrogate function in BO** with the assistance of LLMs. While regression and data fitting are not our primary objectives, we prioritize **a more robust uncertainty quantification** associated with the determined RKHS. This uncertainty quantification is crucial to effectively guide the BO process by **balancing exploration and exploitation effectively**. Besides, we maintain a practical mindset and aim to **balance the accuracy, computation time, and cost**. Another notable difference is that we need to make sure the kernel found by our method is a **valid kernel**, while the search under the symbolic regression has **no such constraint on the math operations**. Nevertheless, we agree that integrating the idea of symbolic regression (especially the genetic programming) with LLM prompt design **may facilitate our kernel search** when the kernel operations are not limited to addition and multiplication. Thank you once again for sharing this interesting idea with us; we will certainly consider this new line of research in the future.
>
> ---
> We hope this addresses your questions and enhances your impression of our work. We remain open to any further questions or suggestions. If your concerns are resolved, we would greatly appreciate if you could reconsider your score.
>
> ---
> ### References
>
> [1] M. Virgolin and S. P. Pissis, "Symbolic regression is NP-hard," *Transactions on Machine Learning Research*, 2022.

---

> > ### Comment · Reviewer_Bsdq · 2024-11-30
> >
> > I will reconsider the score, and it may help the reader get some inituitions about the mechanism if you could illustrate the kernel population at a specific iteration.

---

> > > ### Author Response · Authors · 2024-12-01
> > > **Reply to Reviewer Bsdq**
> > >
> > > Thank you for reconsidering your score. We appreciate your suggestion and we will certainly add a new figure to illustrate this in the camera-ready version.

---

### Official Review · Reviewer_BRo2 · 2024-10-30

**Soundness:** 2
**Presentation:** 3
**Contribution:** 2
**Rating:** 5
**Confidence:** 3

**Summary:**

This submission proposes a novel context-aware kernel search (CAKES) for constructing surrogate models in Bayesian Optimization. CAKES replies on LLM to operate the proposed crossover and mutation method for kernel design. Evolutionary approach is adapted to construct new kernels for the GP models. Some theoretical analysis is presented to show a sub-linear regret bound of the proposed method. Experiments on benchmarks and real-world applications are performed to show CAKES outperforms other surrogate models in BO.

**Strengths:**

The paper is well structured and fluent to read. The idea of CAKES is well explained in this submission. Motivation for a novel surrogate model design method in BO is convincing. The usage of a developed and trained LLM without any fine-tuning in the few-shot learning setting of BO is new. Experiments consider both synthetic functions and real-world application that shows practical potential of CAKES.

**Weaknesses:**

1. This submission does not present adequate contributions in my opinion. The idea of kernel design using transformers and language models has been studied in the field. For example, Simpson et al. (https://proceedings.neurips.cc/paper/2021/hash/56c3b2c6ea3a83aaeeff35eeb45d700d-Abstract.html) proposed their transformer-based architecture for kernel design, and their model is specifically trained using a vocabulary of known kernels. The authors claim the "no need for fine tuning" as an advantage for their method, but feels more like the key weakness to me. Considering all the conditioning (line 193-194, 198-199) and numerical information (such as observations and kernel's characteristic) are fed to the LLM model purely through text prompt, CAKES's performance highly (or even entirely) depends on how familiar the chosen LLM model is with GP and kernel design. In addition, the crossover and mutation operators (line 227-234) utilize very standard kernel grammar and simple replacement, which are already well-developed techniques.

2.  Experiments design is not very consistent. In section 6, the authors use 5 different metrics to measure the performance of methods in different experiments. Given the BO settings, I don't see a good reason for varying performance metric between different experiments when the authors could just simply select one. Simple regret is commonly used and more importantly consistent with the regret bound proven in section 4. The "average rank" metric presented in Table 2 seems to highlight the same information as the "average regret" plots in Figure 2.

**Questions:**

1. In line 406-408, the authors mention SE and Matern-5/2 as the two most commonly-used kernels. Why is Matern-5/2 kernel not included in the four base kernels defined in line 203 then?

2.  How do the authors define the values of number of crossovers $n_c$, probability of mutation $p_m$ and population size $n_p$ (line 163-165 in Algorithm 1) in final implementation?

3.  In line 235-236, the authors mention a brief analysis will be returned by the LLM model. Can the authors provide one actual example of such analysis?

4.  I'm a bit confused by the weighting strategy explained in line 242-243. Does it mean the most ideal kernel with very low BIC (thus low weight) and high acquisition value might not be selected while some worse kernel (higher BIC, lower acquisition value) might? Could the authors further justify the intuition behind such strategy?

5.  In Figure 2, what is "Best" (the fourth method in legend)? Does it refer to the "Utility" method in line 411?

6.  I notice the authors set the maximum number of function evaluation $T$ and number of replications using different random seeds quite small for their experiments. For the benchmark functions (section 6.1), $T = 10 d \leq 50$ since $\max d = 5$ for the chosen test functions. For the hyperparameter tuning tasks (section 6.2), $T=25$. Usually the value of $T$ will be larger (e.g. $50$, $100$) in BO experiments. Similarly the number of random seeds for each experiment (10 for section 6.1, 5 for section 6.2 and not mentioned for section 6.3) is relatively less than what people usually use (e.g. $20$, $25$). How many random seeds did the author use for experiment in section 6.3? Is the reason for such not very generous setting due to long generation time of kernel design?

---

> ### Author Response · Authors · 2024-11-22
> **Response to Reviewer BRo2 (1/4)**
>
> Thank you for your time and effort in reviewing our work. Important changes have been highlighted with blue color in the manuscript. We respond to the questions and comments point-by-point as follows.
>
> ---
> > This submission does not present adequate contributions in my opinion. The idea of kernel design using transformers and language models has been studied in the field. For example, [Simpson et al.](https://proceedings.neurips.cc/paper/2021/hash/56c3b2c6ea3a83aaeeff35eeb45d700d-Abstract.html) proposed their transformer-based architecture for kernel design, and their model is specifically trained using a vocabulary of known kernels.
>
> Thank you for your valuable feedback. We appreciate the opportunity to clarify the contributions of our method, particularly in relation to existing works, including KITT by [1]. We outline the main differences between our method, CAKES, and KITT as follows:
>
> - Our contribution centers on developing **a novel method to automate kernel design toward more efficient BO**, whereas [1] only considered regression tasks. We believe that **the goals of these two tasks differ significantly**: BO emphasizes the balance between exploration and exploitation in the optimization landscapes, while regression focuses on fitting and prediction accuracy.
> - Our method employs in-context learning through few-shot prompting, allowing it to **adaptively generate and refine kernels** based on the observed data **without requiring extensive (pre)training**. This approach contrasts with KITT, which relies on a **fixed vocabulary of known kernels** and necessitates fine-tuning, thereby **limiting its flexibility and generalizability**.
> - Our approach leverages LLMs as **ready-to-use tools** that **can be used by non-experts and seamlessly integrated into various tasks without the need for fine-tuning.** It has been studied that fine-tuning can lead to a loss of general domain knowledge [2-4] and may not be practical or even feasible for every tasks. In contrast, KITT's reliance on fine-tuning **restricts its applicability beyond regression tasks**.
>
> We hope this clarifies the distinct contributions of our work and addresses your concerns.
>
> ---
> >  The authors claim the "no need for fine tuning" as an advantage for their method, but feels more like the key weakness to me. Considering all the conditioning (line 193-194, 198-199) and numerical information (such as observations and kernel's characteristic) are fed to the LLM model purely through text prompt, CAKES's performance highly (or even entirely) depends on how familiar the chosen LLM model is with GP and kernel design.
>
> Thank you for your comments regarding the effectiveness of our method. We acknowledge that the performance of CAKES does depend on the LLM's familiarity with GP and kernel design. However, we consider this reliance not a weakness but a central aspect of our work. Our primary research question investigates the extent to which LLMs, which are **trained on extensive internet-scale data**, possess **valuable prior knowledge** that can be leveraged to enhance kernel design without fine-tuning. We believe that the "no need for fine-tuning" is an advantage because it allows us to utilize LLMs' existing knowledge base, which is derived from **a vast amount of statistical information and code**, to perform tasks in GP kernel design. Recent studies [5-7] have demonstrated that **LLMs can recognize and extrapolate patterns from in-context demonstrations**. In Appendix F.1, **we have provided an empirical evidence** that LLM is indeed familiar with kernel design and can act as a good crossover and mutation operators. Specifically, **we have conducted an additional experiment** to analyze the fitness distributions of an initial population of kernels before and after applying crossover and mutation operations (see Figure 8). We found that the initial population tend to have a wide distribution, with a noticeable peak at lower fitness values. After performing crossover, **the distribution shifts slightly toward higher fitness values**, indicating that the LLM can effectively  combine the kernels and enhance the overall fitness of the population. **We further observed that applying mutation shifts the distribution toward even higher fitness values**, implying that refining the kernels can yield improved fitness values.

---

> ### Author Response · Authors · 2024-11-22
> **Response to Reviewer to BRo2 (2/4)**
>
> > In addition, the crossover and mutation operators (line 227-234) utilize very standard kernel grammar and simple replacement, which are already well-developed techniques.
>
> Thank you for your insightful feedback regarding the crossover and mutation operators used in our work. While these operators may be considered as standard and already well-established, they are in fact **good enough to form a rich and expressive space of kernels**. By repeatedly applying these operations, one can construct **polynomial kernels**, which can be used for capturing non-linear patterns in data. One can also apply these operations independently across each dimension to construct **multi-dimensional kernels**, which can model interactions between different input features. As we mentioned in Appendix H, while the current work focused on these two operators (addition and multiplication), **the kernel grammar can also be expressed using other operators** that preserve the closure properties of kernel functions, such as convolution, composition, and affine transformations [8]. In fact, as stated in Definition 1 of Section 3, our method is built based on this **generalized kernel grammar**, which implies that the other operators can be regarded as direct extensions of our method. We would like to explore these possibilities further in our future work.
>
> ---
> > Experiments design is not very consistent. In section 6, the authors use 5 different metrics to measure the performance of methods in different experiments. Given the BO settings, I don't see a good reason for varying performance metric between different experiments when the authors could just simply select one. Simple regret is commonly used and more importantly consistent with the regret bound proven in section 4.
>
> Thank you for your valuable comments regarding the consistency of our experiment design. In response to your comments, **we have replaced the GAP metric** previously used in Section 6.1 **with the regret metric**, as suggested by the reviewer. Correspondingly, **we have updated the results in Tables 1 and 4** to reflect the average regret obtained by each method. In Section 6.2, **we have used regret** to measure the performance in the hyper-parameter tuning tasks, and further **showcased the regret for each individual tasks** in Figures 14-18 of Appendix G.2. For Section 6.3, we would like to note that the global optimum is unknown for the photonics chip design problem. Therefore, we employed problem-specific metrics such as the score and hypervolume to evaluate the BO performance. We believe these metrics are **more relevant** to the specific problem of photonics chip design and provide **valuable interpretations** for experts and engineers in the field.
>
> ---
> > The "average rank" metric presented in Table 2 seems to highlight the same information as the "average regret" plots in Figure 2.
>
> Thank you for your feedback. We would like to first clarify that the "average rank" metric actually provides additional insights, as it is calculated based on the **performance on individual tasks**. In contrast, the "average regret" displayed in Figure 2 presents the **aggregated performance across all tasks**. In light of the reviewer's feedback, **we have decided to move the "average rank" results to Appendix G.2** as additional results to complement the main experiments. We believe this revision makes that our presentation more focused while still providing comprehensive insights.
>
> ---
> > In line 406-408, the authors mention SE and Matern-5/2 as the two most commonly-used kernels. Why is Matern-5/2 kernel not included in the four base kernels defined in line 203 then?
>
> Thank you for bringing up this important point. Initially, we selected the four base kernels (SE, PER, LIN, RQ) due to their simplicity and ease of combination. These kernels are **well-established in the literature**, particularly in the foundational work by [9], and provide a **solid base for constructing more complex kernels** using the grammar operators. In response to your comment, **we have revised our manuscript to include two variants of the Matern kernel**: Matern-3/2 (M3) and Matern-5/2 (M5). **We also conducted additional experiments with these kernels** and **the new results are given in Appendix G.1** of the revised manuscript. The results and conclusions remain consistent, showing that our method **outperforms all fixed kernel approaches** in terms of both mean and median regret.

---

> ### Author Response · Authors · 2024-11-22
> **Response to Reviewer BRo2 (3/4)**
>
> > In line 235-236, the authors mention a brief analysis will be returned by the LLM model. Can the authors provide one actual example of such analysis?
>
> Thank you for your insightful comment. **We have added a detailed discussion in Appendix F.2 to provide examples of LLM outputs**, including the requested analysis. Specifically, we present sample responses from the LLM for the crossover and mutation steps, **which are visualized in Figure 9**. In Figure 9a, given the parent kernels LIN and SE along with their BIC values, the LLM suggests combining them using the addition operator $(+)$ to minimize the BIC. The analysis explains the rationale behind this combination: the LIN kernel captures linear trends, while the SE kernel models smoothness. By combining these properties, the LIN $+$ SE kernel effectively represents both linearity and smoothness in the data. In Figure 9b, given the LIN $+$ SE kernel and its BIC, the LLM recommends replacing the SE kernel with the RQ kernel to achieve a lower BIC value. The analysis indicates that while the SE kernel assumes fixed smoothness, the RQ kernel introduces varying smoothness, allowing the model to be more suitable for data with fluctuating noise levels. The above results illustrate that **the LLM is familiar with kernel properties and their applications**. Based on its responses, the LLM demonstrates a **strong understanding of the relationships between different kernels and their capacity to address specific data characteristics**, enabling it to make informed suggestions for the kernel selection.
>
> ---
> > I'm a bit confused by the weighting strategy explained in line 242-243. Does it mean the most ideal kernel with very low BIC (thus low weight) and high acquisition value might not be selected while some worse kernel (higher BIC, lower acquisition value) might? Could the authors further justify the intuition behind such strategy?
>
> Thank you for raising this point. In our weighting strategy, the weight for each kernel $k$ is actually defined as follows: $$w_k = \frac{\exp(-\text{BIC}_k)}{{\sum_j
> \exp(-\text{BIC}_j)}},$$
>
> where $\mathrm{BIC}_k$ is the BIC of kernel $k$ and $\mathbb{K}$ denotes the set of kernels. The above definition indicates that **kernels with lower BIC values will have higher weights**, making them more likely to be selected, and **kernels with higher BIC values will be penalized**. We recognize that our previous explanation may not have clearly conveyed this rationale, so **we have revised Section 3 to include the above explicit definition** for improved clarity. Additionally, we want to justify the intuition behind our selection strategy, which is to **balance the kernel's ability to fit the data** (as indicated by $w_k$) **with the expected improvement at the proposed query point** (as measured by $\alpha$). This balance is particularly important for the situations where some kernels may indicate a good fit, but the actual improvement may not be as significant as promised.
>
> ---
> > In Figure 2, what is "Best" (the fourth method in legend)? Does it refer to the "Utility" method in line 411?
>
> Thank you for your careful observation regarding Figure 2. You are correct, the fourth legend refers to the "Utility" method. **We have corrected the typo and updated Figure 2 accordingly**.

---

> ### Author Response · Authors · 2024-11-22
> **Response to Reviewer BRo2 (4/4)**
>
> > How do the authors define the values of number of crossovers $n_c$, probability of mutation $p_m$, and population size $n_p$ (line 163-165 in Algorithm 1) in final implementation?
>
> Thank you for your question regarding the parameters used in our final implementation. In our experiments, we set the number of crossovers $n_c = 5$, the probability of mutation $p_m = 0.7$, and the population size $n_p = 10$. We found that these values provide **a good balance between maintaining a diverse pool of kernels while ensuring manageable computational complexity** for generating and evaluating the kernels. **We have added these details to Appendix D.1**, along with other additional information regarding the setup of CAKES.
>
> ----
> > I notice the authors set the maximum number of function evaluation $T$ and number of replications using different random seeds quite small for their experiments. For the benchmark functions (section 6.1), $T = 10d \leq 50$ since $\max d = 5$ for the chosen test functions. For the hyperparameter tuning tasks (section 6.2), $T = 25$. Usually the value of $T$ will be larger (e.g. 50, 100) in BO experiments. Similarly the number of random seeds for each experiment (10 for section 6.1, 5 for section 6.2 and not mentioned for section 6.3) is relatively less than what people usually use (e.g. 20, 25). How many random seeds did the author use for experiment in section 6.3? Is the reason for such not very generous setting due to long generation time of kernel design?
>
> Thank you for your insightful comments regarding the settings used in our experiments. We would like to clarify that the choice of $T = 10 \times d$ is simply **a rule of thumb for setting the maximum number of function evaluations**, which is also employed by [10]. Regarding the hyper-parameter tuning experiments in Section 6.2, we would like to mention that our setup includes a total of 60 tasks, comprising 12 OpenML datasets paired with 5 machine learning models. For each task, we conducted 5 seeded searches, with each search consisting of 25 trials, resulting in **a total of 7500 trials**. We believe this setup is **comprehensive enough** to identify good hyper-parameter configurations. For the photonic chip design in Section 6.3, we used $T = 250$ trials with 6 random seeds, resulting in **a total of 1500 trials**. We would like to mention that this experiment involves using a physical simulator, which requires a considerable amount of time and cost. We appreciate the reviewer's suggestion regarding the use of a larger number of random seeds. In response, **we have increased the number of random seeds for the benchmark function experiments in Section 6.1 to 20** and have **updated the results in Tables 1 and 5** accordingly.
>
> ---
> ### References
> [1] Simpson, I. Davies, V. Lalchand, A. Vullo, N. Durrande, and C. E. Rasmussen, "Kernel
> identification through transformers," *Advances in Neural Information Processing Systems*,
> vol. 34, pp. 10483–10495, 2021.
>
> [2] Yu, S. Zuo, H. Jiang, W. Ren, T. Zhao, and C. Zhang, “Fine-tuning pre-trained language
> model with weak supervision: A contrastive-regularized self-training approach,” *arXiv
> preprint arXiv:2010.07835*, 2020.
>
> [3] H. Zhang, G. Li, J. Li, Z. Zhang, Y. Zhu, and Z. Jin, “Fine-tuning pre-trained language
> models effectively by optimizing subnetworks adaptively,” *Advances in Neural Information
> Processing Systems*, vol. 35, pp. 21442–21454, 2022.
>
> [4] Kotha, J. M. Springer, and A. Raghunathan, “Understanding catastrophic forgetting in
> language models via implicit inference,” in *International Conference on Learning Representations, 2024.*
>
> [5] J. Wei, X. Wang, D. Schuurmans, M. Bosma, F. Xia, E. Chi, Q. V. Le, D. Zhou, et al.,
> “Chain-of-thought prompting elicits reasoning in large language models,” *Advances in Neural Information Processing Systems*, vol. 35, pp. 24824–24837, 2022.
>
> [6] S. M. Xie, A. Raghunathan, P. Liang, and T. Ma, “An explanation of in-context learning
> as implicit Bayesian inference,” *arXiv preprint arXiv:2111.02080*, 2022.
>
> [7] C. Han, Z. Wang, H. Zhao, and H. Ji, “Explaining emergent in-context learning as kernel
> regression,” *arXiv preprint arXiv:2305.12766*, 2023.
>
> [8] A. J. Smola and B. Sch¨olkopf, *Learning with kernels*, vol. 4. Citeseer, 1998.
>
> [9] D. Duvenaud, J. Lloyd, R. Grosse, J. Tenenbaum, and G. Zoubin, “Structure discovery in
> nonparametric regression through compositional kernel search,” in *International Conference on Machine Learning*, pp. 1166–1174, PMLR, 2013.
>
> [10] G. Malkomes and R. Garnett, “Automating Bayesian optimization with Bayesian optimization,” *Advances in Neural Information Processing Systems*, vol. 31, 2018.

---

> ### Comment · Reviewer_BRo2 · 2024-11-22
> **Response to authors**
>
> Thank you for the clarifications. With the modifications and additional details, I think the submission is more concrete now. I'm happy to increase my score. One additional comment for authors is to update kernels illustrated in $\textit{Initial Population}$ in Figure 1 to make it more consistent.

---

> > ### Author Response · Authors · 2024-11-22
> > **Reply to Reviewer BRo2**
> >
> > Thank you for your thoughtful feedback and for taking the time to carefully reassess our paper. We are pleased to hear that the revisions and clarifications we provided have addressed your concerns and improved your impression of the work.
> >
> > We also appreciate your suggestion regarding Figure 1 and have revised it accordingly to ensure consistency with the updated content in the paper.
> >
> > We truly value your perspective and would like to ensure that the paper meets the highest standards of ICLR. As such, we would greatly appreciate any additional suggestions or concerns you might have that could help us further improve the paper and strengthen its case for acceptance.
> >
> > We remain committed to enhancing the work and are happy to make any further revisions to address your feedback. Thank you again for your time and consideration.

---

> > > ### Author Response · Authors · 2024-11-29
> > > **Updates for Reviewer BRo2**
> > >
> > > Dear Reviewer BRo2,
> > >
> > > Thank you once again for your valuable time and insightful feedback on our work. We have incorporated the following updates, which we believe may be of interest to you:
> > >
> > > - **We conducted a prompt ablation study (see Appendix F.3) to analyze the different components of our prompt**. We found that **removing the domain-specific information (i.e., role assignment and kernel descriptions) does not negatively impact performance**. In fact, this variant still demonstrates **comparable performance** to the standard CAKES setup, suggesting that the **LLM possesses valuable prior knowledge that can help with kernel design**. However, when we remove the optimization history (i.e., observations and fitness values), the performance of this variant **degrades significantly**, indicating that **the optimization history is important and that the LLM can leverage this information effectively**. Finally, we considered a variant that excludes the explicit task instructions and kernel generation guidelines (e.g., how to apply crossover and mutation operators). We observed that this variant exhibits **significantly poorer performance**, primarily due to its **inability to generate valid kernels**. Overall, these findings demonstrate the importance of each component in our prompt to achieve a superior performance.
> > > - As requested by the other reviewers, **we conducted additional experiments using a larger number of iterations $T$ and repetitions**. Specifically, **we evaluated our method on higher-dimensional functions**, with dimensions  $d \in \{10, 20, 30\}$. For these experiments, we set $T = 10 \times d$ **(resulting in $T = 100, 200, 300)$ and repeated the experiments across 20 random seeds**. In Appendix G.3, we detail our tuning of a robot-pushing controller with $d = 14$, where we set **$T=100$ and conducted 10 repetitions using different random seeds**. Additionally, we tuned a lunar lander controller with $d=12$, **setting $T = 250$ and repeating the experiments over 50 random terrains with varying initial positions and velocities**. We believe that these additions enhance the comprehensiveness of our experiments.
> > >
> > > We hope that above updates enhance your impression of our work and strengthen our contributions. If you any additional suggestions or concerns, we would be happy to discuss them further. If your concerns have been adequately addressed, we would greatly appreciate if you could reconsider your score.
> > >
> > > With thanks,
> > >
> > > Authors of #5709

---

> > > > ### Author Response · Authors · 2024-12-02
> > > > **Follow-up for Reviewer BRo2**
> > > >
> > > > Dear respected Reviewer BRo2,
> > > >
> > > > As the author-reviewer discussion phase will be ending soon, we would appreciate your feedback on whether our responses have adequately addressed your concerns. If you have any additional issues you would like to discuss, please kindly let us know. If you find that your concerns have been sufficiently resolved, we kindly request that you reconsider your score. Thank you once again for your time and effort in reviewing our paper.
> > > >
> > > > With thanks,
> > > >
> > > > Authors of #5709

---

### Official Review · Reviewer_wHHf · 2024-11-03

**Soundness:** 2
**Presentation:** 3
**Contribution:** 2
**Rating:** 6
**Confidence:** 3

**Summary:**

The paper presents Context-Aware Kernel Search (CAKES), a novel method for optimizing Bayesian optimization (BO) by leveraging large language models (LLMs) to automate the design of Gaussian process (GP) kernels. CAKES addresses the challenge of sub-optimal kernel selection, which can hinder the efficiency of BO, by using LLMs as genetic operators to adaptively generate and refine kernels based on observed data. Theoretical analysis shows that CAKES achieves sub-linear regret, and experimental results indicate it outperforms existing methods across various optimization tasks, including hyperparameter tuning and photonic chip design, significantly improving performance and reducing design cycle times.

**Strengths:**

One of the key strengths of this paper is its innovative use of large language models (LLMs) for in-context learning to improve Bayesian optimization (BO). The Context-Aware Kernel Search (CAKES) method leverages LLMs' few-shot learning capabilities to adaptively generate and refine Gaussian process kernels based on observed data, without requiring any fine-tuning.

Theoretical analysis indicates that CAKES achieves sub-linear regret relative to the budget for any input dimension. Experimental results demonstrate its superiority over baseline methods in various optimization tasks, including benchmark functions and hyperparameter tuning and photonic chip design.

**Weaknesses:**

Limited testing in high-dimensional spaces: The paper does not demonstrate the effectiveness of CAKES on high-dimensional optimization problems. This leaves uncertainty about how well the method scales to more complex search spaces with many variables.

**Questions:**

Same in weaknesses part

---

> ### Author Response · Authors · 2024-11-22
> **Response to Reviewer wHHf (1/1)**
>
> Thank you for your time and effort in reviewing our work. Important changes have been highlighted with blue color in the manuscript. We respond to the questions and comments point-by-point as follows.
>
> ---
> > Limited testing in high-dimensional spaces: The paper does not demonstrate the effectiveness of CAKES on high-dimensional optimization problems. This leaves uncertainty about how well the method scales to more complex search spaces with many variables.
>
> Thank you for your valuable feedback regarding the validation of our method in high-dimensional spaces. In response to your comments, **we have conducted additional experiments using the Ackley and Levy functions with higher input dimensions**, specifically $d \in$ {10, 20, 30}. We chose these two functions as they are **highly non-convex** and **possess numerous local minima**, making them particularly challenging to optimize. We set $T = 10 \times d$ and repeat each experiment using 20 different random seeds. The results, presented in Table 6 of Appendix G.1, demonstrate that **CAKES consistently outperforms the other methods** by minimizing the regret, even as $d$ increases. This corroborates that **our method can scale effectively** in more complex search spaces.
>
> |   |    SE    |    M5    |   Random  |  Utility  |    BIC   |    DGP   |    EGP   |   CAKES   |
> |:--------------:|:-------:|:-------:|:--------:|:---------:|:-------:|:-------:|:-------:|:---------:|
> |    Ackley-2    |  0.1773  |  0.1220  |  0.1358  |  0.1062   |  0.1863  |  0.2510  |  0.1878  | **0.0783** |
> |    Ackley-5    |  0.3185  |  0.2369  | **0.1722** |  0.2278   |  0.3590  |  0.3110  |  0.2285  |  0.1732   |
> |   Ackley-10    |  0.3645  |  0.3225  |  0.2725  |  0.4122   |  0.3157  |  0.4557  |  0.3108  | **0.2356** |
> |   Ackley-20    |  0.4523  |  0.4152  |  0.5103  |  0.5509   |  0.5021  |  0.5043  |  0.4237  | **0.3281** |
> |   Ackley-30    |  0.5331  |  0.5114  |  0.6460  |  0.5966   |  0.5664  |  0.6056  |  0.6210  | **0.4596** |
> |    Levy-2      |  0.1562  |  0.0418  |  0.0835  | **0.0255** |  0.1145  |  0.1965  |  0.0765  |  0.0353   |
> |    Levy-3      |  0.1141  |  0.1422  |  0.1495  |  0.0880   |  0.1125  |  0.2265  | 0.0805 | **0.0505** |
> |   Levy-10      |  0.3215  |  0.2514  |  0.2310  |  0.1545   |  0.1940  |  0.2845  | 0.1069 | **0.0941** |
> |   Levy-20      |  0.3657  |  0.4061  |  0.3815  |  0.1905   |  0.2762  |  0.3450  | 0.1703 | **0.1208** |
> |   Levy-30      |  0.4205  |  0.4598  |  0.4515  |  0.2405   |  0.3147  |  0.4534  | 0.2040 | **0.1483** |

---

> > ### Comment · Reviewer_wHHf · 2024-11-22
> >
> > Thank you for running CAKES on high dimensional test functions. I am impressed by the empirical perfomance of the CAKES method on the 10,20,30 dimensions. I will increase my score.

---

> > > ### Author Response · Authors · 2024-11-22
> > > **Reply to Reviewer wHHf**
> > >
> > > Thank you for your thoughtful comments and for taking the time to review our revised submission. We greatly appreciate your positive feedback on the empirical performance of our method and are glad to hear that our revisions have addressed your concerns.
> > >
> > > As we remain dedicated to further enhancing the quality and impact of our submission, we would greatly value any additional suggestions or specific aspects you feel could further strengthen our work.
> > >
> > > Thank you once again for your time and support.

---

> > > > ### Author Response · Authors · 2024-11-29
> > > > **Updates for Reviewer wHHf**
> > > >
> > > > Dear Reviewer wHHf,
> > > >
> > > > Thank you once again for your valuable time and insightful feedback on our work, particularly your comments regarding the validation of our method in high-dimensional spaces.
> > > >
> > > > In addition to running CAKES on high dimensional test functions as per your suggestion, **we have conducted two additional experiments on real world tasks** that we believe will be of particular interest to you. Specifically, **we applied our method to tune controllers for robot pushing tasks ($d = 14$, see Appendix G.4) and lunar lander tasks ($d = 12$, see Appendix G.5)**. Our method consistently achieved **the highest rewards** and **the lowest regret**, demonstrating its effectiveness on tasks with more complex search spaces.
> > > >
> > > > We hope that above updates enhance your impression of our work and strengthen our contributions. If you any additional suggestions or concerns, we would be happy to discuss them further. If your concerns have been adequately addressed, we would greatly appreciate if you could reconsider your score.
> > > >
> > > > With thanks,
> > > >
> > > > Authors of #5709

---

> > > > > ### Author Response · Authors · 2024-12-02
> > > > > **Follow-up for Reviewer wHHf**
> > > > >
> > > > > Dear respected Reviewer wHHf,
> > > > >
> > > > > As the author-reviewer discussion phase will be ending soon, we would appreciate your feedback on whether our responses have adequately addressed your concerns. If you have any additional issues you would like to discuss, please kindly let us know. If you find that your concerns have been sufficiently resolved, we kindly request that you reconsider your score. Thank you once again for your time and effort in reviewing our paper.
> > > > >
> > > > > With thanks,
> > > > >
> > > > > Authors of #5709

---

### Official Review · Reviewer_4qNv · 2024-11-03

**Soundness:** 2
**Presentation:** 3
**Contribution:** 2
**Rating:** 5
**Confidence:** 4

**Summary:**

- This work proposes a large language model-based kernel design method for Bayesian optimization.
- Since the performance of Bayesian optimization depends on kernel choice, the kernel search is crucial.
- This work utilizes a large language model to select kernel design.
- The authors test their method in diverse benchmark functions.

**Strengths:**

- It shows an ability of large language models to understand natural language description in the problem of kernel search.
- Paper is generally well-written.

**Weaknesses:**

- More real-world benchmarks are missing.
- Some baseline results of photonic chip design might be missing.
- More recent baselines are missing.

**Questions:**

- My concerns are all related to experiments.
- I think that hyperparameter tuning tasks are involved with small simple models.  If more sophisticated models are tuned, it would be better.
- I don't know why only two baseline methods are compared to CAKES in photonic chip design.
- Some baselines such as (Malkomes and Garnett, 2018) should be compared.
- I think that few-shot learning is not correct in this context.  The authors didn't fine-tune large language models.  It should be few-shot prompting.
- Why did you use gpt-4o-mini only?  If other large language models such as gpt-4o and llama-3 are used, can the empirical results change?  And why did you choose a temperature of 0.7 and a top_p of 0.95 specifically?

**Details Of Ethics Concerns:**

There are no ethics concerns for this work.

---

> ### Author Response · Authors · 2024-11-22
> **Response to Reviewer 4qNv (1/3)**
>
> Thank you for your time and effort in reviewing our work. Important changes have been highlighted with blue color in the manuscript. We respond to the questions and comments point-by-point as follows.
>
> ---
> > I think that few-shot learning is not correct in this context. The authors didn't fine-tune large language models. It should be few-shot prompting.
>
> Thank you for your feedback regarding the terminology used in our manuscript. We agree that the term "few-shot prompting" is more appropriate in this context. In response, **we have revised the manuscript to replace "few-shot learning" with "few-shot prompting"** throughout the relevant sections.
>
> ---
> > More real-world benchmarks are missing. I think that hyperparameter tuning tasks are involved with small simple models. If more sophisticated models are tuned, it would be better.
>
> Thank you for your feedback on the importance of conducting real-world benchmarks and including more sophisticated tasks in our work. In response to your suggestions, **we have conducted two additional experiments in Appendix G that involve tuning a controller** for the following tasks:
>
> - Robot pushing $(d = 14)$: This task was initially introduced in [1], which involves tuning a controller for two robotic hands to push objects to target positions. The objective is to minimize the total distance from the initial and final positions of the objects to their respective goal positions, thus maximizing the reward.  We ran simulations for $T = 100$ iterations with 10 random seeds. **The results, summarized in Figure 12 of Appendix G.4**, show that **CAKES outperformed other methods** in achieving higher rewards and minimizing the regret.
> - Lunar lander $(d = 12)$: In this task, we consider tuning a controller for a lunar lander in [OpenAI Gym](https://www.gymlibrary.dev/environments/box2d/lunar_lander). The objective is to land the lunar module between two flags on the landing pad without crashing, while minimizing the engine use. We ran the simulations for $T = 250$ iterations using 50 randomly generated terrains, initial positions, and velocities. As shown in Figure 13 of Appendix G.5, **CAKES consistently achieved or exceeded the target objective of 200 points** while minimizing the penalties and optimizing the landing strategies.
>
> We believe these additional experiments have demonstrated the effectiveness of our method in **more sophisticated, real-world tasks**.

---

> ### Author Response · Authors · 2024-11-22
> **Response to Reviewer 4qNv (2/3)**
>
> > More recent baselines are missing. Some baselines such as (Malkomes and Garnett, 2018) should be compared.
>
> Thank you for your feedback regarding the inclusion of more recent baselines for comparison. We focused on GP-based BO methods as the primary baselines because we believe these are **the most relevant comparisons to our method**, which specifically addresses kernel design. As emphasized in Section 1, one of our contributions is to highlight the urgent need for a **more careful treatment of kernel design** in BO. We argue that conventional setups, which often rely on fixed kernels or naive selection strategies, do not perform uniformly well across different tasks. Still, we appreciate the reviewer's suggestion to include additional recent baselines, such as those from [2]. In response, **we have incorporated comparisons against the following additional methods**:
>
> - **Bag-of-models (BOM)** [3]: This method maintains a collection of kernels, each capturing different structures that best explain the observed data.
> - **Markov Chain Monte Carlo (MCMC)** [4]: This method uses a Metropolis-Hastings algorithm to randomly sample kernels from a grammar based on a proposal distribution.
> - **Automated BO (ABO)** [2]: This method utilizes BO to dynamically select kernels that best explain the observed data.
>
> **We provide additional details regarding these baselines and the experimental setup in Appendix G.1**. The results, summarized in Table 5, indicate that our proposed method, CAKES, **outperforms all the baselines in terms of mean and median regret**. Notably, CAKES **ranked the first in 12 instances out of 15** and **secured a top-two ranking in all instances**.
>
> | Function              | BOM     | MCMC    | ABO     | CAKES   |
> |:----------------------:|:---------:|:---------:|:---------:|:---------:|
> | Ackley-2             | 0.1016  | 0.1012  | 0.1020  | **0.0783** |
> | Ackley-5             | 0.5907  | **0.1312** | 0.1910  | 0.1733   |
> | Beale                | 0.3559  | 0.4040  | 0.3118  | **0.2570** |
> | Branin               | 0.0499  | 0.0301  | 0.0101  | **0.0071** |
> | Dropwave             | 0.5040 | 0.5788  | 0.5529  | **0.4693** |
> | Eggholder            | 0.4709  | 0.4536  | 0.4210 | **0.1242** |
> | Griewank-2           | 0.0460  | 0.0589  | 0.0357 | **0.0266** |
> | Griewank-5           | 0.2437  | 0.0258 | 0.0320  | **0.0189** |
> | Hartmann             | 0.0299 | **0.0001** | **0.0001** | **0.0001** |
> | Levy-2               | 0.0626  | 0.0668  | 0.0519 | **0.0352** |
> | Levy-3               | 0.0867  | 0.0580  | **0.0290** | 0.0507 |
> | Rastringin-2         | 0.3618  | 0.3722  | 0.3420 | **0.3342** |
> | Rastringin-4         | 0.3177 | 0.3285  | 0.3179  | **0.3125** |
> | Rosenbrock           | 0.0901 | 0.0907  | 0.0901 | **0.0484** |
> | Six-Hump Camel       | 0.1102  | 0.1071  | **0.1002** | 0.1013 |
> | Mean regret      | 0.2281  | 0.1871  | 0.1725  | **0.1358** |
> | Median regret    | 0.1102  | 0.1012  | 0.1002  | **0.0783** |
>
> ---
> > Some baseline results of photonic chip design might be missing. I don't know why only two baseline methods are compared to CAKES in photonic chip design.
>
> Thank you for your valuable feedback. We would like to first clarify the current baselines we included are **BO-based methods**. To the best of our knowledge, these were the only two GP-based BO methods specifically applied to this problem in the literature. However, we appreciate your suggestion to include additional baselines in this experiment. In response, **we have added more baseline methods from the literature**, specifically those published in top optics and photonics journals, as detailed in Appendix G.3. **The results of these comparisons are summarized in Table 8**. Our findings demonstrate that the chip designed by CAKES **remains competitive or even superior to these additional baselines.**
>
> | Method                     | Q-factor          | Wavelength | Lasing area         | Power | Divergence |
> |---------------------------|-------------------|------------|---------------------|-------|------------|
> | Ohnishi et. al (2004)    | NA                | 959.4     | $2.8 \times 10^{-9}$ | NA    | 1.1        |
> | Hsu et. al (2017) | $5.0 \times 10^3$ | 1299      | $1.8 \times 10^{-8}$ | NA    | 2.0        |
> | Chen et. al (2021) | $2.9 \times 10^3$ | 948       | $6.2 \times 10^{-9}$ | NA    | 0.75       |
> | Ours                      | $2.5 \times 10^7$ | 1309.3    | $1.3 \times 10^{-13}$ | 0.99  | 1.2        |

---

> ### Author Response · Authors · 2024-11-22
> **Response to Reviewer 4qNv (3/3)**
>
> > Why did you use gpt-4o-mini only? If other large language models such as gpt-4o and llama-3 are used, can the empirical results change? And why did you choose a temperature of 0.7 and a top\_p of 0.95 specifically?
>
> Thank you for your feedback regarding our choice of LLM and parameter settings. As we elaborated in Appendix D.1, we selected gpt-4o-mini for our implementation because it provides **a good balance of affordability, fast inference, and intelligence**. While we acknowledge that other LLMs, such as gpt-4o and Llama-3, may yield different empirical results, we believe that **the overall methodology and key insights presented in our work extend beyond the specifics of any particular model**. This is further discussed in Appendix H, where we emphasize the general applicability of our method. Regarding the choice of parameters, we specified a temperature of 0.7 and top\_p of 0.95 to strike **an optimal balance between generating relevant outputs and introducing diversity**. As mentioned in Appendix D.1, these settings were determined based on our large-scale experimental results.
>
> ---
> ### References
> [1] Wang, C. Gehring, P. Kohli, and S. Jegelka, “Batched large-scale Bayesian optimization in high-dimensional spaces,” in *International Conference on Artificial Intelligence and Statistics*, pp. 745–754, PMLR, 2018.
>
> [2] Malkomes and R. Garnett, “Automating Bayesian optimization with Bayesian optimization,” *Advances in Neural Information Processing Systems*, vol. 31, 2018.
>
> [3] K. Kandasamy, J. Schneider, and B. Poczos, “High dimensional Bayesian optimisation and bandits via additive models,” in *International conference on machine learning*, pp. 295–304, PMLR, 2015.
>
> [4] J. Gardner, C. Guo, K. Weinberger, R. Garnett, and R. Grosse, “Discovering and exploiting additive structure for Bayesian optimization,” in *Artificial Intelligence and Statistics*, pp. 1311–1319, PMLR, 2017.
>
> [5] D. Ohnishi, T. Okano, M. Imada, and S. Noda, “Room temperature continuous wave operation of a surface-emitting two-dimensional photonic crystal diode laser,” *Optics Express*, vol. 12, no. 8, pp. 1562–1568, 2004.
>
> [6] M.-Y. Hsu, G. Lin, and C.-H. Pan, “Electrically injected 1.3-μm quantum-dot photonic-crystal surface-emitting lasers,” *Optics Express*, vol. 25, no. 26, pp. 32697–32704, 2017.
>
> [7] L.-R. Chen, K.-B. Hong, K.-C. Huang, H.-T. Yen, and T.-C. Lu, “Improvement of output efficiency of p-face up photonic-crystal surface-emitting lasers,” *Optics Express*, vol. 29, no. 7, pp. 11293–11300, 2021.

---

> > ### Author Response · Authors · 2024-11-29
> > **Updates for Reviewer 4qNv**
> >
> > Dear Reviewer 4qnV,
> >
> > Thank you once again for your valuable time and insightful feedback on our work. We have carefully considered your comments and made the following revisions to address your concerns:
> >
> > - As per your suggestion, **we have replaced "few-shot learning" with "few-shot prompting"** throughout the relevant sections of the manuscript to better reflect our methodology.
> > - In response to your feedback regarding the inclusion of more real-world benchmarks and sophisticated tasks, **we have conducted two additional experiments** that involve tuning a controller for robot pushing (see Appendix G.4) and for lunar lander tasks (see Appendix G.5). **Our method consistently achieved the highest rewards and the lowest regret**, demonstrating the effectiveness of our method in real-world, sophisticated tasks.
> > - We appreciate your recommendation to include additional recent baselines, such as those from Malkolmes and Garnett (2018). Accordingly, **we have added comparisons against the following methods**: Bag-of-Models (BOM), Markov Chain Monte Carlo (MCMC), and Automated Bayesian optimization (ABO). The results, summarized in Table 6, show that **CAKES outperforms all baselines in terms of mean and median regret**.
> > - For the photonic chip design experiment, **we have included additional baseline methods from the literature, specifically those published in leading optics and photonics journals**, as detailed in Appendix G.3. The results presented in Table 8 indicate that the chip designed by **CAKES remains competitive or even superior to these additional baselines**.
> > - In response to your comments regarding our choice of LLM, **we have conducted an additional experiment in Appendix F.4 to analyze the impact of different LLMs** on the performance of our method. **We considered several state-of-the-art language models**, including Meta's `llama-3.1-8b`, Google's `gemini-1.5-flash`, Anthropic's `claude-3.5-haiku`, and OpenAI's `gpt-4o-mini`. The results support our decision to choose `gpt-4o-mini` as the default model for CAKES, as it provides an **excellent balance of cost, speed, and intelligence**. While different LLMs may yield varying empirical results, we believe that **the overall methodology and key insights presented in our work extend beyond the specifics of any particular model**.
> > - Regarding the parameter choices, we specified a temperature of 0.7 and a top_p of 0.95 to achieve an **optimal balance between generating relevant outputs and introducing diversity**, based on our large-scale experimental results. In light of your comments, **we have included these details in Appendix D.1**.
> >
> > We hope that the above revisions satisfactorily address your concerns. If you find that any of your concerns remain unresolved, we would be happy to discuss them further. If your concerns have been adequately addressed, we would greatly appreciate if you could reconsider your score.
> >
> > With thanks,
> >
> > Authors of #5709

---

> > > ### Author Response · Authors · 2024-12-02
> > > **Follow-up for Reviewer 4qNv**
> > >
> > > Dear respected Reviewer 4qNv,
> > >
> > > As the author-reviewer discussion phase will be ending soon, we would appreciate your feedback on whether our responses have adequately addressed your concerns. If you have any additional issues you would like to discuss, please kindly let us know. If you find that your concerns have been sufficiently resolved, we kindly request that you reconsider your score. Thank you once again for your time and effort in reviewing our paper.
> > >
> > > With thanks,
> > >
> > > Authors of #5709

---

### Author Response · Authors · 2024-11-24
**New Ablation Study**

We thank the reviewers for their time and effort in reviewing our work. In response to the reviewers' feedback on our prompt design and LLM-related analysis, **we have conducted a new ablation study in Section F.3** to systematically evaluate the impact of each component in our prompt. Specifically, **we have highlighted the key components of our prompt**, which include:
- **Domain information:** This includes the role assignment as well as description of kernel functions and their properties.
- **Problem context:** This includes the optimization history, which consists of the observations as well as the fitness values.
- **Task instruction:** This includes the guidelines on how to generate the kernels based on the kernel grammar.

Based on the above components, we conduct an ablation study using the following configurations:
- **CAKES:** This is the vanilla CAKES setup employed in our experiments, incorporating all three components.
- **CAKES-NoInfo:** This variant assesses the impact of domain-specific information on performance. Specifically, it omits the role assignment and kernel descriptions while keeping the other components (see Figure 15).
- **CAKES-NoContext:** This variant evaluates the importance of problem context by removing the optimization history (i.e., the observations and fitness values), while keeping the other components  (see Figure 16).
- **CAKES-NoInstruct:** This variant excludes task instructions and omits the explicit kernel generation guidelines (i.e., how to use the crossover and mutation operators), while keeping the other components (see Figure 17).

Our ablation revealed that:
- **Standard CAKES setup outperforms the other variants**, highlighting the importance of each prompt component in enhancing overall performance.
- **CAKES-NoInfo demonstrated similar optimization performance to CAKES** despite lacking domain-specific information (i.e., the role assignment and kernel descriptions in the prompt). This suggests that the **LLM may already possess valuable prior knowledge**, which is likely derived from its pre-training on a vast amount of statistical information and code.
- **CAKES-NoContext exhibited worse performance than CAKES**, indicating that **the optimization history is crucial** and that the LLM effectively leverages this information.
- **CAKES-NoInstruct performed poorly compared to the other variants**. To investigate this, we examined the percentage of valid kernels, defined as the proportion of kernels that satisfy the kernel properties. We found that **CAKES-NoInstruct generated valid kernels only 68.72% ($\pm$ 0.45%) of the time, while the other variants consistently produced valid kernels**. This is due to **the absence of the generation guidelines from the kernel grammar**, which underscores its importance in the kernel generation process.

We hope this additional study enhances the quality of our paper and strengthens its case for acceptance. We remain open to any further feedback from the reviewers.


With thanks,

Authors of #5709

---

### Author Response · Authors · 2024-11-26
**New LLM-Related Analysis**

We thank the reviewers once again for taking time in reviewing our submission. In response to the reviewers’ comments and requests to justify our choice of LLM and provide more LLM-related analysis, **we have conducted an additional experiment in Appendix F.4 to analyze the impact of the LLM choice** on the performance of our method. Specifically, **we consider the following state-of-the-art models**: Meta's `llama-3.1-8b`, Google's `gemini-1.5-flash`, Anthropic's `claude-3.5-haiku`, and OpenAI's `gpt-4o-mini`. We evaluate these models on the benchmark functions considered in Section 6.1.

Based on the results presented in Table 4, we can observe that `gpt-4o-mini` **outperforms the other models** in 8 out of 15 test functions. `claude-3.5-haiku` also demonstrates **competitive performance**, securing the best results in 6 out of 15 test functions. Moreover, as of November 2024, `gpt-4o-mini` is priced at 0.15 USD per million input tokens and 0.60 USD per million output tokens, making it **significantly more affordable** than `claude-3.5-haiku`, which costs 1 USD per million input tokens and 5 USD per million output tokens. In terms of speed, `gpt-4o-mini` delivers an output speed of approximately 104.5 tokens per second, **outperforming** `claude-3.5-haiku`'s 64.3 tokens per second.

Considering the above metrics, we choose `gpt-4o-mini` as the default model in CAKES as it offers **an excellent balance of cost, speed, and intelligence for our implementation**. We hope that this additional analysis enhances the comprehensiveness and overall quality of our work.

With thanks,

Authors of #5709


| Function                | llama-3.1-8b | gemini-1.5-flash | claude-3.5-haiku | gpt-4o-mini   |
|-------------------------|---------------|-------------------|-------------------|----------------|
| Ackley-2                | 0.1019        | 0.0916            | 0.0812        | **0.0783**     |
| Ackley-5                | 0.2106        | 0.1907            | **0.1621**        | 0.1732     |
| Beale                   | 0.2621        | 0.2595            | 0.2640            | **0.2565**     |
| Branin                  | 0.0131        | 0.0199            | 0.0091        | **0.0070**     |
| Dropwave                | 0.4941        | **0.4640**        | 0.4878            | 0.4690     |
| Eggholder               | 0.1323        | 0.1309            | **0.1156**        | 0.1241     |
| Griewank-2             | 0.0337        | 0.0360            | 0.0289       | **0.0267**     |
| Griewank-5             | 0.0301        | 0.0247            | **0.0158**        | 0.0185     |
| Hartmann                | 0.0012        | 0.0005        | **0.0001**        | **0.0001**     |
| Levy-2                  | 0.0459        | 0.0426            | 0.0368        | **0.0353**     |
| Levy-3                  | 0.0710        | 0.0667            | **0.0480**        | 0.0505     |
| Rastringin-2           | 0.3595        | 0.3611            | **0.3322**        | 0.3341     |
| Rastringin-4           | 0.3301        | 0.3147        | 0.3253            | **0.3128**     |
| Rosenbrock              | 0.0623        | 0.0501       | 0.0507            | **0.0483**     |
| Six-Hump Camel          | 0.1103        | **0.1002**        | 0.1073            | 0.1015     |

---

### Author Response · Authors · 2024-12-03
**Rebuttal Summary**

We thank the reviewers for their constructive feedback and for recognizing the contributions
of our work.

We are encouraged that the reviewers found CAKES to address a **compelling
topic** in Gaussian process modeling and Bayesian optimization (BO), with its kernel design
deemed **crucial** (`Bsdq`, `BRo2`, `4qNv`). Our integration of large language models (LLMs) for kernel
search was appreciated as **innovative**, with the usage of a pre-trained LLM without fine-tuning
considered **novel and impactful** in the BO setting (`BRo2`, `wHHf`). We are pleased that the reviewers
found the paper **well-written, easy to follow, and systematic** in its structure, with a **strong motivation** for
the proposed surrogate model design (`Bsdq`, `BRo2`, `4qNv`). Experiments demonstrating CAKES'
effectiveness across synthetic benchmarks and real-world applications, such as hyperparameter
tuning and photonic chip design, were highlighted as showcasing its **practical potential** (`BRo2`,
`wHHf`). Theoretical analysis showing sub-linear regret relative to the budget further reinforced
its **strength** (`wHHf`).

We have also taken the reviewers' feedback into account and made the following key changes
to improve the paper:

- **Provided LLM-related analysis**, including analysis of the kernels' fitness distribution (Appendix
F.1), sample outputs from the LLMs (Appendix F.2), prompt ablation study (Appendix
F.3), and comparison of different LLMs (Appendix F.4).
- **Standardized the experiment design** by replacing the GAP metric with the regret across
all experiments (Section 6, Appendix G).
- **Conducted additional experiments to include two variants of the Matern kernel**: Matern-
3/2 and Matern-5/2 (Appendix G.1).
- **Added the explicit formulation** for the weighting strategy (Section 3).
- **Increased the number of random seeds** for the benchmark function experiments to 20
(Section 6.1).
- **Conducted additional experiments** using the Ackley and Levy functions **with higher input
dimensions**, i.e., $d \in$ {10, 20, 30} (Appendix G.1)
- **Included two new real-world benchmarks** that involve tuning a controller for robot pushing
(Appendix G.4) and lunar lander (Appendix G.5).
- **Added comparisons against three recent BO baselines**, BOM, MCMC, and ABO (Appendix
G.1).
- **Included additional baseline methods from the literature** for the photonics chip design
experiment (Appendix G.3).

We hope these revisions adequately address the reviewers’ concerns. We remain open to further feedback.

With thanks,

Authors of #5709

---

### Note · Authors · 2025-02-03

I have read and agree with the venue's withdrawal policy on behalf of myself and my co-authors.